# UNTRAINED NETWORKS' CLASS BIAS: A THEORETICAL INVESTIGATION

## ABSTRACT

The initial state of neural networks plays a central role in conditioning the subsequent training dynamics. In the context of classification problems, we provide a theoretical analysis demonstrating that the structure of a neural network can condition the model to assign all predictions to the same class, even before the beginning of training, and in the absence of explicit biases. We show that the presence of this phenomenon, which we call "Initial Guessing Bias" (IGB), depends on architectural choices such as activation functions, max-pooling layers, and network depth. Our analysis of IGB has practical consequences, in that it guides architecture selection and initialization. We also highlight theoretical consequences, such as the breakdown of node-permutation symmetry, the violation of self-averaging, the validity of some mean-field approximations, and the non-trivial differences arising with depth.

## 1 INTRODUCTION

We study how untrained neural networks assign data points to different classes before they start learning. In the absence of explicit bias weights, our intuition might suggest an even partitioning between classes. Instead, we demonstrate that the network architecture can induce an imbalance effect, resulting in a large fraction of data points being classified as a single class. We call this phenomenon *Initial Guessing Bias* (IGB) which we illustrate in Fig. 1.

Importantly, the phenomenon we describe appears at initialization, which can have a significant impact on the training process. Indeed, the initial distribution of the weights can, for example, determine an amplification/decay of the signal coming from the input, or even limit the depth to which signals can propagate through random neural networks (Schoenholz et al., 2016; Hanin & Rolnick, 2018; Glorot & Bengio, 2010; Saxe et al., 2013). The influence of the initial weights distribution is of even greater importance in over-parametrized networks, for which, weights trained with gradient-based optimizers do not move far away from initialization (Jacot et al., 2018; Du et al., 2019; 2018; Allen-Zhu et al., 2019; Chizat et al., 2019). While the study of the initial state of neural networks has received increasing attention over the past few years, our study differs from past work in some important key aspects which we detail next.

**Sources of randomness** A deep neural network (DNN) at initialization can be interpreted as a random function parameterized by random weights, and whose inputs are sampled from an unknown data distribution. There are thus two distinct and independent sources of randomness at initialization: weights and data. Previous theoretical studies of DNNs typically consider, for a given input, the whole ensemble of random initializations of the weights (Poole et al., 2016; Matthews et al., 2018; Novak et al., 2018). In contrast, we fix the weight initialization and study the network's behavior by taking expectations over the data. Technically speaking, while previous work first averages over the weights and then over the data, our averages are first over the data and then over the weights. Note that this approach is closer to the natural order followed in practice, where data is classified by a single neural network (and thus for a single weight realization).

**Breaking of self-averageness** The mentioned inversion might seem a technicality but it actually constitutes a fundamental point in the study of IGB. The observable that characterizes the phenomenon is the fraction of points classified as class $c$, and denoted by $f_c(\mathcal{W})$, (this quantity will be formally introduced shortly) in fact does not respect the self-averaging property (Mézard et al., 1987; Dotsenko,

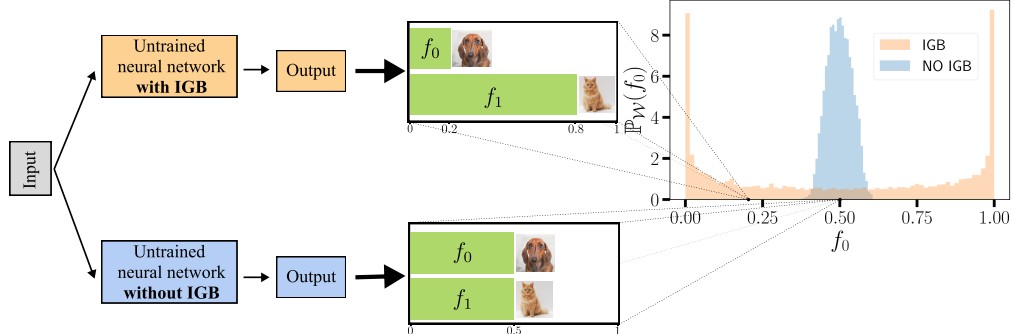

Figure 1: Initial Guessing Bias (IGB). Consider a task where we classify a binary dataset using an untrained network. Does it assign half of the examples to each class, or does it privilege one class? The answer depends on the architecture. In the bottom left, we classify a binary dataset with an untrained network *without IGB*. This model will generally assign half of the examples to each class (histogram on the bottom center). In the top left, we classify the same dataset using an untrained network *with IGB*. In this case, most of the guesses will usually go to one of the two classes (histogram on the top center). As an example, we take the dog/cat classes (label 0 / label 1) from CIFAR10, and pass them through an untrained CNN with 2 layers, each followed by pooling. The non-IGB model uses tanh activations and average pooling, the IGB model uses ReLU and Max pooling. We show in the top-right the distribution over different initializations, $\mathbb{P}_{\mathcal{W}}(f_0)$, of the fraction $f_0$ of times that each model guessed dog (equivalently, $f_1 = 1 - f_0$ indicates the fraction of images guessed as cat). While for the non-IGB models, $f_0$ is most often 50%, with IGB it most often is either 0% or 100%.

1994).[1] In other words, even in the infinite dataset/network size limit, the value of $f_0(\mathcal{W})$ obtained on a given realization of the network initialization, $\mathcal{W}$, differs from that computed from the average over the ensemble of initializations. Self-averaging is often exploited (*e.g.* in *dynamical mean field theory* (Sompolinsky et al., 1988), employed also in the context of mean-field theories of deep learning (Schoenholz et al., 2016; Poole et al., 2016)), as it can lead to a simplification of the analysis. For the phenomena related to IGB we cannot exploit self-averaging.

**Nodes Symmetry Breaking**   When we think of an untrained multi-layer perceptron (MLP), we are naturally inclined to assume a permutation symmetry among the various nodes of a given layer. Instead, we will show that this symmetry can be broken, and indeed, this symmetry breaking (*i.e.* difference in the distribution of nodes in a given layer) constitutes the foundation of the IGB. Nodes Permutation Symmetry Breaking (PSB) was already reported in previous work, *e.g.* in specific shallow networks with a large number of examples (Kang et al., 1993).
We will see that the choice of architecture significantly influences the presence of IGB. Architecture design, particularly the selection of activation functions, has been extensively studied (Dubey et al., 2022), with a focus on ReLU versus differentiable activations such as sigmoid. For instance, sigmoid activations can achieve dynamical isometry, maximizing signal propagation depth (Schoenholz et al., 2016), unlike ReLUs (Pennington et al., 2017). These activations are also compared in terms of generalization performance, revealing distinct behaviors for ReLUs and sigmoids (Oostwal et al., 2021). Notably, these studies often consider averaging over weight initializations.

## 2   MAIN CONTRIBUTIONS

Our main contribution is a new analysis that characterizes the conditions that trigger the IGB phenomenon. More precisely we demonstrate that, beyond effects that may be induced by the structure of the data itself, it is the design of the architecture that plays a crucial role.

---

[1]In a system defined over an ensemble of realizations (in our case, each realization is a different weight initialization), a self-averaging quantity is one that can be equivalently calculated either by averaging over the whole ensemble or on a single, sufficiently large, realization; in such a situation, a single huge system is adequate to represent the entire ensemble.

Specifically, the main contributions of this paper are:

- The observation and formalization of IGB. To our knowledge, this phenomenon was not known before our work. **IGB is relevant because** it:
  - ○ Shows that a naive intuition about DNNs (but not about linear models) is wrong; ignoring this can lead to wrong assumptions.
  - ○ Guides architecture selection, the choice of initialization, the data standardization.
  - ○ Can be exploited for new algorithmic solutions (*e.g.* to counter class imbalance).
  - ○ Displays a symmetry breaking and a violation of self-averaging.
  - ○ It has an effect on the dynamics, particularly in the initial phase, which appears qualitatively different with and without the presence of IGB.

- The demonstration of the **robustness of the phenomenon** through an extensive study across a wide range of settings; in particular:
  - ○ We demonstrate that IGB depends on the choice of activation function and provide general rules for identifying those that give rise to IGB.
  - ○ We show that IGB is generated and exacerbated by max pooling.
  - ○ We show that network depth does not cause IGB, but it intensifies it if it is already present.
  - ○ We develop a theory that analytically describes IGB in MLPs with random data (including all the settings mentioned above).
  - ○ We provide evidence of how IGB emerges in an even broader set of settings (real data, CNNs, ResNets, Vision Transformers, etc.).

**Structure of the paper**   In this paper, we formally characterize IGB, explaining its occurrence and practical implications. While we focus on binary classification in the main paper for clarity ($N_C = 2$), our results extend to multiclass classification as well, detailed in App. H.

We devote Sec. 3 to formally define IGB and to provide, through a qualitative argument, an intuition into the phenomenon. We dedicate Sec. 4 to a quantitative analysis of IGB, by studying various setups and highlighting the differences between them. In particular, we show that IGB arises because of a PSB, and we identify a control parameter that triggers the phenomenon. We find that some kinds of activations induce this symmetry breaking, and, consequently, the IGB. In Sec. 4.2 we show how max-pooling and depth exacerbate IGB. In Sec. 5 we discuss the impact of our results. Due to space reasons, and to keep the narrative simpler, we provide detailed derivations in the appendix.

## 3   EMERGENCE OF IGB: A FIRST INSIGHT INTO THE PHENOMENON

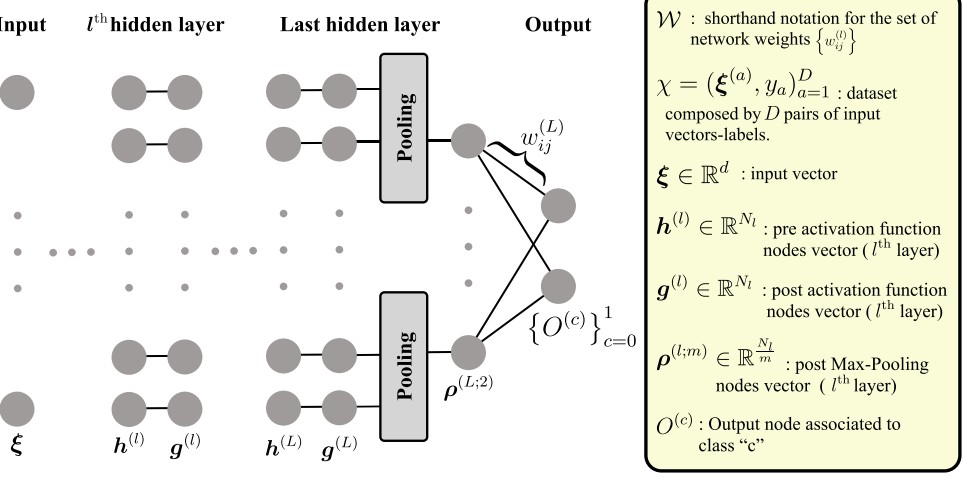

Figure 2: Scheme of a generic neural network for a binary classification problem (left) and main symbols (right); for a more complete notation see App. A.

**Setting and main notation** We consider the generic architecture shown in Fig. 2. The inputs $\{\boldsymbol{\xi}^{(a)}\}_{a=1}^{D}$ propagate through a set of $L$ hidden layers, until they reach the output layer. The last hidden layer is connected to the output layer through a dense weight matrix, with elements $w_{ij}^{(L)}$; the output layer is composed of a set of two nodes (one for each class), $\left\{O^{(c)}\right\}_{c=0,1}$. Each input is classified by selecting the class $c$ with the largest output value.

Our main notation is described in Fig. 2–right. For data and weights we respectively use:

- $\xi_b^{(a)} \sim \mathcal{N}(0,1) \ \forall a,b$: $b^{\text{th}}$ component of the $a^{\text{th}}$ dataset input.

- $w_{ij}^{(l)} \sim \mathcal{N}\left(0, \frac{\sigma_w^2}{N_{l-1}}\right) \ \forall i,j,l$: weight connecting the $j^{\text{th}}$ node of the $l^{\text{th}}$ layer to the $i^{\text{th}}$ node of the following one. We fix all the biases equal to $0$.

We focus on a simple data structure, to underline effects induced by architecture design. In particular, the choice of this data is motivated by two fundamental reasons:

- $\diamondsuit$ We place ourselves in a setting where the effect of IGB is minimal; we expect, indeed, that correlation in the data amplifies IGB.
  To gain a better understanding of this concept, we can consider a pathological extreme case of a dataset composed of clones of the same element; in this case we will clearly get the same predicted class for the whole dataset, regardless of the neural network design.

- $\diamondsuit$ In our work, we analyze how network design can induce a predictive bias; by using unstructured data, we can ensure that the sources of IGB do not stem from dataset characteristics. For the same reason, we use identically distributed data for various classes; this way, we avoid bias effects related to class differences (e.g., difficulty of the class).

Beyond the analytical results, we show that when using structured data, not only is IGB still present, but it becomes exacerbated (see App. I.1).

**Two Kinds of Averages** Since we need to average over both the dataset and the weights, for the sake of compactness we employ the shorthand notation:

$$\langle x \rangle \equiv \mathbb{E}_\chi \left( x \mid \mathcal{W} \right) \quad \text{and} \quad \bar{x} \equiv \mathbb{E}_\mathcal{W} \left( x \right)$$

to indicate the expectation over the two different sources of randomness (see App. A for more details).

**Initialization** In order to make our calculations more quantitative, we make a precise choice for the initialization of the weights, employing the *Kaiming Normal initialization* (He et al., 2015), which is common in the literature. Note however that our procedure applies to any set of independent weights drawn from a centered distribution with finite variance, with the scaling of the variance being the crucial element.

**Permutation Symmetry Breaking: the foundation of IGB** The key quantity in our analysis is the distribution (over $\mathcal{W}$) of the fraction $f_c\left(\mathcal{W}\right)$ of datapoints classified as class $c \in \{0,1\}$. We have $f_c = 1/2$ in the absence of IGB, and a different value otherwise.

Since our model's guess is assigned to the class with the largest output value, $O^{(c)}$, in the limit of infinite datapoints, the *Law of large numbers* gives

$$\lim_{D \to \infty} f_c\left(\mathcal{W}\right) = \mathbb{P}_\chi \left( O^{(c)} > O^{(1-c)} \mid \mathcal{W} \right) = \int_0^\infty \mathbb{P}_\chi \left( O^{(c)} - O^{(1-c)} = x \mid \mathcal{W} \right) \ dx. \quad (1)$$

Without loss of generality, we will often use the class 0 as a representative class, but the same discussion applies to any class. For class 0, equation 1 simplifies (again in the $D \to \infty$ limit) to

$$f_0\left(\mathcal{W}\right) = \int_0^\infty \mathbb{P}_\chi \left( \Delta_O = x \mid \mathcal{W} \right) \ dx, \quad (2)$$

where we defined, for the sake of compactness, $\Delta_O \equiv O^{(0)} - O^{(1)}$. We also define $\Delta_\mu \equiv \left\langle O^{(0)} \right\rangle - \left\langle O^{(1)} \right\rangle$ the difference between the distributions mean values. While $f_c$ is a convenient and interpretable metric related to performance, it is essential to note that our analysis begins with

a thorough a derivation of the distribution of output layer nodes, which encodes richer information about the proximity of points to decision boundaries.

equation 2 connects $f_0$ (*i.e.* the observable we are interested in) to the nodes variables $\{O^{(c)}\}$ (*i.e.* the set of variables we analyze through our investigation). The fraction $f_0$ depends on how often the output related to class 0 has a higher value of the output than that of class 1. This is essentially obtained by comparing the output distributions $\mathbb{P}_\chi\left(O^{(c)} \mid \mathcal{W}\right)$ related to each class. We illustrate this in Fig. 3. In the example of the figure, the output distribution related to class 0 is centered around higher values than that of class 1. Therefore, we will have $f_0 > f_1$, *i.e.* IGB. We will show that, for MLPs, $\mathbb{P}_\chi\left(O^{(c)} \mid \mathcal{W}\right)$, is asymptotically a Gaussian whose center, $\langle O^{(c)} \rangle$, is itself a random variable (r.v.), which is drawn from a Normal distribution $\mathbb{P}_\mathcal{W}\left(\langle O^{(c)} \rangle\right)$ that has a wide support:

$$\text{Distribution of outputs:} \quad \mathbb{P}_\chi\left(O^{(c)} \mid \mathcal{W}\right) \xrightarrow{|\mathcal{W}| \to \infty} \mathcal{N}\left(\langle O^{(c)} \rangle, \text{Var}_\chi\left(O^{(c)}\right)\right) \quad (3)$$

$$\text{Distribution of centers:} \quad \mathbb{P}_\mathcal{W}\left(\langle O^{(c)} \rangle\right) \xrightarrow{|\mathcal{W}| \to \infty} \mathcal{N}\left(0, \text{Var}_\mathcal{W}\left(\langle O^{(c)} \rangle\right)\right), \quad (4)$$

where $|\mathcal{W}|$ indicates the cardinality of the set $\mathcal{W}$.[2]   In other words, the outputs of different classes are distributed according to *p.d.f.*s each centered on a different value:

As $\langle O^{(c)} \rangle$ are r.v.s, varying across output nodes, they are not all identically distributed. This difference results in a breakdown of node-permutation symmetry. As we will explain now, this asymmetry is directly related to the emergence of IGB.

In fact, the study of IGB can be conceptually summarized as a comparison between the fluctuations of $\mathbb{P}_\mathcal{W}\left(\langle O^{(c)} \rangle\right)$, which define the distance between the Gaussians in Fig. 3, and those of $\mathbb{P}_\chi\left(O^{(c)} \mid \mathcal{W}\right)$, which define how wide each of these Gaussians is. We will consider the two extreme cases to underline our point. Starting from equation 1, we will discuss how the integral on the *r.h.s.* varies in these two scenarios:

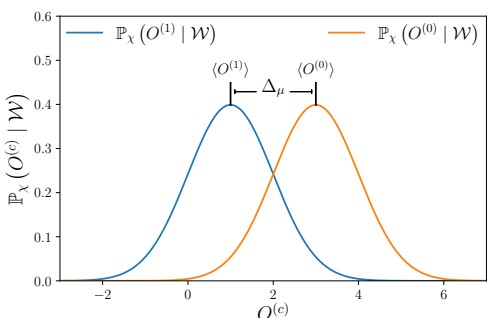

Figure 3: Illustration of the key quantities used in the analysis. The blue and orange curves represent the distributions of the two output nodes for a fixed set of network weights, $\mathcal{W}$.

- **Absence of IGB (Fig. 4 (left))**:
  If the fluctuations of $\mathbb{P}_\mathcal{W}\left(\langle O^{(c)} \rangle\right)$ are much smaller than the ones of $\mathbb{P}_\chi\left(O^{(c)} \mid \mathcal{W}\right)$, we will have two Gaussian r.v. centered almost on the same point, therefore, $\mathbb{P}_\chi\left(O^{(0)} > O^{(1)} \mid \mathcal{W}\right) \simeq 1/2$.
  In fact, the difference between two Gaussian r.v. is itself a Gaussian r.v., centered on the difference between the mean values of the original distributions, $\Delta_\mu$. If the fluctuations of $\Delta_\mu$ are much smaller than those of $\Delta_O$, we will typically have that $\mathbb{P}_\chi\left(\Delta_O \mid \mathcal{W}\right)$ is a symmetric distribution centered very close to the origin. Therefore

$$\mathbb{P}_\chi\left(\Delta_O > 0 \mid \mathcal{W}\right) \equiv \int_0^\infty \mathbb{P}_\chi\left(O^{(0)} - O^{(1)} = x \mid \mathcal{W}\right) \, dx \simeq 1/2, \quad (5)$$

  *i.e.* the probability of both classes is equal to 1/2.

- **Deep IGB (Fig. 4 (right))**:
  If, instead, the scale of $\mathbb{P}_\mathcal{W}\left(\langle O^{(c)} \rangle\right)$ fluctuations is much bigger than the one of $\mathbb{P}_\chi\left(O^{(c)} \mid \mathcal{W}\right)$ we will typically fall in the opposing scenario where the two Gaussian distributions, $\mathbb{P}_\chi\left(O^{(c)} \mid \mathcal{W}\right)$, are well separated. We can assume, without loss of generality, that $\langle O^{(0)} \rangle > \langle O^{(1)} \rangle$. In this case we will have $\mathbb{P}_\chi\left(O^{(0)} > O^{(1)} \mid \mathcal{W}\right) \simeq 1$.

This difference between the two scenarios just discussed suggests the following formal definition for IGB (we write it for a generic number $N_C$ of classes).

---

[2]Note that the writing $|\mathcal{W}| \to \infty$ is not completely unambiguous since there are several ways to take the infinite size limit; in App. C (see **Remark 1.**) we discuss how not all of them are valid.

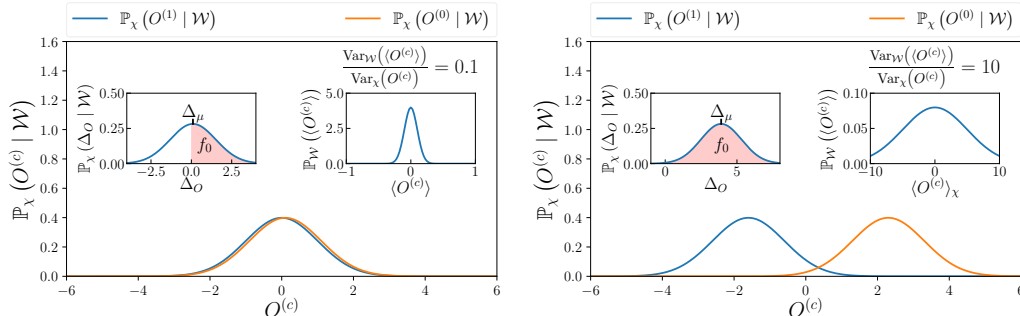

Figure 4: Comparison of two extreme scenarios: no IGB on the left, and strong IGB on the right. If the centers of the distributions, $\langle O^{(c)} \rangle$, have small fluctuations compared to the ones of the distributions $\mathbb{P}_\chi \left( O^{(c)} \mid \mathcal{W} \right)$, the two distributions almost completely overlap, resulting in a similar probability that one output node exceeds the other (left). If, instead, the centers are typically much further apart than the fluctuations scale of the distributions $\mathbb{P}_\chi \left( O^{(c)} \mid \mathcal{W} \right)$, the values drawn from one distribution exceed the other one with high probability (right). Each plot contains two inset plots. The inset plot in the upper left represents the distribution of the difference of the r.v.s shown in the main plot, $(\Delta_O)$. Note that, fixing the set $\mathcal{W}$ in a given experiment, and assuming a dataset big enough, equation 2 holds (the probability mass of the *r.h.s.* is depicted with a red area bounded by the distribution and the integration extremes). The inset plot in the upper right shows instead $\mathbb{P}_\mathcal{W} \left( \langle O^{(c)} \rangle \right)$ to give an idea of the fluctuations of $\langle O^{(c)} \rangle$ for the two cases.

---

**Definition 3.1** (IGB). We have an absence of IGB if and only if:

$$\lim_{D \to \infty} f_c \left( \mathcal{W} \right) = \frac{1}{N_C}, \ \forall c \in \{0, \ldots, N_C - 1\}. \tag{6}$$

Instead, we have IGB if we observe a disproportion between the values $\{f_c\}$, even in the limit $D \to \infty$ (so not due to finite size effects).

---

Given the asymptotically Gaussian distribution of output, increasing IGB leads to points uniformly moving away from decision boundaries.

Understanding the emergence of different node distributions due to Permutation Symmetry Breaking (PSB) is key to grasping IGB. Regardless of the activation function in the first hidden layer, PSB doesn't occur. However, from the second layer onward, the choice of activation function becomes decisive and can lead to symmetry breaking in that layer.

In the next section, we analyze a single hidden layer network, where symmetry breaking only occurs in the output layer. As we deepen the network, the second layer, formerly holding the output nodes, becomes the second hidden layer. In deep neural networks, this shift allows for symmetry breaking in the hidden layers (from the second layer onward). We'll observe that starting from a layer with PSB and propagating the signal to the next layer increases the difference between node distributions, exacerbating IGB with increasing network depth.

## 4 QUANTITATIVE ANALYSIS

We formalize and expand upon the insights introduced in Sec.3 by deriving $\mathbb{P}_\mathcal{W} \left( f_0 \right)$ for various architectural choices, from simple models to more complex ones. This helps us identify the key architectural elements that influence IGB. Detailed derivations can be found in the appendices. Furthermore, in App.I.2, in addition to our analytical findings, we demonstrate the presence of IGB across an even wider range of architectures, including ResNets and Vision Transformers

### 4.1 SINGLE HIDDEN LAYER

We start by sketching the derivation of $\mathbb{P}_\mathcal{W} \left( f_0 \right)$ for a perceptron with a single hidden layer of $N_1$ nodes (the full calculation is in App. D.1):[3]

---

[3]In this case, the infinite size limit ($|\mathcal{W}| \to \infty$) means $N_1 \to \infty$.

I. First, we show that, for every output node, $\mathbb{P}_\chi\left(O^{(c)} \mid \mathcal{W}\right)$ asymptotically converges to a Gaussian distribution, whose mean, $\left\langle O^{(c)}\right\rangle$, is itself a r.v.. Instead, its variance $\mathrm{Var}_\chi\left(O^{(c)}\right)$ converges to a deterministic value $\sigma_\infty^2$:

$$\mathbb{P}_\chi\left(O^{(c)} \mid \mathcal{W}\right) \xrightarrow{N_1 \to \infty} \mathcal{N}\left(\left\langle O^{(c)}\right\rangle, \sigma_\infty^2\right) . \tag{7}$$

Consequently for the difference between the two nodes we will have:

$$\mathbb{P}_\chi\left(\Delta_O \mid \mathcal{W}\right) \xrightarrow{N_1 \to \infty} \mathcal{N}\left(\Delta_\mu, 2\sigma_\infty^2\right) \tag{8}$$

II. From the definition of $f_0$ [equation 2], we get an implicit expression for $\Delta_\mu(f_0)$

$$f_0 = \int_0^\infty \mathcal{N}\left(y; \Delta_\mu(f_0), 2\sigma_\infty^2\right) dy . \tag{9}$$

We can then compute $\mathbb{P}_\mathcal{W}(f_0)$ as

$$\mathbb{P}_\mathcal{W}(f_0) = \mathbb{P}_\mathcal{W}(\Delta_\mu(f_0)) . \tag{10}$$

III. We show that $\mathbb{P}_\mathcal{W}\left(\left\langle O^{(c)}\right\rangle\right)$ asymptotically converges to a Gaussian distribution, with $0$ mean and a finite deterministic variance, $\hat{\sigma}_\infty^2$, *i.e.*:

$$\mathbb{P}_\mathcal{W}\left(\left\langle O^{(c)}\right\rangle\right) \xrightarrow{N_1 \to \infty} \mathcal{N}\left(0, \hat{\sigma}_\infty^2\right) . \tag{11}$$

So, from equation 11 and the definition of $\Delta_\mu$, the *r.h.s.* of equation 10 becomes

$$\mathbb{P}_\mathcal{W}(f_0) = \mathbb{P}_\mathcal{W}(\Delta_\mu(f_0)) = \mathcal{N}\left(\Delta_\mu(f_0); 0, 2\hat{\sigma}_\infty^2\right) . \tag{12}$$

In our analysis we consider three different setups to highlight the different behaviours. Fig. 5 shows $\mathbb{P}_\mathcal{W}(f_0)$ (both empirical histogram and theoretical curves) for the following cases:

- **Linear**: the distribution asymptotically converges to a delta distribution peaked on $f_0 = 1/2$.[4] The theoretical curve shown in the plot takes into account the finite dataset size effects[5] (since in real simulation $D < \infty$) as discussed in App. C.

- **ReLU**: the distribution, in this case, does not concentrate at $f_0 = 1/2$ and stays asymptotically wide (we detail this in App. C, Fig. 6), so $f_0$ will, with high probability (*w.h.p.*), be away from 1/2. The mode of the distribution remains, as for the linear case, at $f_0 = 1/2$. Yet, in this case, the fluctuations from the peak are not due to finite size effects and keep finite in the limit of infinite data points.

- **ReLU + MaxPool**: With respect to the previous case we add a pooling layer defined as

$$\rho_l^{(1;m)} = \rho^{(m)}\left(\left\{g\left(h_j^{(1)}\right)\right\}_{j \in S_l^m}\right) := \max_{j \in S_l^m}\left\{\max(0, g_j^{(1)})\right\}, \tag{13}$$

where $S_l^m$ indicate the $l$ subgroup of $m$ nodes.
As in the case of the ReLU, we have a wide distribution that does not concentrate in the limit of infinite data. The difference with the previous case is that now the distribution is peaked at the extremes (we will elaborate more on this in Sec. 4.2); in this case, it is very likely that the untrained network will classify most of the dataset as belonging to one of the two classes.

Note how, for the output layer, permutation symmetry corresponds to symmetry between classes. The latter is always preserved at the ensemble level, in fact in all three cases $\overline{f_0(\mathcal{W})} = 1/2$, where $\overline{f_0(\mathcal{W})} \equiv \int \mathbb{P}\left(\mathcal{W} = \tilde{\mathcal{W}}\right) f_0\left(\tilde{\mathcal{W}}\right) d\tilde{\mathcal{W}}$ indicate the average of $f_0(\mathcal{W})$ over the ensemble of weight initializations. However, while in the linear case, the symmetry between classes is also conserved on the single element of the ensemble,[6] in the other two cases the single realization diverges, *w.h.p.* from the mean over the ensemble. PSB thus results in a break in symmetry between classes and a consequent self-averaging breakdown for the observable $f_0$. We stress how this difference between the two estimates is not due to finite size effects of the network; our predictions (theoretical curves in Fig. 5) are, instead, asymptotically exact in this limit.

---

[4] By asymptotically, we mean in the $D \to \infty$ limit.

[5] To take finite-dataset-size effects into account, we substitute the population data distribution, with the empirical one, defined over the finite set of $D$ points.

[6] In the linear case, we have $\lim_{D \to \infty} f_0(\mathcal{W}) = \overline{f_0(\mathcal{W})} = 1/2$.

**Which activations cause IGB and which do not** Our results can be extended to a generic activation function. In particular, we can clarify (App. G) what is the fundamental attribute of the activated nodes $g_i^{(1)}$ that triggers PSB (and consequently IGB), $\forall i \in [0, N_1]$:

$$\text{if } \left\langle \rho^{(m)}\left(g_i^{(1)}\right)\right\rangle = 0 \;\Rightarrow\; \text{no IGB} \qquad (14)$$

$$\text{if } \left\langle \rho^{(m)}\left(g_i^{(1)}\right)\right\rangle \neq 0 \;\Rightarrow\; \text{IGB}$$

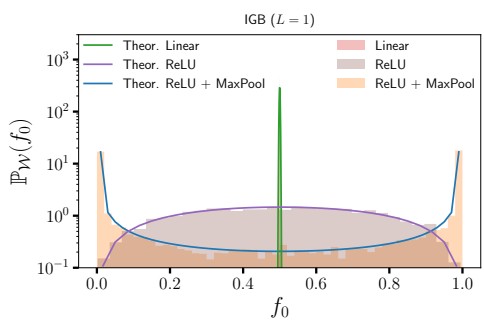

Figure 5: $\mathbb{P}_{\mathcal{W}}\left(f_0\right)$ in a single-hidden-layer perceptron, for different choices of activation functions and with/without max pooling.

Thus, the results we obtained on IGB apply to any kind of activation function: activations without IGB align with the description for linear activations, while those with IGB qualitatively resemble ReLU. We emphasize that Condition equation 14 relies on data averages $\langle \ldots \rangle$, which means IGB can be controlled by data standardization. Although our analysis centers on data around 0, other standardization methods (e.g., inputs in [0,1]) will induce IGB.

As a general guideline, antisymmetric activations exhibit no IGB when the data is centered around zero inputs. Condition equation 14 also suggests that activations can be redefined to gain or lose their IGB property. For instance, in App. G.3, we demonstrate that shifting ReLU functions appropriately can eliminate IGB.

## 4.2 Extension to Multi-layer case & Amplification of IGB

IGB arises when the distribution $\mathbb{P}_{\chi}\left(O^{(c)} \mid \mathcal{W}\right)$ is not the same for all classes (recall Fig. 4). In Sec. 4 we saw how $\mathbb{P}_{\chi}\left(O^{(c)} \mid \mathcal{W}\right)$ asymptotically tends to a Gaussian, whose variance is concentrated around a deterministic value, while the mean value is a r.v.. Therefore, to characterize the level of discrepancy in the distributions, we can compare the fluctuations of $\mathbb{P}_{\mathcal{W}}\left(\left\langle O^{(c)}\right\rangle\right)$ with the asymptotic value of the variance of $\mathbb{P}_{\chi}\left(O^{(c)} \mid \mathcal{W}\right)$. To set up this comparison, in our analysis, we will consider the variances $\text{Var}_{\chi}\left(O^{(c)}\right)$ and $\text{Var}_{\mathcal{W}}\left(\left\langle O^{(c)}\right\rangle\right)$ as a measure of the fluctuations for the two distributions. We will now show that this phenomenon can be amplified, consequently causing an amplification of IGB.

In particular, building on the picture presented in Sec. 3 we will discuss how, in certain regimes, we have a shift toward the scenario depicted in the right plot of Fig. 4. To do so it is sufficient to show that the gap between the variance of $\mathbb{P}_{\chi}\left(O^{(c)} \mid \mathcal{W}\right)$, $\text{Var}_{\chi}\left(O^{(c)}\right)$, and the one of $\mathbb{P}_{\mathcal{W}}\left(\left\langle O^{(c)}\right\rangle\right)$, $\text{Var}_{\mathcal{W}}\left(\left\langle O^{(c)}\right\rangle\right)$, increases. In particular, we want to show that the latter becomes much greater than the former. Note, in fact, that

$$\frac{\text{Var}_{\mathcal{W}}\left(\left\langle O^{(c)}\right\rangle\right)}{\text{Var}_{\chi}\left(O^{(c)}\right)} = \infty \Longrightarrow \mathbb{P}_{\mathcal{W}}\left(f_0 = x\right) = \tfrac{1}{2}\delta\left(x\right) + \tfrac{1}{2}\delta\left(x - 1\right) \qquad (15)$$

which means that in each experiment the dataset is completely classified either as belonging to the class 0 or to the class 1.

We will specifically discuss how this happens in a single hidden layer network, through the introduction of Max-Pooling (see Fig. 5). We will also show how the depth of the network itself has a similar effect to max-pooling in amplifying the discrepancy (see Fig. 7).

**IGB amplification with Max-Pooling** Max-Pooling layer changes node distribution; particularly, larger kernel size $m$ increases the mean-to-standard deviation ratio. By employing activation functions such that

$$\left\langle \rho_i^{(1;m)}\right\rangle \equiv \left\langle \rho^{(m)}\left(g\left(h_i^{(1)}\right)\right)\right\rangle = \left\langle \rho^{(1;m)}\right\rangle \neq 0, \qquad (16)$$

we get

$$\mathbb{P}_{\mathcal{W}}\left(\left\langle O^{(c)}\right\rangle\right) \xrightarrow{N_1 \to \infty} \mathcal{N}\left(0, \sigma_w^2 \left\langle \rho^{(1;m)}\right\rangle^2\right). \qquad (17)$$

(details in App. C.2). In the second equality of equation 16 we used the fact that, for every activation function employed, the variables $\{h_i^{(1)}\}$ are identically distributed.[7] Further

$$\mathbb{P}_{\chi}\left(O^{(c)} \mid \mathcal{W}\right) \xrightarrow{N_1 \to \infty} \mathcal{N}\left(\left\langle O^{(c)}\right\rangle, \text{Var}_{\chi}\left(\rho^{(1;m)}\right)\sigma_w^2\right). \qquad (18)$$

---

[7]We do not have PSB in the first hidden layer, irrespective of the employed activation function.

(details in App. C.3). In App. E) we prove that

$$\lim_{m\to\infty} \frac{\langle \rho^{(1;m)} \rangle^2}{\mathrm{Var}_\chi\big(\rho^{(1;m)}\big)} = \infty \Rightarrow \frac{\mathrm{Var}_\mathcal{W}\big(\langle O^{(c)} \rangle\big)}{\mathrm{Var}_\chi\big(O^{(c)}\big)} \xrightarrow{m\to\infty} \infty \,. \tag{19}$$

**IGB amplification with depth**    The analysis of Sec. 4.1 can be extended to deep architectures (App. F.1). Let us consider a MLP with ReLU activation function (no max-pooling). From the second hidden layer on we have PSB;[8] nodes belonging to the same layer are not anymore identically distributed. By increasing the depth of the neural network we obtain, similarly to the previous case, an amplification of distributions discrepancy that leads $\mathbb{P}_\mathcal{W}(f_0)$ to concentrate on the two extremes of the support (details in App. F.2), *i.e.*

$$\frac{\mathrm{Var}_\mathcal{W}\big(\langle O^{(c)} \rangle\big)}{\mathrm{Var}_\chi\big(O^{(c)}\big)} \xrightarrow{L\to\infty} \infty \,. \tag{20}$$

**Difference between max-pooling and depth**    Max-pooling and network depth both induce IGB amplification [equation 19 and equation 15]. However, a fundamental aspect differentiates the two cases; while the max-pooling layer can cause the emergence of IGB (as well as amplify it), depth can only amplify it. Indeed, from equation 16 we see that the condition for the emergence of IGB depends jointly effect of the activation function and the pooling layer. For a (deep) network without Pooling layers, however, we have $\rho^{(m=1)}\big(g_i^{(1)}\big) = g_i^{(1)}$;[9] therefore, the condition for the emergence of IGB [equation 16] remains constrained to the only choice of activation function.

## 5    DISCUSSION

We examined the classification bias in untrained neural networks, uncovering a phenomenon named IGB that arises from architectural choices that break permutation symmetry among hidden and output nodes within the same layer. IGB conditions are related to activation and pooling choices. Factors such as max-pooling or network depth can exacerbate IGB to the extent that all dataset elements are assigned to a single class.

This phenomenon can be summarized in a node-symmetric phase, where node-permutation symmetry is contingent on the activation functions and type of pooling being used. Interestingly, while depth does not break this symmetry, it amplifies IGB when present.

Our study of IGB also sheds light on the properties of different initialization and mean-field schemes. In fact, a key factor for IGB is that the weights in each layer $l$ are distributed with a standard deviation of order $1/\sqrt{N_l}$. This is the scaling that one has in most initializations schemes, in some mean-field (Schoenholz et al., 2016; Poole et al., 2016) and neural tangent kernel limits (Jacot et al., 2018). This kind of initialization has beneficial properties in terms of *e.g.* vanishing/exploding gradients (Arpit et al., 2019). If however we use initial conditions of order $1/N_l$ (as is done in other mean-field limits (Mei et al., 2018)), we do not have IGB. It is not clear whether IGB helps or hinders the initial learning (this could even be problem-dependent, *e.g.* with or without class imbalance, or in the presence of subclasses); nevertheless, its presence does create qualitative differences in the dynamics, making it a relevant phenomenon (see App. I.3).

In addition to the settings analyzed in our theoretical work, we demonstrated IGB's generality across a wider range of settings (App.I). While our analysis is based on random i.i.d. data, we anticipate IGB's presence in real datasets due to correlated data patches, increasing the likelihood of similar classifications. Hence, we expect stronger IGB in datasets other than random data (as shown in App.I.1 with experiments on real data).

Our experimental results in App. I.3 demonstrate that IGB significantly alters the training dynamics when using gradient-based methods. Due to its data and architecture-dependent nature, IGB remains a complex phenomenon that requires further investigation to fully understand its implications. IGB could be exploited to achieve better model training, *e.g.* with imbalanced datasets, where increasing the size of the gradients of the minority classes can significantly improve the initial phases of learning (Francazi et al., 2022). Furthermore, many works set the dynamics in the small learning rate regime (Francazi et al., 2022; Sarao Mannelli et al., 2020; Tarmoun et al., 2021); also in this context, the presence of IGB could turn out to be relevant since the dynamics are more bound to the initial state.

---

[8]With one hidden layer the nodes in the hidden layer are identically distributed and we only have PSB in the output layer.

[9]The absence of a pooling layer is formally equivalent to a pooling layer with kernel size, $m = 1$.

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

# Appendix

**Organization of the Appendix**   We first provide a summary of the notations in App. A. We then provide derivations on: the validity of Central Limit Theorems in our calculations (App. B); the output distributions at fixed weights, with an emphasis on ReLU and max-pooling (App. C); the fraction of initial guesses for each class in a 1-layer perceptron (App. D); an explicit analysis on how max-pooling intensifies IGB (App. E); the effect of depth (App. F); necessary and sufficient conditions for activations to give rise to IGB (App. G); multiple classes (App. H). Finally, App. I contains some experiments showing IGB on real datasets, and additional architectures. App. J describes the limitations and ethics of this work.

## A   NOTATION

- $\{\cdot\}_{i=0}^{M-1}$: set of $M$ elements. If some of the indices of the variables are fixed (*i.e.* equal for every element of the set), the set indices (indices that vary across different elements of the set) are reported explicitly on the right. If the index of the set elements is not explicitly reported, it means the absence of fixed indices for the set variables (*i.e.* all indices are set indices).

- $\mathbb{E}(x)$: Indicate the expectation value of the argument, $x$. If the average involves only one source of randomness this is explicitly indicated, *e.g.* $\mathbb{E}_\chi(x)$ indicates an average over the dataset distribution, while $\mathbb{E}_\mathcal{W}(x)$ an average over the distribution of network weights. For the sake of compactness, we will employ, where necessary the shorthand notation $\langle x \rangle \equiv \mathbb{E}_\chi(x)$ and $\bar{x} \equiv \mathbb{E}_\mathcal{W}(x)$.

- $\mathrm{erf}(\cdot)$: Error function.

- $f_c^{(M)}$: fraction of dataset elements classified as belonging to class $c$. The argument $M$ indicates the total number of output nodes for the variable definition, *i.e.* the number of classes considered. For binary problems we omit this argument ($f_0$) as there is only one non-trivial possibility, *i.e.* $M = 2$ .

- $f_{\tilde{c}}^{(M)}$: the set $\left\{ f_c^{(M)} \right\}_{c=0}^{M-1}$ contain the same elements of $\left\{ f_{\tilde{c}}^{(M)} \right\}_{c=0}^{M-1}$, but these are ranked by magnitude, such that $f_{\tilde{0}}^{(M)}$ is the greatest output value between the $M$ $f_{\tilde{1}}^{(M)}$ the second one and so on.

- $\boldsymbol{g}^{(l)}$: vector of the $l^{\text{th}}$ hidden layer nodes, $\left( g_0^{(l)}, \ldots, g_{N_l-1}^{(l)} \right)$, after passing through the activation function; $g_i^{(l)}$ indicate the component corresponding to node $i$.

- $\boldsymbol{h}^{(l)}$: vector of the $l^{\text{th}}$ hidden layer nodes, $\left( h_0^{(l)}, \ldots, h_{N_l-1}^{(l)} \right)$, before passing through the activation function; $h_i^{(l)}$ indicate the component corresponding to node $i$.

- $N_C \equiv N_{L+1}$: number of output nodes, *i.e.* the number of classes.

- $N_l$: number of nodes in the $l$-layer; $N_0 \equiv d$ indicates the dimension of the input data (number of input layer nodes) while $N_{L+1}$ the number of classes (number of output layer nodes).

- $\mathcal{N}\left( \mu, \sigma^2 \right)$: Given a Gaussian r.v., $X$, with mean $\mu$ and variance $\sigma^2$, $\mathcal{N}\left( \mu, \sigma^2 \right)$ indicate the distribution of $X$, *i.e.* $\mathbb{P}(X) = \mathcal{N}\left( \mu, \sigma^2 \right)$.

- $\mathcal{N}\left( x; \mu, \sigma^2 \right)$: Given a Gaussian r.v. $X$, we indicate with $\mathcal{N}\left( x; \mu, \sigma^2 \right)$ the probability density computed at $X = x$, *i.e.* $\mathcal{N}\left( x; \mu, \sigma^2 \right) \equiv \mathbb{P}(X = x) = \frac{e^{-\frac{1}{2\sigma^2}(x-\mu)^2}}{\sqrt{2\pi\sigma^2}}$

- $O_M^{(c)}$: output layer node; $c$ is the node index; the index M, instead, indicate the total number of nodes considered. For binary problems we omit the subscript index to keep the notation lighter, *i.e.* $O^{(c)}$ .

- $O_M^{(\tilde{0})}$: the set $\left\{ O_M^{(\tilde{c})} \right\}_{c=0}^{M-1}$ contain the same elements of $\left\{ O_M^{(\tilde{c})} \right\}_{\tilde{c}=0}^{M-1}$, but these are ranked by magnitude, such that $O_M^{(\tilde{0})}$ is the greatest output value between the $M$, $O_M^{(\tilde{1})}$ the second one, and so on.

- $\mathbb{P}\left( \cdot \right)$: depending on the argument this notation may indicate:
  - *p.d.f.* associated to a given r.v. $X$, $\mathbb{P}\left( X \right) = \mathbb{P}\left( X = x \right)$
  - a probability mass, *e.g.* $\mathbb{P}\left( X < x \right) = \int_{-\infty}^{x} \mathbb{P}\left( X = x' \right) dx'$

  we use the notation $\mathbb{P}\left( A \mid B \right)$ to indicate the probability of the event $A$ conditioned to event $B$.
  Finally, for variables with multiple sources of randomness, if some of them are either fixed or marginalized, we will report the remaining ones (the ones that actually induce randomness) as subscript. *E.g.* $\mathbb{P}_{\mathcal{W}}\left( f_0 \right)$ indicate the *p.d.f.* of $f_0\left( \mathcal{W} \right)$, which, fixed the dataset (or its distribution), is a function of the random set $\mathcal{W}$.

- $\mathrm{Var}\left( \cdot \right)$: Indicate the variance of the argument. Since we have r.v.s with multiple sources of noise where necessary we will specify in the subscript the source of noise used to compute the expectation. For example $\mathrm{Var}_{\chi}\left( \cdot \right) \equiv \left\langle \cdot - \left\langle \cdot \right\rangle \right\rangle^2$. For the sake of compactness, we will employ sometimes the shorthand notation $\mathrm{Var}_{\chi}\left( \cdot \right) = \sigma_{\cdot}^2$.

- $\mathcal{W}$: shorthand notation for the set of network weights, $\{w_{ij}^{(l)}\}$. We use, instead the notation $\mathcal{W}^l \subseteq \mathcal{W}$ to indicate the subset of weights relative to a specific layer, *i.e.* $\mathcal{W}^l \equiv \{w_{ij}^{(l)}\}_{\substack{j \in [0,\ldots,N_l] \\ i \in [0,\ldots,N_{l+1}]}}$. $\mathcal{W}^{<l}$, $\mathcal{W}^{>l}$, ... are defined analogously.

- $w_{ij}^{(l)}$: element $ij$ of the matrix $\boldsymbol{W}^{(l)}$, connecting two consecutive layers ($l \in [0,\ldots,L]$). Given the matrix $\boldsymbol{W}^{(l)}$ we use a 'placeholder' index, $\cdot$, to return column and row vectors from the weight matrices. In particular $w_{j\cdot}^{(l)}$ denotes row $j$ of the weight matrix $\boldsymbol{W}^{(l)}$; similarly, $w_{\cdot j}^{(l)}$ denotes column j.

- $\boldsymbol{\rho}^{(l;m)}$: vector of the $l^{\text{th}}$ hidden layer, $\left( \rho_0^{(l;m)}, \ldots, \rho_{\lceil * \rceil \frac{N_l-1}{m}}^{(l;m)} \right)$, after passing through the max-pooling layer with kernel size $m$; $\rho_i^{(l;m)}$ indicate the component corresponding to node $i$.

- $\chi = (\boldsymbol{\xi}^{(a)}, y_a)_{a=1}^{D}$: dataset composed by $D$ pairs of input vectors-labels.

- $\boldsymbol{\xi}^{(a)} \in \mathbb{R}^d$: $a^{\text{th}}$ input vector; when the index $a$ is omitted we mean a generic vector, $\boldsymbol{\xi}$, drawn from the population distribution.

- $\Theta\left( x \right)$: Heaviside step function.

- $\delta\left( x \right)$: Dirac delta function.

# B  LIMIT DISTRIBUTIONS

Here we discuss results relative to asymptotic distribution convergence.

## B.1  DISTRIBUTION CONVERGENCE

We start discussing the convergence of the distribution of r.v.s combination to an asymptotic form; we will see under which condition this convergence is guaranteed and how to characterize the asymptotic distribution. In particular we will start from the *Central Limit Theorem*, discussing after its extensions.

### B.1.1  CENTRAL LIMIT THEOREM

In the analysis we present we will make extensive use of the *Central Limit Theorem* (CLT). We present below an essential discussion of the theorem (without proof). For further details on the topic see, for example, (Gnedenko et al., 1968; Uchaikin & Zolotarev, 2011; Paul & Baschnagel, 1999). Given a sequence of *i.i.d.* r.v.s $x_1, x_2, x_3, \ldots$ drawn from a population with overall mean $\mu < \infty$

and finite variance $\sigma^2 < \infty$,[10] called $\bar{x}_n$ the sample mean of the first $n$ samples, then the limiting form of the distribution, $Z = \lim_{n\to\infty}\left(\frac{\bar{x}_n - \mu}{\sigma_{\bar{x}_n}}\right)$ with $\sigma_{\bar{x}_n} \equiv \frac{\sigma}{\sqrt{n}}$ is a standard normal distribution, *i.e.* $\mathcal{N}(0,1)$. For $1 \ll n < \infty$ this represent the leading term of an expansion; the correction to the asymptotic Gaussian profile are $o\left(\frac{1}{\sqrt{n}}\right)$ (see for example Keller & Kuske (2001)).

### B.1.2 CLT EXTENTION

The assumption of identically distributed variables can be relaxed; it is possible to formulate an extension of the CLT for combination of (not necessarily identically distributed) r.v.s. In particular, we can use, in this case, several conditions that guarantee the validity of the CLT, such as the Lyapunov condition or the Lindeberg condition.
Given a sequence of independent r.v.s $x_1, x_2, x_3, \ldots$, with $\mu_i = \mathbb{E}(x_i)$, the Lindeberg condition can be formalized as:

$$\lim_{n\to\infty} \sum_{i=1}^{n} \frac{1}{s_n^2} \int_{|x|\geq \epsilon s_n} (x - \mu_i)^2 dF_{X_i}(x) = 0\,, \tag{21}$$

where $s_n^2 = \sum_{i=1}^{n} \mathbb{E}\left((x_i - \mu_i)^2\right)$ and $F_{X_i}(x)$ is the *c.d.f.* of $x_i$.

**Theorem B.1** (Lindeberg theorem). *For a set of independent r.v.s $\{X_i\}_{i=1}^n$, if Lindeberg's condition holds for all positive $\epsilon$, then*

$$\frac{S_n - \sum_i \mu_i}{s_n} \xrightarrow{n\to\infty} \mathcal{N}(0,1), \tag{22}$$

*where $S_n = X_1 + \cdots + X_n$.*

For our analysis we will employ an alternative set of necessary and sufficient conditions to guarantees the asymptotic convergence to a normal distribution (for more details see for example Petrov (2012)).

We will first introduce, with Thm.B.2, a set of necessary and sufficient conditions that guarantee the convergence for the sum of r.v.s to a Gaussian distribution.
Then in Thm. B.3 we will provide a set of sufficient conditions (easy to verify for our interest cases). We will show the sufficiency of these conditions by proving that a set of r.v.s which satisfy them, satisfy also the conditions of Thm.B.2.

**Theorem B.2** (Distribution convergence). *Let us consider a set of independent zero-mean r.v.s $\{X_i\}_{i=1}^n$. If and only if, for every fixed $\epsilon > 0$, the following conditions are satisfied:*

*Concentration:*

$$\sum_{i=1}^{n} \mathbb{P}\left(|X_i| \geq \epsilon\right) \xrightarrow{n\to\infty} 0 \quad \forall \epsilon \in \mathbb{R}^+\,, \tag{23}$$

*Mean Normalization:*

$$\sum_i \left(\int_{|x|<\epsilon} x\mathbb{P}\left(X_i = x\right) dx\right) \xrightarrow{n\to\infty} 0\,, \tag{24}$$

*Variance Normalization:*

$$s_n^2 = \sum_i \left(\int_{|x|<\epsilon} x^2 \mathbb{P}\left(X_i = x\right) dx - \left(\int_{|x|<\epsilon} x\mathbb{P}\left(X_i = x\right) dx\right)^2\right) \xrightarrow{n\to\infty} \sigma^2\,, \tag{25}$$

*the distributions of the sum $\sum_i X_i$ will converge to $\mathcal{N}\left(0, \sigma^2\right)$.*

---

[10]This condition can be relaxed by using a generalized CLT (Darling, 1956; Lam et al., 2011).

For the proof of Thm.B.2 see Chapter 4 of Petrov (2012).

---

**Theorem B.3** (Sufficient condition for Thm. B.2). *Let us consider a set $\{X_i\}_{i=1}^n$ of independent, zero-mean,[a] r.v.s satisfying, for some constants $\tilde{\sigma}^2 \in \mathbb{R}^+$, the following conditions*

*Variance convergence:*

$$\sum_{i=1}^n \mathbb{E}\left(X_i^2 - \mathbb{E}(X_i)^2\right) \xrightarrow{n\to\infty} n\tilde{\sigma}^2, \tag{26}$$

*Fast decreasing tails:*

$$\lim_{x\to\pm\infty} \mathbb{P}(X_i = x) = \mathcal{O}\left(\frac{1}{x^4}\right), \quad \forall i. \tag{27}$$

*Let us define the new set $\{\tilde{X}_i\}_{i=1}^n$ such that*

$$\tilde{X}_i \equiv \frac{X_i}{c\sqrt{n}}, \tag{28}$$

*where $c > 0$ is a constant. We now proof that the set $\{\tilde{X}_i\}_{i=1}^n$ satisfy equation 23, equation 24, equation 25 leading to the convergence of $\tilde{S}_n = \sum_i \tilde{X}_i$ to $\mathcal{N}(0, \sigma^2)$.*

---

[a]In general if $\mathbb{E}(X_i) \neq 0$ we can define a new set of variables $\{(X_i - \mathbb{E}(X_i))\}$.

*Proof.* We will now prove that, starting from equation 26 and equation 27, the conditions of Thm. B.2 holds.
We start showing the Concentration condition [equation 23]

$$\sum_{i=1}^n \mathbb{P}\left(|\tilde{X}_i| \geq \epsilon\right) = \sum_{i=1}^n \left(\int_{-\infty}^{-\epsilon} \mathbb{P}\left(\tilde{X}_i = \tilde{x}\right) d\tilde{x} + \int_\epsilon^\infty \mathbb{P}\left(\tilde{X}_i = \tilde{x}\right) d\tilde{x}\right) =$$

$$\sum_{i=1}^n \left(\int_{-\infty}^{-c\sqrt{n}\epsilon} \mathbb{P}(X_i = x)\, dx + \int_{c\sqrt{n}\epsilon}^\infty \mathbb{P}(X_i = x)\, dx\right) \overset{n\geqslant 1}{=}$$

$$\sum_{i=1}^n \left(\int_{-\infty}^{-c\sqrt{n}\epsilon} \mathcal{O}\left(\frac{1}{x^4}\right) dx + \int_{c\sqrt{n}\epsilon}^\infty \mathcal{O}\left(\frac{1}{x^4}\right) dx\right) = \sum_{i=1}^n \left(\mathcal{O}\left(\int_{-\infty}^{-c\sqrt{n}\epsilon} \left|\frac{1}{x^4}\right| dx\right) + \mathcal{O}\left(\int_{c\sqrt{n}\epsilon}^\infty \left|\frac{1}{x^4}\right| dx\right)\right) =$$

$$\sum_{i=1}^n \mathcal{O}\left(\frac{1}{n^{\frac{3}{2}}}\right) = \mathcal{O}\left(\frac{1}{n^{\frac{1}{2}}}\right) \xrightarrow{n\to\infty} 0. \tag{29}$$

In the third line we used the *Fast decreasing tails* condition [equation 27].
Now we show the validity of mean normalization condition [equation 24]:

$$\sum_i^n \left(\int_{|\tilde{x}|<\epsilon} \tilde{x}\,\mathbb{P}\left(\tilde{X}_i = \tilde{x}\right) d\tilde{x}\right) \overset{n\geqslant 1}{=} \sum_i^n \underbrace{\mathbb{E}\left(\tilde{X}_i\right)}_{=0} - \sum_i^n \left(\mathcal{O}\left(\int_{-\infty}^{-c\sqrt{n}\epsilon} \frac{1}{c\sqrt{n}}\left|\frac{1}{x^3}\right| dx\right) + \mathcal{O}\left(\int_{c\sqrt{n}\epsilon}^\infty \frac{1}{c\sqrt{n}}\left|\frac{1}{x^3}\right| dx\right)\right) =$$

$$= \mathcal{O}\left(\frac{1}{n^{\frac{1}{2}}}\right) \xrightarrow{n\to\infty} 0. \tag{30}$$

Finally, for the Variance normalization condition [equation 25], we can now replace equation 30 into the definition of the variance. This gives

$$\sum_i \left( \int_{|\tilde{x}|<\epsilon} \tilde{x}^2 \mathbb{P}\left(\tilde{X}_i = \tilde{x}\right) d\tilde{x} - \left( \int_{|\tilde{x}|<\epsilon} \tilde{x}\mathbb{P}\left(\tilde{X}_i = \tilde{x}\right) d\tilde{x} \right)^2 \right) \overset{n\gg 1}{=} \tag{31}$$

$$\frac{\tilde{\sigma}^2}{c^2} - \sum_i^n \left( \mathcal{O}\left( \int_{-\infty}^{-c\sqrt{n}\epsilon} \frac{1}{c^2 n} \left| \frac{1}{x^2} \right| dx \right) + \mathcal{O}\left( \int_{c\sqrt{n}\epsilon}^{\infty} \frac{1}{c^2 n} \left| \frac{1}{x^2} \right| dx \right) \right) + \mathcal{O}\left( \frac{1}{n} \right) = \tag{32}$$

$$\frac{\tilde{\sigma}^2}{c^2} + \mathcal{O}\left( \frac{1}{n^{\frac{1}{2}}} \right) \xrightarrow{n\to\infty} \frac{\tilde{\sigma}^2}{c^2}. \tag{33}$$

$\square$

## B.2 CONCENTRATION OF R.V.S DISTRIBUTION

If we consider a set of $n$ r.v.s whose fluctuations (namely their variance) scale with $n$ we can have concentration phenomena for the distribution of their sum. In App. B.2.1 we consider some examples, relevant for our study, to show the point and the different scenarios induced by difference in the scaling; we will see one case where the distribution stays asymptotically stable and another where it concentrates (*i.e.* the distribution narrows around a single value).
In App. B.2.2 we will then move from the sum of r.v.s to the combination of r.v.s. In particular we will analyze the distribution $\mathbb{P}_\chi\left(h_i^{(l+1)} \mid \mathcal{W}\right)$ (fixed $\mathcal{W}$, we can indeed express $h_i^{(l+1)}$ as a combination of r.v.s). We will show how the nodes $\left\{h_i^{(l+1)}\right\}_{i=1}^{N_{l+1}}$ may not be identically distributed, for $l \geq 1$.

### B.2.1 RESCALED GAUSSIAN VARIABLES

Let us consider a set of independent r.v.s $\{X_i\}_{i=1}^n$ such that $X_i \sim \mathcal{N}\left(0, \frac{\sigma^2}{n}\right)$, $\forall i$, with $\sigma^2$ not scaling with $n$, *i.e.* $\sigma^2 = \mathcal{O}(1)$. We are interested in analyze the distribution of sum of r.v.s, in particular to understand if it shows asymptotic concentration phenomena. Let us start defining $S_x^{(n)} = \sum_{i=1}^n X_i$. To understand if the distribution of the r.v. $S_x^{(n)}$ narrows in the limit of $n \to \infty$, we can look at the standard deviation as an estimate for the order of magnitude of the fluctuations from the mean value. From the definition of $X_i$ and the additivity of the variance it follows that

$$\sigma_{S_x^{(n)}} = \sqrt{\mathbb{E}\left(S_x^{(n)} - \mathbb{E}\left(S_x^{(n)}\right)\right)^2} = \mathcal{O}(1). \tag{34}$$

In other words the distribution of the sum stys asymptotically stable, meaning that it doesn't narrow around the mean value.

Let us consider now, instead, the r.v. $S_{x^2}^{(n)} = \sum_{i=1}^n X_i^2$. Since for a Gaussian variable $\sigma_{x^2}^2 = \mathcal{O}\left(\sigma_x^4\right)$, in this case, again using the additivity of the variance, we have

$$\sigma_{S_{x^2}^{(n)}} = \sqrt{\mathbb{E}\left(S_{x^2}^{(n)} - \mathbb{E}\left(S_{x^2}^{(n)}\right)\right)^2} = \mathcal{O}\left(\frac{1}{\sqrt{n}}\right). \tag{35}$$

In this case, we have a concentration phenomenon; the fluctuations of $S_{x^2}^{(n)}$ go asymptotically to 0, *i.e.* the measure of the distribution concentrate around the mean value.

### B.2.2 COMBINATION OF RESCALED GAUSSIAN VARIABLES

To study how does the node distribution change passing from one layer to the next one, we have to analyze the distribution of $h_i^{(l+1)} = \sum_j w_{ij}^{(l)} g_j^{(l)}$, where the set $\mathcal{W}$ is fixed. We have thus a linear combination of r.v.s. It is easy to show that for this set of r.v.s the hypothesis of Thm. B.3 are satisfied (see for example B.2.3 ). We will have therefore

$$\mathbb{P}_\chi\left(h_i^{(l+1)} \mid \mathcal{W}\right) = \mathcal{N}\left( \sum_j w_{ij}^{(l)} \left\langle g_j^{(l)} \right\rangle, \sum_j \left(w_{ij}^{(l)}\right)^2 \text{Var}_\chi\left(g_j^{(l)}\right) \right). \tag{36}$$

We are now interested to understand how $\mathbb{P}_\chi\left(h_i^{(l+1)} \mid \mathcal{W}\right)$ change over the elements of the set $\{h_i^{(l+1)}\}_{i=1}^{N_{l+1}}$. To evaluate this, since we have a Gaussian distribution, is sufficient to study how much its mean and variance vary over the different nodes $\{h_i^{(l+1)}\}_{i=1}^{N_{l+1}}$, i.e. employing different set of weights $\{w_{i\cdot}^{(l)}\}_{i=1}^{N_{l+1}}$. Let us start from the mean $\left\langle h_i^{(l+1)} \right\rangle = \sum_j w_{ij}^{(l)} \left\langle g_j^{(l)} \right\rangle$ The problem, formulated from this perspective is completely reversed, meaning that now, to evaluate the fluctuations of $\left\langle h_i^{(l+1)} \right\rangle$, the set $\left\{ \left\langle g_j^{(l)} \right\rangle \right\}_{j=1}^{N_l}$ play the role of fixed constants set,[11] while the set $\left\{ w_{ij}^{(l)} \right\}_{j=1}^{N_l}$ change with the node index $i$. Again, it is easy to see that, for this combination of r.v.s, the hypothesis of Thm. B.3 are satisfied. We will then get

$$
\mathbb{P}_{w_{i\cdot}^{(l)}}\left( \left\langle h_i^{(l+1)} \right\rangle \,\middle|\, \left\{ \left\langle g_j^{(l)} \right\rangle \right\}_{j=1}^{N_l} \right) =
$$

$$
\mathcal{N}\left( \sum_j \underbrace{\overline{w_{ij}^{(l)}}}_{=0} \left\langle g_j^{(l)} \right\rangle, \sum_j \underbrace{\mathrm{Var}_\mathcal{W}\left(w_{ij}^{(l)}\right)}_{\equiv \frac{\sigma_w^2}{N_l}} \left\langle g_j^{(l)} \right\rangle^2 \right) = \mathcal{N}\left( 0, \sigma_w^2 \frac{1}{N_l} \sum_{j=1}^{N_l} \left\langle g_j^{(l)} \right\rangle^2 \right), \tag{37}
$$

where $w_{i\cdot}^{(l)} \equiv \left\{ w_{ij}^{(l)} \right\}_{j=1}^{N_l}$ If now we assume that $\left\langle g_j^{(l)} \right\rangle^2$ respect the CLT[12] (we will prove it for our case of interest) we have

$$
\frac{1}{N_l} \sum_{j=1}^{N_l} \left\langle g_j^{(l)} \right\rangle^2 \xrightarrow{N_l \to \infty} \mathbb{E}_{w_{j\cdot}^{(l-1)}}\left( \left\langle g_j^{(l)} \right\rangle^2 \right), \tag{38}
$$

leading to

$$
\mathbb{P}_{w_{i\cdot}^{(l)}}\left( \left\langle h_i^{(l+1)} \right\rangle \,\middle|\, \left\{ \left\langle g_j^{(l)} \right\rangle \right\}_{j=1}^{N_l} \right) \xrightarrow{N_l \to \infty} \mathcal{N}\left( 0, \sigma_w^2 \mathbb{E}_{w_{j\cdot}^{(l-1)}}\left( \left\langle g_j^{(l)} \right\rangle^2 \right) \right). \tag{39}
$$

In App. C we show (see **Remark 1.**) how the elements in the set $\left\{ g_j^{(l)} \right\}_{j=1}^{N_l}$ are asymptotically independent; therefore

$$
\mathbb{E}_{w_{j\cdot}^{(l-1)}}\left( \left\langle g_j^{(l)} \right\rangle^2 \right) = \overline{\left\langle g_j^{(l)} \right\rangle^2}. \tag{40}
$$

### B.2.3 A CONCRETE EXAMPLE

We will see now an example where all the elements just introduced are discussed in detail. The computation presented in this section represent an important piece in the derivation of $\mathbb{P}_\mathcal{W}\left(f_0\right)$ for deep architectures. In App. F.1 we show that

$$
\mathbb{P}_\chi\left( h_i^{(3)} \mid \mathcal{W} \right) \xrightarrow{N_2 \to \infty} \mathcal{N}\left( \left\langle h_i^{(3)} \right\rangle, \mathrm{Var}_\chi\left( h_i^{(3)} \right) \right)
$$

$$
\equiv \mathcal{N}\left( \sum_{j=1}^{N_2} w_{ij}^{(2)} \left\langle g_j^{(2)} \right\rangle, \sum_{j=1}^{N_2} \left( w_{ij}^{(2)} \right)^2 \mathrm{Var}_\chi\left( g_j^{(2)} \right) \right). \tag{41}
$$

We want to apply the analysis described in B.2.2 to understand the differences between the set of distributions $\left\{ \mathbb{P}_\chi\left( h_i^{(3)} \mid \mathcal{W} \right) \right\}_{i=1}^{N_3}$. The differences in the distributions are induced by the difference in the corresponding set of weight vectors $\{w_{i\cdot}^{(2)}\}_{i=1}^{N_3}$. The sets $\left\{ \left\langle g_j^{(2)} \right\rangle \right\}_{j=1}^{N_2}$ and $\left\{ \mathrm{Var}_\chi\left( g_j^{(2)} \right) \right\}_{j=1}^{N_2}$, instead, are the same for each node $h_i^{(3)}$. To study the differences in $\{h_i^{(3)}\}_{i=1}^{N_3}$ statistics we will

---

[11]This is the same set of elements $\forall i \in [1, \ldots, N_{l+1}]$.

[12]as we are not studying the convergence to the mean value, also the *law of large numbers* would be enough in this case

consider therefore the latter as sets of fixed random coefficients, while $\{w_{i\cdot}^{(2)}\}_{i=1}^{N_3}$ constitute the r.v.s changing from node to node. With this idea in mind let us start considering the variance expression

$$\text{Var}_\chi\left(h_i^{(3)}\right) = \sum_{j=1}^{N_2}\left(w_{ij}^{(2)}\right)^2\text{Var}_\chi\left(g_j^{(2)}\right).\tag{42}$$

Again, to characterize how much the r.v. $\text{Var}_\chi\left(h_i^{(3)}\right)$ varies, we can proceed by computing its mean and variance to have an idea of the asymptotic order of magnitude of the fluctuations. Recall that the set $\left\{\text{Var}_\chi\left(g_j^{(2)}\right)\right\}$ is fixed as we are not evaluating randomness induced by the whole initialized set of weights, but only by the differences in the set of vectors $\{w_{i\cdot}^{(2)}\}$. So we have

$$\mathbb{E}_{w_{i\cdot}^{(2)}}\left(\sum_{j=1}^{N_2}\left(w_{ij}^{(2)}\right)^2\text{Var}_\chi\left(g_j^{(2)}\right)\right) = \sum_{j=1}^{N_2}\mathbb{E}_{w_{i\cdot}^{(2)}}\left(\left(w_{ij}^{(2)}\right)^2\right)\text{Var}_\chi\left(g_j^{(2)}\right) = \sigma_w^2\frac{1}{N_2}\sum_{j=1}^{N_2}\text{Var}_\chi\left(g_j^{(2)}\right).\tag{43}$$

The sum in the last member involve a set of fixed random objects; in particular as we are using the ReLU activation function, the explicit expression for $\text{Var}_\chi\left(g_j^{(2)}\right)$ will be

$$\text{Var}_\chi\left(g_j^{(2)}\right) = \left(\int_0^\infty x^2\mathcal{N}\left(x;\mu_j,\sigma_h^2\right)dx\right) - \left(\int_0^\infty x\mathcal{N}\left(x;\mu_j,\sigma_h^2\right)dx\right)^2\tag{44}$$

where, to make the notation lighter we defined the r.v. $\mu_j \equiv \left\langle h_j^{(2)}\right\rangle = \left\langle g^{(1)}\right\rangle\sum_{i=1}^{N_1}w_{ji}^{(1)}$, $\sigma_h^2 = \text{Var}_\chi\left(g^{(1)}\right)\sigma_w^2$. We will show now that

$$\sigma_h^2 = \sup_{\mu_j\in\mathbb{R}}\text{Var}_\chi\left(g_j^{(2)}\right).\tag{45}$$

*Proof.*

$$\sigma_h^2 = \underbrace{\left(\int_{-\infty}^0(x-\mu_j)^2\mathcal{N}\left(x;\mu_j,\sigma_h^2\right)dx\right)}_{>0} + \text{Var}_\chi\left(g_j^{(2)}\right) > \text{Var}_\chi\left(g_j^{(2)}\right),\tag{46}$$

and

$$\lim_{\mu_j\to\infty}\int_{-\infty}^0(x-\mu_j)^2\mathcal{N}\left(x;\mu_j,\sigma_h^2\right)dx = 0 \implies \lim_{\mu_j\to\infty}\text{Var}_\chi\left(g_j^{(2)}\right) = \sigma_h^2.\tag{47}$$

$\square$

This means that

$$\overline{\left(\text{Var}_\chi\left(g_j^{(2)}\right)\right)^2} < \int_\mathbb{R}\sigma_h^4\mathbb{P}_\mathcal{W}\left(\mu_j=x\right)dx = \int_\mathbb{R}\sigma_h^4\mathcal{N}\left(x;0,\sigma_w^2\left\langle g^{(1)}\right\rangle^2\right)dx < \infty.\tag{48}$$

Therefore the set $\left\{\text{Var}_\chi\left(g_j^{(2)}\right)\right\}$ follow the CLT and, in particular, combining with equation 42 and equation 43 we get

$$\mathbb{E}_{w_{i\cdot}^{(2)}}\left(\text{Var}_\chi\left(h_i^{(3)}\right)\right) \xrightarrow{N_2\to\infty} \sigma_w^2\overline{\text{Var}_\chi\left(g_j^{(2)}\right)},\tag{49}$$

with

$$\overline{\text{Var}_\chi\left(g_j^{(2)}\right)} = \int_\mathbb{R}\text{Var}_\chi\left(g_j^{(2)}\right)(\mu_j)\ \mathcal{N}\left(\mu_j;0,\sigma_w^2\left\langle g^{(1)}\right\rangle^2\right)d\mu_j,\tag{50}$$

where with $\text{Var}_\chi\left(g_j^{(2)}\right)(\mu_j)$ we underlined the dependence of $\text{Var}_\chi\left(g_j^{(2)}\right)$ from the r.v. $\mu_j$. Note that in the integrand of equation 50 there is no scaling dependence with respect to $N_2$; this means that

$$\overline{\text{Var}_\chi\left(g_j^{(2)}\right)} = \mathcal{O}(1) \implies \mathbb{E}_{w_{i\cdot}^{(2)}}\left(\text{Var}_\chi\left(h_i^{(3)}\right)\right) = \mathcal{O}(1).\tag{51}$$

To evaluate the order of magnitude of the fluctuations, we recall that, given a fixed coefficient $\mathrm{Var}_\chi \left( g_j^{(2)} \right)$,

$$\mathrm{Var} \left( \left( w_{ij}^{(2)} \right)^2 \mathrm{Var}_\chi \left( g_j^{(2)} \right) \right) = \mathrm{Var}_\chi \left( g_j^{(2)} \right)^2 \mathrm{Var} \left( \left( w_{ij}^{(2)} \right)^2 \right) . \tag{52}$$

Also, for gaussian variables $\sigma_{x^2}^2 = \mathcal{O} \left( \sigma_x^4 \right)$ so $\mathrm{Var} \left( \left( w_{ij}^{(2)} \right)^2 \right) = \mathcal{O} \left( \frac{1}{N_2^2} \right)$. Then from the extensivity of the variance follows that

$$\sqrt{ \mathbb{E}_{w_{i\cdot}^{(2)}} \left( \sigma_{h_i^{(3)}}^4 \right) - \mathbb{E}_{w_{i\cdot}^{(2)}} \left( \sigma_{h_i^{(3)}}^2 \right)^2 } = \mathcal{O} \left( \frac{1}{\sqrt{N_2}} \right) . \tag{53}$$

By this we conclude that, in the $N_2 \to \infty$ limit, the distribution of the r.v. $\sum_{j=1}^{N_2} \left( w_{ij}^{(2)} \right)^2 \mathrm{Var}_\chi \left( g_j^{(2)} \right)$ narrows around the mean value $\sigma_w^2 \overline{\mathrm{Var}_\chi \left( g_j^{(2)} \right)}$.

We can proceed analogously to evaluate the mean value fluctuations; therefore we have to repeat the same analysis on the mean of $\mathbb{P}_\chi \left( h_i^{(3)} \mid \mathcal{W} \right)$, *i.e.* (from equation 41) $\sum_{j=1}^{N_2} w_{ij}^{(2)} \left\langle g_j^{(2)} \right\rangle$. In this case, we have

$$\mathbb{E}_{w_{i\cdot}^{(2)}} \left( \sum_{j=1}^{N_2} w_{ij}^{(2)} \left\langle g_j^{(2)} \right\rangle \right) = \sum_{j=1}^{N_2} \underbrace{\mathbb{E}_{w_{i\cdot}^{(2)}} \left( w_{ij}^{(2)} \right)}_{=0} \left\langle g_j^{(2)} \right\rangle = 0 . \tag{54}$$

Proceeding as in equation 52 we get

$$\sqrt{ \mathbb{E}_{w_{i\cdot}^{(2)}} \left( \mu_{h_i^{(3)}}^2 \right) - \mathbb{E}_{w_{i\cdot}^{(2)}} \left( \mu_{h_i^{(3)}} \right)^2 } = \mathcal{O}(1) . \tag{55}$$

This means that the center of the distribution $\mathbb{P}_\chi \left( h_i^{(3)} \mid \mathcal{W} \right)$ keeps fluctuating from node to node even in the limit $N_2 \to \infty$.

More specifically

$$\left( \mathbb{E}_{w_{i\cdot}^{(2)}} \left( \mu_{h_i^{(3)}}^2 \right) - \mathbb{E}_{w_{i\cdot}^{(2)}} \left( \mu_{h_i^{(3)}} \right)^2 \right) = \sigma_w^2 \frac{1}{N_2} \sum_{j=1}^{N_2} \left\langle g_j^{(2)} \right\rangle^2 . \tag{56}$$

Using the fact that $\left\langle g_j^{(2)} \right\rangle^2$ is definite positive we can proceed as in equation 46 and equation 47. It is easy then to show that the sum in the last member of equation 56 converges to a well-defined mean value. Also, it is easy to show that the set $\left\{ w_{ij}^{(2)} \left\langle g_j^{(2)} \right\rangle \right\}_{j=1}^{N_2}$ respect the conditions of Thm. B.3; this means that

$$\mathbb{P}_\mathcal{W} \left( \left\langle h_i^{(3)} \right\rangle \right) \xrightarrow{N_2 \to \infty} \mathcal{N} \left( 0, \sigma_w^2 \overline{\left\langle g_j^{(2)} \right\rangle^2} \right) , \tag{57}$$

while

$$\mathbb{P}_\mathcal{W} \left( \sigma_{h_i^{(3)}}^2 \right) \xrightarrow{N_2 \to \infty} \delta \left( \sigma_w^2 \overline{\mathrm{Var}_\chi \left( g_j^{(2)} \right)} \right) . \tag{58}$$

## C    DISTRIBUTIONS DERIVATION

In this section, we will focus on the r.v. $O^{(c)}$. In particular, the final aim here is to derive an expression for $\mathbb{P}_\mathcal{W} \left( \left\langle O^{(c)} \right\rangle \right)$ and $\mathbb{P}_\chi \left( O^{(c)} \mid \mathcal{W} \right)$, which, as seen in App. D.1, constitute the fundamental ingredients to derive $\mathbb{P}_\mathcal{W} \left( f_0 \right)$. To do so we will proceed by steps:

- In App. C.1 we will derive an expression for $\left\langle O^{(c)} \right\rangle$ for the various cases of interest.

- Once derived an expression for $\left\langle O^{(c)} \right\rangle$ we will use it to derive the asymptotic expression for the distribution $\mathbb{P}_{\mathcal{W}} \left( \left\langle O^{(c)} \right\rangle \right)$ in App. C.2.

- Finally in App. C.3 we will derive $\mathbb{P}_{\chi} \left( O^{(c)} \mid \mathcal{W} \right)$.

**Remark 1.**

$$h_i^{(1)} = \sum_j w_{ij}^{(0)} \xi_j \tag{59}$$

the independence follows directly from the independence hypothesis of the input components.
Since we use the CLT also for

$$O^{(c)} \left( \boldsymbol{\xi}; \mathcal{W} \right) = \sum_{m=1}^{N_1} w_{cm}^{(1)} g_m^{(1)}. \tag{60}$$

independence must also be fulfilled by the set $\{g_m^{(1)}\}$. As the activation function is an element-wise transformation of the set $\{h_m^{(1)}\}$, it is sufficient to have independence of the latter. The problem of the independence of $h_i^{(1)}$ and $h_j^{(1)}$ can then be reformulated like this:
Let us consider three independent random vectors $\boldsymbol{X}, \boldsymbol{Y}, \boldsymbol{Z} \in \mathbb{R}^n$ such that

$$\mathbb{P} \left( \boldsymbol{X} \right) = \mathcal{N} \left( 0, \mathbb{I} \right) \tag{61}$$

$$\mathbb{P} \left( \boldsymbol{Y} \right) = \mathcal{N} \left( 0, \frac{1}{\sqrt{n}} \mathbb{I} \right) \tag{62}$$

$$\mathbb{P} \left( \boldsymbol{Z} \right) = \mathcal{N} \left( 0, \frac{1}{\sqrt{n}} \mathbb{I} \right). \tag{63}$$

We fix two vectors $\tilde{\boldsymbol{Y}}, \tilde{\boldsymbol{Z}}$, drawn from equation 62 and equation 63 respectively. We want to prove, in the limit $n \to \infty$, the independence of the two r.v.s, $\boldsymbol{X}_{\tilde{\boldsymbol{Z}}}$ and $\boldsymbol{X}_{\tilde{\boldsymbol{Y}}}$, where we indicated with the notation $\boldsymbol{A}_{\boldsymbol{B}}$ the scalar product between $\boldsymbol{A}$ and $\boldsymbol{B}$. This may seem counterintuitive as $\boldsymbol{X}_{\tilde{\boldsymbol{Z}}}$ and $\boldsymbol{X}_{\tilde{\boldsymbol{Y}}}$ are both function of the random vector $\boldsymbol{X}$, while to have independence we have to reduce to the factorized form

$$\mathbb{P} \left( \boldsymbol{X}_{\tilde{\boldsymbol{Z}}}, \boldsymbol{X}_{\tilde{\boldsymbol{Y}}} \mid \tilde{\boldsymbol{Z}}, \tilde{\boldsymbol{Y}} \right) = \mathbb{P} \left( \boldsymbol{X}_{\tilde{\boldsymbol{Z}}} \mid \tilde{\boldsymbol{Z}}, \tilde{\boldsymbol{Y}} \right) \mathbb{P} \left( \boldsymbol{X}_{\tilde{\boldsymbol{Y}}} \mid \tilde{\boldsymbol{Z}}, \tilde{\boldsymbol{Y}} \right). \tag{64}$$

We will argue now that as $n$ grows, the projections of a random vector over two random directions are independent. From the linearity of the scalar product it follows that

$$\boldsymbol{A}_{\boldsymbol{B}+\boldsymbol{C}} = \boldsymbol{A}_{\boldsymbol{B}} + \boldsymbol{A}_{\boldsymbol{C}}. \tag{65}$$

If we could find a vector $\boldsymbol{W}$ such that, for some $a, b \in \mathbb{R}$,

$$\tilde{\boldsymbol{Y}} = a\boldsymbol{W} + b\tilde{\boldsymbol{Z}} \tag{66}$$

then the factorization in equation 64 would not be possible because of the dependence of $\tilde{\boldsymbol{Y}}$ on $\tilde{\boldsymbol{Z}}$ and vice versa.
On the other hand, for $n \to \infty$, the two vectors $\tilde{\boldsymbol{Y}}$ and $\tilde{\boldsymbol{Z}}$ are, *w.h.p.*, orthogonal; it is therefore not possible to find a decomposition as equation 66.
The orthogonality follows directly from the CLT; in particular from the defintion of equation 62 and equation 63 it follows that

$$\tilde{\boldsymbol{Y}}_{\tilde{\boldsymbol{Z}}} \xrightarrow{n \to \infty} \frac{1}{\sqrt{n}} \mathcal{N} \left( 0, 1 \right) \tag{67}$$

Therefore, *w.h.p.*, $\tilde{\boldsymbol{Y}}_{\tilde{\boldsymbol{Z}}} \xrightarrow{n \to \infty} 0$. Note that, because of the difference in the normalization, $\boldsymbol{X}_{\tilde{\boldsymbol{Z}}}$ (and $\boldsymbol{X}_{\tilde{\boldsymbol{Y}}}$) stays asymptotically finite. In fact, again from the CLT

$$\boldsymbol{X}_{\tilde{\boldsymbol{Z}}} \xrightarrow{n \to \infty} \mathcal{N} \left( 0, 1 \right). \tag{68}$$

Let us now focus on our case of interest. To apply the above argument to our case ($\boldsymbol{\xi}^{(i)} \in \mathbb{R}^d$ plays the role of $\boldsymbol{X}$), we need $d \to \infty$. In the above discussion we considered two fixed random vectors

($\tilde{\boldsymbol{Z}}$ and $\tilde{\boldsymbol{Y}}$), but for our analysis we will have, instead, $N_1 \to \infty$ fixed random vectors $\left\{ w_{j\cdot}^{(0)} \right\}_{i=1}^{N_1}$, with $w_{j\cdot}^{(0)} \in \mathbb{R}^d$. When we pass from two to an infinite number of fixed random vector we have to be more careful; the number of vectors, $N_1$, cannot grow too fast with respect to the dimension of the vector space, $d$. We know, in fact, that in a vector space of dimension $d$ it is not possible to identify a set of linear independent vectors with cardinality greater than $d$. A necessary condition for the set $\left\{ w_{j\cdot}^{(l)} \right\}_{i=1}^{N_1}$ to be mutually orthogonal is, therefore, $N_1 < d$. When we take the limits we should therefore enforce this condition, namely

$$\lim_{d, N_1 \to \infty} , \ \frac{N_1}{d} < 1. \tag{69}$$

Finally note that, if we work with a deep architecture (see App. F), once we end up with a set $\{h_m^{(1)}\}$ of linear independent r.v.s, we can repeat the same argument to iterate to the next layer. In our analysis, we will send the number of nodes of each hidden layer to infinity (to get asymptotic converge of distributions). Analogously to as we discussed above, calling $N_l$ the number of nodes in the $l^{\text{th}}$ layer, we will have (in a neural network with $L$ hidden layers) the following constraints on the asymptotic growth of the numbers of nodes:

$$\lim_{\{N_l\}_{l=1}^{L} \to \infty} , \ \frac{N_{l+1}}{N_l} < 1, \ \forall l. \tag{70}$$

## C.1 $\left\langle O^{(c)} \right\rangle$

As discussed in Sec. 1, our system is characterized by two different sources of noise. In each single experiment we set the configuration of the network with the initialization and we keep it fixed to evaluate the initial guessing distribution. The randomness coming from the set of weight is therefore fixed, while the output nodes values, $\{O^{(c)}\}$, still fluctuate varying the input $\boldsymbol{\xi}^{(i)}$. $\left\langle O^{(c)} \right\rangle$, computed on a given experiment, will be therefore a r.v. varying with the weights initialization. Here we will derive the expression for the r.v. $\left\langle O^{(c)} \right\rangle$ for different choices of the activation function. In App. C.2 we will employ the expressions derived here to get the distribution $\mathbb{P}_{\mathcal{W}} \left( \left\langle O^{(c)} \right\rangle \right)$. In the following computation we will consider a single hidden layer perceptron. These result will also constitute the starting point to formulate the case of deep architectures (see, for example, Sec. F.1). We will employ the same setting discussed at the beginning of Sec. 4.

Fixing the set $\mathcal{W}$, $h_i^{(1)}$ is a linear combination of r.v.s which respect the conditions of Thm. B.3. This means that in the limit $d \to \infty$

$$\mathbb{P}_{\chi} \left( h_i^{(1)} \mid \mathcal{W} \right) \xrightarrow{d \to \infty} \mathcal{N} \left( \sum_j w_{ij}^{(0)} \langle \xi_j \rangle, \sum_j \left( w_{ij}^{(0)} \right)^2 \sigma_{\xi_j}^2 \right) = \mathcal{N} \left( 0, \sum_j \left( w_{ij}^{(0)} \right)^2 \right) \tag{71}$$

where in the last step we used the definition of $\boldsymbol{\xi}$, for the random data used in our analysis, to substitue $\langle \xi_j \rangle = 0$, $\sigma_{\xi_j}^2 = 1 \ \forall j$. As shown in B.2.1

$$\sum_{j=1}^{d} \left( w_{ij}^{(0)} \right)^2 \xrightarrow{d \to \infty} d \ \overline{\left( w_{ij}^{(0)} \right)^2} = \sigma_w^2 \implies \mathbb{P}_{\chi} \left( h_i^{(1)} \mid \mathcal{W} \right) \xrightarrow{d \to \infty} \mathcal{N} \left( 0, \sigma_w^2 \right) \tag{72}$$

- **Linear** Let us consider the activation function $g \left( h_i^{(1)} \right) = h_i^{(1)}$. The distribution of $g_i^{(1)} \equiv g \left( h_i^{(1)} \right)$ therefore will match that of $h_i^{(1)} \equiv \sum_{j=1}^{d} w_{ij}^{(0)} \xi_j$, *i.e.*

$$\mathbb{P}_{\chi} \left( g_i^{(1)} \mid \mathcal{W} \right) \xrightarrow{d \to \infty} \mathcal{N} \left( 0, \sigma_w^2 \right) \tag{73}$$

By definition

$$O^{(c)} \left( \boldsymbol{\xi}; \mathcal{W} \right) = \sum_{m=1}^{N_1} w_{cm}^{(1)} g_m^{(1)}. \tag{74}$$

We are interested in the expression of the mean value computed over the dataset elements, for a given set of fixed weights. In other words $\mathcal{W}$ is a set of random coefficients. We have that

$$
\begin{aligned}
\left\langle O^{(c)} \right\rangle &= \int_{\mathbb{R}} \sum_{j=1}^{N_1} w_{cj}^{(1)} x_j \prod_k \mathbb{P}_\chi \left( h_k^{(1)} = x_k \mid \mathcal{W} \right) dx_k = \\
&= \sum_j w_{cj}^{(1)} \underbrace{\int_{-\infty}^{+\infty} x_j \mathcal{N} \left( x_j; 0, \sigma_w^2 \right) dx_j}_{=0} \underbrace{\prod_{k \neq j} \int_{-\infty}^{+\infty} \mathcal{N} \left( x_k; 0, \sigma_w^2 \right) dx_k}_{=1} = 0
\end{aligned}
\tag{75}
$$

We have used the fact that the set $\mathcal{W}$ is independent from $h_j^{(1)}$ at initialization and we assumed the validity of the CLT for the set of variables $\{h_j^{(1)}\}$. This assumption is formally true in simplified settings (e.g. if the components of the input data are independent r.v.s with finite variance). For real input the assumption should be evaluated case by case [13]. Also we used the independence of the elements in the set $\{h_j^{(1)}\}$ [14] to factorize the corresponding joint probability distribution (see discussion in App. C, **Remark 1.**).

Note that it was not necessary to invoke the CLT in equation 75 order to conclude that $\left\langle O^{(c)} \right\rangle = 0$; in fact

$$
\left\langle O^{(c)} \right\rangle = \left\langle \sum_{j=1}^{N_1} h_j^{(1)} w_{cj}^{(1)} \right\rangle = \sum_{j=1}^{N_1} \left\langle h_j^{(1)} \right\rangle w_{cj}^{(1)} = 0
\tag{76}
$$

On the other hand, by employing the CLT, we can go further and derive the distribution $\mathbb{P}_{\mathcal{W}} \left( \left\langle O^{(c)} \right\rangle \right)$ (see App. C.2 and Fig.6).

It is worth noting that equation 75 is formally true only in the limit of infinite size dataset; if we work with a finite size dataset, instead, we have to compute the average over the empirical distribution, $\mathbb{P}^{(E)} \left( h_k^{(1)} \mid \mathcal{W} \right) \equiv \sum_{a=1}^{D} \frac{1}{D} \delta \left( h_k^{(1)} - \hat{h}_k^{(1)} \left( \boldsymbol{\xi}^{(a)} \right) \right)$, where the set $\{\hat{h}_k^{(1)} \left( \boldsymbol{\xi}^{(a)} \right)\}_{i=1}^{D}$ contains the values mapped from each element of the dataset to the $k^{\text{th}}$ node of the first hidden layer. We will have then

$$
\begin{aligned}
\left\langle O^{(c)} \right\rangle &= \int_{\mathbb{R}} \sum_{j=1}^{N_1} h_j^{(1)} w_{cj}^{(1)} \prod_k \mathbb{P}^{(E)} \left( h_k^{(1)} \mid \mathcal{W} \right) dh_k^{(1)} = \\
&\sum_{j=1}^{N_1} w_{cj}^{(1)} \left( \frac{1}{D} \sum_{a=1}^{D} \hat{h}_k^{(1)} \left( \boldsymbol{\xi}^{(a)} \right) \right) \sim \sum_{j=1}^{N_1} w_{cj}^{(1)} \frac{1}{\sqrt{D}} Z_j = \sum_{j=1}^{N_1} w_{cj}^{(1)} \tilde{Z}_j
\end{aligned}
\tag{77}
$$

where $Z_j \sim \mathcal{N}(0, \sigma_w^2)$, while $\tilde{Z}_j \sim \mathcal{N} \left( 0, \frac{\sigma_w^2}{D} \right)$.

Note that in the last line of equation 77 we again used the CLT for the sum of r.v.s. The crucial difference with the previous derivation is that we don't use the CLT for $\mathbb{P}_\chi \left( h_k^{(1)} \mid \mathcal{W} \right)$; we substituted instead the empirical distribution, $P^{(E)} \left( h_k^{(1)} \mid \mathcal{W} \right)$. The latter being defined on the dataset elements, takes into account the finite size effect of the dataset itself (see dependence from $D$). Note that the distribution of the variables involved in the sum in the last side of equation 77 ($\tilde{Z}_j$) narrows increasing in the limit $D \to \infty$; in this limit, the finite size effects disappear and we converge to the result derived in equation 75.

- **ReLU** Now we repeat the computation of equation 75 using the activation function $g_j^{(1)} \equiv g \left( h_j^{(1)} \right) = \max\{0, h_j^{(1)}\}$ (introduced in Hahnloser et al. (2000)). $g_j^{(1)}$ will follow the

---

[13] for example the pixel of an image could be highly correlated

[14] which follows from the independence of the random vectors $\{w_{i\cdot}^{(0)}\}$

same distribution of $h_j^{(1)}$ on the positive support [15]. The probability density on the negative support of $\mathbb{P}_\chi \left( h_j^{(1)} \mid \mathcal{W} \right)$ will instead collapse on 0 [16]. Substituting to $\mathbb{P}_\chi \left( h_j^{(1)} \mid \mathcal{W} \right)$ the Gaussian distribution of equation 72 we get

$$
\mathbb{P}_\chi \left( g_j^{(1)} = x \mid \mathcal{W} \right) = \left( \int_{\mathbb{R}^-} \mathbb{P}_\chi \left( h_j^{(1)} = \tilde{x} \mid \mathcal{W} \right) d\tilde{x} \right) \delta \left( x \right) + \Theta \left( x \right) \mathbb{P}_\chi \left( h_j^{(1)} = x \mid \mathcal{W} \right) =
$$
$$
= \frac{1}{2}\delta \left( x \right) + \Theta \left( x \right) \mathcal{N} \left( x; 0, \sigma_w^2 \right).
$$
(78)

where $\delta \left( x \right)$ stands for the Dirac's delta distribution while $\Theta \left( x \right)$ stands for the Heaviside step function.

Having the distribution we can integrate over it, proceeding as in the previous case:

$$
\left\langle O^{(c)} \right\rangle = \int_{\mathbb{R}^{N_1}} \sum_{j=1}^{N_1} w_{cj}^{(1)} x_j \prod_{k=1}^{N_1} \mathbb{P}_\chi \left( g_k^{(1)} = x_k \mid \mathcal{W} \right) dx_k =
$$

$$
\sum_{j=1}^{N_1} w_{cj}^{(1)} \underbrace{\left( \prod_{k \neq j} \int_{\mathbb{R}} \left( \frac{1}{2}\delta \left( x_k \right) + \Theta \left( x_k \right) \mathcal{N} \left( x_k; 0, \sigma_w^2 \right) \right) dx_k \right)}_{=1} \int_{\mathbb{R}} x_j \left( \frac{1}{2}\delta(x_j) + \Theta(x_j)\mathcal{N} \left( x_j; 0, \sigma_w^2 \right) \right) dx_j =
$$

$$
\sum_{j=1}^{N_1} w_{cj}^{(1)} \int_0^{+\infty} x_j \mathcal{N} \left( x_j; 0, \sigma_w^2 \right) dx_j = \sum_{j=1}^{N_1} w_{cj}^{(1)} \sigma_w^2 \int_0^{+\infty} \mathcal{N} \left( x_j; 0, \sigma_w^2 \right) d\frac{x_j^2}{2\sigma_w^2} = \sum_{j=1}^{N_1} w_{cj}^{(1)} \frac{\sigma_w}{\sqrt{2\pi}}.
$$
(79)

Note that the last side of equation 79 is, in general, not equal to 0.

Note also that, in this case, we don't need to take into account the finite size effects of the dataset as did at the end of App. C.1. In fact while for a linear activation function $\lim_{D \to \infty} \left\langle O^{(c)} \right\rangle = 0$ (see equation 75) for the ReLU $\lim_{D \to \infty} \left\langle O^{(c)} \right\rangle \neq 0$ (see equation 79), so the finite size corrections of $\mathcal{O} \left( \frac{1}{D} \right)$ are negligible for $D \left\langle O^{(c)} \right\rangle \gg 1$.

In Fig. 6 the distribution $\mathbb{P}_\mathcal{W} \left( \left\langle O^{(c)} \right\rangle \right)$ is compared for the two choices of the activation function. As expected from equation 75 and equation 79 we see that, in the linear case, the distribution narrows around 0 increasing $D$ while in the ReLU case, it stays stable.

**ReLU + Max-Pool**  With the Max Pool, we group the set $\{g_k^{(1)}\}$ in subgroups and select for each subgroup the maximum value; in this way is possible to reduce the dimensionality of the layer. Adding this to the analysis in App. C.1 translate in having

$$
\rho_l^{(1;m)} = \rho \left( \left\{ g \left( h_j^{(1)} \right) \right\}_{j \in S_l^m} \right) = \max_{j \in S_l^m} \{ \max\{0, g_j^{(1)}\} \},
$$
(80)

where $S_l^m$ indicate the $l$ subgroup of $m$ nodes.

To proceed with the computation of the previous section we have to derive $\mathbb{P}_\chi \left( \rho_l^{(1;m)} \mid \mathcal{W} \right)$.

In general if we have $Y \equiv \max\{X_1, \ldots, X_m\}$, with $\{X_i\}_{i=1}^m$ i.i.d. , we have:

$$
F_Y(y) \equiv \mathbb{P} \left( Y \leq y \right) = \mathbb{P} \left( \max\{X_1, \ldots, X_m\} \leq y \right) = \prod_{i=1}^m \mathbb{P} \left( X_i \leq y \right) \equiv \prod_{i=1}^m F_X(y) = F_X(y)^m
$$
(81)

By differentiating, we find that

$$
\mathbb{P} \left( Y = y \right) = m\mathbb{P} \left( X = y \right) F_X(y)^{m-1}
$$
(82)

For our case, we substitue to $\mathbb{P} \left( X = x \right)$ the distribution used in App. C.1, i.e.

$$
\mathbb{P} \left( X = x \right) \equiv \mathbb{P}_\chi \left( g_j^{(1)} = x \mid \mathcal{W} \right) = \frac{1}{2}\delta \left( x \right) + \Theta \left( x \right) \mathcal{N} \left( x; 0, \sigma_w^2 \right).
$$
(83)

---

[15] since $g_j^{(1)} = h_j^{(1)}$ for $h_j^{(1)} > 0$

[16] since since $g_j^{(1)} = 0$ for $h_j^{(1)} < 0$

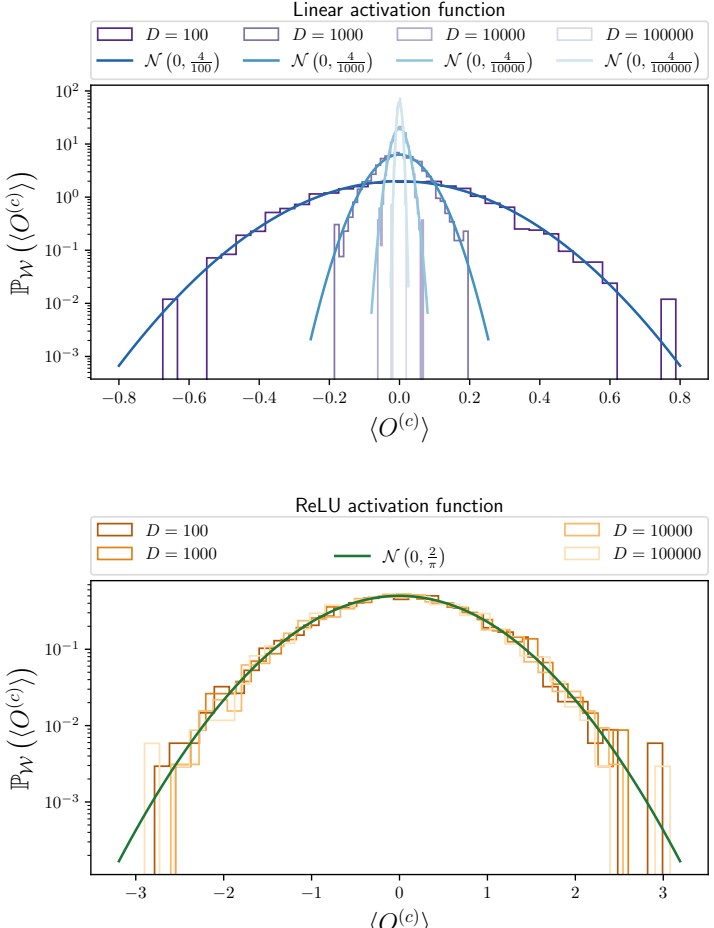

Figure 6: Comparison of the $\mathbb{P}_{\mathcal{W}}\left(\langle O^{(c)}\rangle\right)$ between the linear (upper plot) and ReLU case (bottom plot) for different choice of $D$. In both plots is also present the theoretical distribution (derived in App. C.2). Note the difference in the two cases: in the Linear case the distribution narrows increasing $D$ while in the ReLU counterparts it stays stable (as we already expected from equation 79 and equation 77). For these simulations we used the dataset GB on the model SHLP. (see App. I.4 for more details).

From Equation 83 follows

$$F_X(y) \equiv \mathbb{P}_\chi \left( g_j^{(1)} \leq y \mid \mathcal{W} \right) = \int_{-\infty}^y \frac{1}{2} \delta(x) + \Theta(x) \mathcal{N}(x; 0, \sigma_w^2) \, dx =$$

$$\int_{-\infty}^y \frac{1}{2} \delta(x) + \int_0^y \mathcal{N}\left(x; 0, \sigma_{h_j}^2\right) dx = \frac{1}{2} + \frac{1}{2} \operatorname{erf}\left(\frac{y}{\sqrt{2\sigma_{h_j}^2}}\right) \Theta(y) \tag{84}$$

Sending $m \to \infty$ using the Fisher–Tippett–Gnedenko theorem we can find an asymptotic result for the distribution $\mathbb{P}_\chi \left( \rho_l^{(1;m)} \mid \mathcal{W} \right)$. For general $m$ values, putting all the pieces together we have

$$\left\langle O^{(c)} \right\rangle = \int_{\mathbb{R}^{\frac{N_1}{m}}} \sum_l x_l w_{cl}^{(1)} \prod_k \mathbb{P}_\chi \left( \rho_l^{(1;m)} = x_l \mid \mathcal{W} \right) dx_l =$$

$$\sum_l w_{cl}^{(1)} \int_0^\infty m x_l \mathcal{N}\left(x_l; 0, \sigma_w^2\right) \left( \frac{1}{2} + \frac{1}{2} \operatorname{erf}\left(\frac{x_l}{\sqrt{2\sigma_w^2}}\right) \right)^{m-1} dx_l \tag{85}$$

which we can approximate by an asymptotic expansion, or by evaluating it numerically. The combination of Max pooling with linear activation function can be computed similarly substituting the simple Gaussian distribution for $\mathbb{P}(X = x)$.

Note that the integral in the last expression of equation 85 is independent from the set $\{w_{cl}^{(1)}\}_{l=1}^{\frac{N_1}{m}}$ [17]. To lighten the notation, we define

$$c_m(\sigma_w) \equiv \int_0^\infty m x \mathcal{N}\left(x; 0, \sigma_w^2\right) \left( \frac{1}{2} + \frac{1}{2} \operatorname{erf}\left(\frac{x}{\sqrt{2\sigma_w^2}}\right) \right)^{m-1} dx. \tag{86}$$

## C.2 $\mathbb{P}_\mathcal{W} \left( \left\langle O^{(c)} \right\rangle \right)$

We now want to derive the whole distribution $\mathbb{P}_\mathcal{W} \left( \left\langle O^{(c)} \right\rangle \right)$. Let us consider again case by case:

- **Linear**: From equation 77

$$\left\langle O^{(c)} \right\rangle \sim \sum_{j=1}^{N_1} w_{cj}^{(1)} \tilde{Z}_j \tag{87}$$

So $\left\langle O^{(c)} \right\rangle$ is a combination of $N_1$ independent r.v.s. Notice that if we are in the $\lim_{N_1 \to \infty}$ in equation 87 we have again a sum of i.i.d. r.v.s with finite variance. The distribution of the sum will then converge, for the CLT, to a Gaussian. In particular, for independent r.v.s, we have that, since $\overline{w_{cj}^{(1)}} = \mathbb{E}\left(\tilde{Z}_j\right) = 0$,

$$\operatorname{Var}\left(w_{cj}^{(1)} \tilde{Z}_j\right) = \operatorname{Var}\left(w_{cj}^{(1)}\right) \operatorname{Var}\left(\tilde{Z}_j\right) = \frac{\sigma_w^2}{N_1} \frac{\sigma_w^2}{D} \tag{88}$$

Then, from the CLT,

$$\left\langle O^{(c)} \right\rangle \sim \mathcal{N}\left(0, \frac{\sigma_w^2 \sigma_w^2}{D}\right). \tag{89}$$

- **ReLU**: for the case of ReLU the derivation is even more straightforward. In this case, is not even necessary to apply the CLT. From equation 79, indeed, we have

$$\left\langle O^{(c)} \right\rangle = \sum_{j=1}^{N_1} w_{cj}^{(1)} \frac{\sigma_w}{\sqrt{2\pi}}. \tag{90}$$

$\left\langle O^{(c)} \right\rangle$ is therefore a linear combination of Gaussian r.v.s, which is also Gaussian; more precisely

$$\left\langle O^{(c)} \right\rangle \sim \mathcal{N}\left(0, \frac{\sigma_w^4}{2\pi}\right) \tag{91}$$

---

[17] also from the index $l$ if $\sigma_l = \sigma \forall l$ as assumed in equation 85

- **ReLU + MaxPool**: This case is conceptually analogous to the previous one. In fact, from equation 85,

$$\left\langle O^{(c)} \right\rangle = \sum_{j=1}^{N_1} w_{cj}^{(1)} c_m(\sigma_w).$$

(92)

Again we have a linear combination of Gaussian r.v.s, therefore

$$\left\langle O^{(c)} \right\rangle \sim \mathcal{N}\left(0, \sigma_w^2 c_m^2(\sigma_w)\right)$$

(93)

In general, If $\left\langle \rho^{(1;m)} \right\rangle \neq 0$ we will get

$$\mathbb{P}_{\mathcal{W}}\left(\left\langle O^{(c)} \right\rangle\right) \xrightarrow{N_1 \to \infty} \mathcal{N}\left(0, \sigma_w^2 \left\langle \rho^{(1;m)} \right\rangle^2\right).$$

(94)

Note that $\rho_i^{(1;1)} = g_i^{(1)}$.

## C.3 $\mathbb{P}_\chi\left(O^{(c)} \mid \mathcal{W}\right)$

As discussed $\left\langle O^{(c)} \right\rangle$ is a r.v. that varies with the set of weights $\mathcal{W}$. In App. C.2 we derived its distribution. Here we are interested in the distribution of the output, fixed a given configuration for the set of weights, $\mathbb{P}_\chi\left(O^{(c)} \mid \mathcal{W}\right)$.

- **Linear**: from equation 74, substituting, the linear activation function $g\left(h_i^{(1)}\right) = h_i^{(1)}$, we get

$$O^{(c)} = \sum_{m=1}^{N_1} w_{cm}^{(1)} h_m^{(1)}.$$

(95)

Since the random vector $w_{c\cdot}^{(1)}$ is fixed we have a linear combination of Gaussian r.v.s. In this case is not even necessary to invoke Thm. B.2; the linear combination of Gaussian r.v.s is itself a Gaussian

$$\mathbb{P}_\chi\left(O^{(c)} \mid \mathcal{W}\right) = \mathcal{N}\left(\sum_i w_{ci}^{(1)} \left\langle h_i^{(1)} \right\rangle, \sum_i \left(w_{ci}^{(1)}\right)^2 \mathrm{Var}_\chi\left(h_i^{(1)}\right)\right) =$$
$$\mathcal{N}\left(0, \sigma_w^2 \sum_i \left(w_{ci}^{(1)}\right)^2\right)$$

(96)

Note that also in this case, in principle, we could proceed analogously to App. C.1 and take into account the correction due to the finite size of the dataset.

- **ReLU**: in this case we have:

$$O^{(c)} = \sum_{m=1}^{N_1} w_{cm}^{(1)} g_m^{(1)}$$

(97)

where $\{g_m^{(1)}\}$ are distributed according to equation 78. If $w_{c\cdot}^{(1)}$ is fixed we have a linear combination of r.v.s.

From equation 78 we have

$$\left\langle g_m^{(1)} \right\rangle = \frac{\sigma_w}{\sqrt{2\pi}}$$

(98)

$$\mathrm{Var}_\chi\left(g_m^{(1)}\right) = \int_{\mathbb{R}^+} x_m^2 \mathbb{P}_\chi\left(g_m^{(1)} = x_m \mid \mathcal{W}\right) dx_m - \mu_g^2 =$$
$$\frac{1}{2}\underbrace{\int_{\mathbb{R}} x_m^2 \mathbb{P}_\chi\left(h_m^{(1)} = x_m \mid \mathcal{W}\right) dx_m}_{\sigma_w^2} - \mu_g^2 = \frac{\sigma_w^2(\pi-1)}{2\pi}$$

(99)

where in the second step we used the symmetry of $\mathbb{P}_\chi\left(h_m^{(1)} \mid \mathcal{W}\right)$. Since the r.v.s involved in the sum of equation 97 are not Gaussian r.v.s (as in the previous case) we cannot use the same argument. However, the r.v.s involved in the sum have finite variance and exponentially fast decreasing tails; it is, therefore, trivial to prove that the conditions of Thm. B.3 are satisfied. Assuming $N_1 \gg 1$, we can then apply the Thm. B.2 for the linear combination of r.v.s. In particular, in the $\lim_{N_1 \to \infty}$, we have:

$$
\begin{aligned}
\mathbb{P}_\chi\left(O^{(c)} \mid \mathcal{W}\right) &= \mathcal{N}\left(\sum_i w_{ci}^{(1)}\left\langle g_i^{(1)}\right\rangle, \sum_i \left(w_{ci}^{(1)}\right)^2 \operatorname{Var}_\chi\left(g_i^{(1)}\right)\right) = \\
&\mathcal{N}\left(\frac{\sigma_w}{\sqrt{2\pi}}\sum_i w_{ci}^{(1)}, \frac{\sigma_w^2(\pi-1)}{2\pi}\sum_i \left(w_{ci}^{(1)}\right)^2\right).
\end{aligned}
\tag{100}
$$

- **ReLU + MaxPool**: This case is conceptually equal to the previous one in the sense that again we have a linear combination of i.i.d. r.v.s. The only difference is that we don't have an analytical expression for $\left\langle \rho_i^{(1;m)}\right\rangle$ and $\operatorname{Var}_\chi\left(\rho_i^{(1;m)}\right)$. This doesn't represent a problem; by fixing the $m$ parameter we can numerically compute the two cumulants whose expression is

$$
\begin{aligned}
\left\langle \rho^{(1;m)}\right\rangle &= \int_0^\infty mx\left(\frac{1}{2}\delta(x) + \mathcal{N}\left(x;0,\sigma_w^2\right)\right)\left(\frac{1}{2}+\frac{1}{2}\operatorname{erf}\left(\frac{x}{\sqrt{2\sigma_w^2}}\right)\right)^{m-1} dx = \\
&\int_0^\infty mx\mathcal{N}\left(x;0,\sigma_w^2\right)\left(\frac{1}{2}+\frac{1}{2}\operatorname{erf}\left(\frac{x}{\sqrt{2\sigma_w^2}}\right)\right)^{m-1} dx
\end{aligned}
\tag{101}
$$

$$
\begin{aligned}
\operatorname{Var}_\chi\left(\rho^{(1;m)}\right) &= \left(\int_0^\infty mx^2\left(\frac{1}{2}\delta(x) + \mathcal{N}\left(x;0,\sigma_w^2\right)\right)\left(\frac{1}{2}+\frac{1}{2}\operatorname{erf}\left(\frac{x}{\sqrt{2\sigma_w^2}}\right)\right)^{m-1} dx\right) \\
&- \left\langle \rho^{(1;m)}\right\rangle^2 = \left(\int_0^\infty mx^2\mathcal{N}\left(x;0,\sigma_w^2\right)\left(\frac{1}{2}+\frac{1}{2}\operatorname{erf}\left(\frac{x}{\sqrt{2\sigma_w^2}}\right)\right)^{m-1} dx\right) - \left\langle \rho^{(1;m)}\right\rangle^2.
\end{aligned}
\tag{102}
$$

So analogously to equation 100 we get

$$
\begin{aligned}
\mathbb{P}_\chi\left(O^{(c)} \mid \mathcal{W}\right) &= \mathcal{N}\left(\sum_i w_{ci}^{(1)}\left\langle \rho^{(1;m)}\right\rangle, \sum_i \left(w_{ci}^{(1)}\right)^2 \operatorname{Var}_\chi\left(\rho^{(1;m)}\right)\right) = \\
&\mathcal{N}\left(\left\langle \rho^{(1;m)}\right\rangle\sum_i w_{ci}^{(1)}, \operatorname{Var}_\chi\left(\rho^{(1;m)}\right)\sum_i \left(w_{ci}^{(1)}\right)^2\right).
\end{aligned}
\tag{103}
$$

# D  SINGLE HIDDEN LAYER PERCEPTRON

Here we will use the results derived in App. C to derive our final target $\mathbb{P}_\mathcal{W}\left(f_0\right)$. In particular we will focus here on the single hidden layer perceptron. The results will be extended to deep architectures in App. F

## D.1  DERIVATION OF $\mathbb{P}_\mathcal{W}\left(f_0\right)$

Let us consider a binary classification problem with a single hidden layer perceptron. From now on we will focus on architectures with the structure depicted in Fig.2. The nodes of the last hidden layer are therefore connected to two output nodes $\left\{O^{(c)}\right\}_{c=0}^1$ through two random vectors $\left\{w_{c\cdot}^{(1)}\right\}_{c=0}^1$. Each output node $O^{(c)}\left(\boldsymbol{\xi}; \mathcal{W}^0, w_{c\cdot}^{(1)}\right)$ will follow the distributions derived in App. C.3.

Note that, fixed a set $\mathcal{W}$, different output nodes depends on a different subset of $\mathcal{W}$ as the last matrix of weights is not shared over all the nodes. The writing $O^{(c)}\left(\boldsymbol{\xi};\mathcal{W}^0,w_c^{(1)}\right)$ underlines this aspect. This is crucial to understand why the output nodes have different distributions. For the sake of compactness, we will, anyhow, indicate a common generalized dependence from the whole set $\mathcal{W}$, *i.e.* we will write for example $O^{(c)}\left(\boldsymbol{\xi};\mathcal{W}\right)$. To derive the distribution $\mathbb{P}_{\mathcal{W}}\left(f_0\right)$ we can argue as follows: For a single experiment the set $\left\{w_c^{(1)}\right\}_{c=0}^1$ is fixed. We can then compute $\mathbb{P}_{\chi}\left(O^{(c)}\mid\mathcal{W}\right)$. Now, recalling that $f_0$ represent the fraction of the dataset classified as belonging to class 0, for the *Law of the Large Numbers*, increasing the size of the dataset, $D$, this fraction will converge to the probability

$$f_0\left(\mathcal{W}\right)=\mathbb{P}_{\chi}\left(O^{(0)}>O^{(1)}\mid\mathcal{W}\right)=\mathbb{P}_{\chi}\left(O^{(0)}-O^{(1)}>0\mid\mathcal{W}\right)=\int_0^\infty\mathbb{P}_{\chi}\left(\Delta_O=z\mid\mathcal{W}\right)\,dz$$
(104)

where $f_0\left(\mathcal{W}\right)$ indicates the fraction $f_0$ that we get for the specific fixed configuration $\mathcal{W}$. $f_0$ is a quantity clearly dependent from the set of r.v.s $\mathcal{W}$; so it is itself a r.v. which change with the set of weight of the network. Defining $\Delta_O=O^{(0)}-O^{(1)}$ we can write

$$\mathbb{P}_{\mathcal{W}}\left(f_0\right)=\int_\Omega\mathbb{P}\left(\mathcal{W}=\tilde{\mathcal{W}}\right)\delta\left(\left(\int_0^\infty\mathbb{P}_{\chi}\left(\Delta_O=z\mid\tilde{\mathcal{W}}\right)\,dz\right)-f_0\right)d\tilde{\mathcal{W}}$$
(105)

where $\Omega$ is the space of all the possibles initial configurations. equation 105 provides us an expression for the $\mathbb{P}_{\mathcal{W}}\left(f_0\right)$; on the other end the computation of the integral in the *r.h.s.* is not trivial. We now introduce a second approach to avoid the computation of this integral; we will show that this approach is asymptotically exact in the limit of $N_1\to\infty$.

In App. C.3 we derive an expression for $\mathbb{P}_{\chi}\left(O^{(c)}\mid\mathcal{W}\right)$ for the different cases; we saw that, in general, these are Gaussian distributions whose cumulants depends on $S_{\boldsymbol{w}_c}\equiv\sum_j^{N_1}w_{cj}^{(1)}$ and $S_{\boldsymbol{w}_c^2}\equiv\sum_j^{N_1}\left(w_{cj}^{(1)}\right)^2$. We can look at the standard deviations as an estimate for the order of magnitude of the fluctuations from the mean value. In App. B.2.1 we show that

$$\sqrt{\overline{S_{\boldsymbol{w}_c}-\overline{S_{\boldsymbol{w}_c}}}^2}=\mathcal{O}(1)$$
(106)

$$\sqrt{\overline{S_{\boldsymbol{w}_c^2}-\overline{S_{\boldsymbol{w}_c^2}}}^2}=\mathcal{O}\left(\frac{1}{\sqrt{N_1}}\right).$$
(107)

In other words, the distribution of $S_{\boldsymbol{w}_c}$ stays asymptotically stable, while the distribution of $S_{\boldsymbol{w}_c^2}$ asymptotically concentrates around the mean value. This difference in the scaling of the fluctuations for the two cases will be relevant for the following discussion.

To use a common notation we can say that, in general, from App. C.3, we got

$$\mathbb{P}_{\chi}\left(O^{(c)}\mid\mathcal{W}\right)=\mathcal{N}\left(\mu\left(S_{\boldsymbol{w}_c}\right),\sigma^2\left(S_{\boldsymbol{w}_c^2}\right)\right)$$
(108)

where $\mu\left(S_{\boldsymbol{w}_c}\right)$ and $\sigma\left(S_{\boldsymbol{w}_c^2}\right)$ are functions of the set of r.v.s $\{w_{cj}^{(1)}\}_{j=1}^{N_1}$ dependent from the specific setting (in our case the choice of activation function). In particular we derived in App. C.2 the distribution of $\mu\left(S_{\boldsymbol{w}_c}\right)$. From equation 107 we know, instead, that the distribution of $\sigma\left(S_{\boldsymbol{w}_c^2}\right)$ narrows around its mean values for $N_1\to\infty$; so, in this limit, $\sigma\left(S_{\boldsymbol{w}_c^2}\right)$ converge to a deterministic quantity. As the distribution of $\mu\left(S_{\boldsymbol{w}_c}\right)$ does not asymptotically narrow around a deterministic value, $\mathbb{P}_{\chi}\left(O^{(c)}\mid\mathcal{W}\right)$ will be different for different nodes, $\{O^{(c)}\}$, even in the infinite size limit. We call the emergence of this difference between distributions of nodes belonging to the same layer *Nodes Symmetry Breaking* (PSB). The discussion of Sec. 3 shows how PSB and IGB represent two sides of the same coin; more precisely IGB is a direct consequence of this symmetry breaking. In Sec. 4.2 we will discuss it more in details, showing also how the same phenomenon, in deep architectures happens not only inn the output layers, but also in the intermediate hidden layers.

Now, coming back to the problem that opened this section, we can imagine that, for a given initialization, we are fixing the configuration $\mathcal{W}$, and, in particular, the two random vectors $w_0^{(1)}$ and $w_1^{(1)}$. We will have then two corresponding distribution for $\mathbb{P}_{\chi}\left(O^{(0)}\mid\mathcal{W}\right)$ and $\mathbb{P}_{\chi}\left(O^{(1)}\mid\mathcal{W}\right)$. To compute

the probability that $O^{(0)} > O^{(1)}$ we can first pass to the distribution $\mathbb{P}_\chi(\Delta_O \mid \mathcal{W})$. Since $\Delta_O$ is a difference between two Gaussian r.v.s we have

$$\mathbb{P}_\chi(\Delta_O \mid \mathcal{W}) = \mathcal{N}\left(\mu(S_{\boldsymbol{w}_0}) - \mu(S_{\boldsymbol{w}_1}), \sigma^2\left(S_{\boldsymbol{w}_0^2}\right) + \sigma^2\left(S_{\boldsymbol{w}_1^2}\right)\right) \tag{109}$$

From equation 107 we know that the distribution of $S_{\boldsymbol{w}_c^2}$ asymptotically narrows around its mean value, so that, in the limit of $N_1 \to \infty$ we can neglect the fluctuations of the r.v. $\sigma^2\left(S_{\boldsymbol{w}_c^2}\right)$; substituting $\sigma^2\left(S_{\boldsymbol{w}_c^2}\right) = \sigma^2\left(\overline{S_{\boldsymbol{w}_c^2}}\right) \equiv \sigma_\infty^2$, we get

$$\mathbb{P}_\chi(\Delta_O \mid \mathcal{W}) \xrightarrow{N_1 \to \infty} \mathcal{N}\left(\mu(S_{\boldsymbol{w}_0}) - \mu(S_{\boldsymbol{w}_1}), 2\sigma_\infty^2\right) \tag{110}$$

This convergence is crucial to get rid of the explicit dependence from the random vectors $\{w_{c\cdot}^{(1)}\}$ and avoid the integration over all the possible configurations. In fact, since we have an explicit form for the distribution of $\mu(S_{\boldsymbol{w}_c})$ (see App. C.2), calling

$$\Delta_\mu = \mu(S_{\boldsymbol{w}_0}) - \mu(S_{\boldsymbol{w}_1}) \tag{111}$$

we can find an implicit expression for $\Delta_\mu$, i.e.

$$f_0(\mathcal{W}) = \int_0^\infty \mathcal{N}\left(y; \Delta_\mu(f_0), 2\sigma_\infty^2\right) dy. \tag{112}$$

Given the centers of the two Gaussian distributions $\mathbb{P}_\chi\left(O^{(c)} \mid \mathcal{W}\right)$, and in particular their difference, $\Delta_\mu$, equation 112 gives back the corresponding value of $f_0$. We can invert equation 112 (for example calculating it numerically) and obtain $\Delta_\mu(f_0)$ associated to a given value $f_0$. Note that $\mu(S_{\boldsymbol{w}_c}) \sim \mathbb{P}_\mathcal{W}\left(\langle O^{(c)} \rangle\right)$ is a r.v.; consequently so will $\Delta_\mu(f_0)$.
Reformulated in these terms, $\mathbb{P}_\mathcal{W}(f_0)$ is simply $\mathbb{P}_\mathcal{W}(\Delta_\mu(f_0))$. More precisely, since $\mu(S_{\boldsymbol{w}_0})$ and $\mu(S_{\boldsymbol{w}_1})$ are i.i.d. r.v.s we have

$$\begin{aligned}
\mathbb{P}_\mathcal{W}(f_0) = \mathbb{P}_\mathcal{W}(\Delta_\mu(f_0)) &= \int_{-\infty}^\infty \mathbb{P}_\mathcal{W}\left(\langle O^{(1)} \rangle = \tilde{x}\right) \mathbb{P}_\mathcal{W}\left(\langle O^{(0)} \rangle = \tilde{x} + \Delta_\mu(f_0)\right) d\tilde{x} = \\
&= \int_{-\infty}^\infty \mathbb{P}_\mathcal{W}\left(\langle O^{(c)} \rangle = \tilde{x}\right) \mathbb{P}_\mathcal{W}\left(\langle O^{(c)} \rangle = \tilde{x} + \Delta_\mu(f_0)\right) d\tilde{x},
\end{aligned} \tag{113}$$

where we integration come from the fact that the condition we are enforcing is only on the difference between the nodes mean values and therefore is invariant under a translation of both values. In the last equality, we used the fact that $\langle O^{(0)} \rangle$ and $\langle O^{(1)} \rangle$ are identically distributed. Since $\mathbb{P}_\mathcal{W}\left(\langle O^{(c)} \rangle\right) \xrightarrow{N_1 \to \infty} \mathcal{N}\left(0, \hat{\sigma}_\infty^2\right)$,[18] equation 113 is an integral of a product between two independent Gaussian distributions.[19] We can solve it and get

$$\mathbb{P}_\mathcal{W}(f_0) = \mathbb{P}_\mathcal{W}(\Delta_\mu(f_0)) = \mathcal{N}\left(\Delta_\mu(f_0); 0, 2\hat{\sigma}_\infty^2\right) \tag{114}$$

**Remark 2.** equation 114 shows a Gaussian distribution in terms of the variable $\Delta_\mu(f_0)$. Note however that $\mathbb{P}_\mathcal{W}(f_0)$, in general, will not be a Gaussian since $\Delta_\mu(f_0)$ is a non-linear function.

### D.2 MAX-POOL PEAKS: AN EXAMPLE ON HOW TO COMPUTE THE THEORETICAL PREDICTION FOR $\mathbb{P}_\mathcal{W}(f_0)$

We can use the result derived in App. D.1 to compute the theoretical prediction for $\mathbb{P}_\mathcal{W}(f_0)$. For example, we can quantify the density on the extreme peaks, empirically observed in presence of Max-Pooling. Let us say that we have a dataset of $D$ elements, and we want to compute the probability density $\mathbb{P}_\mathcal{W}(f_0 = 0)$. Sec. 4.1 gives us a recipe on how to do it. First of all we have to identify the value $\Delta^{(T)} = \Delta_\mu\left(\frac{1}{D}\right)$ inverting equation 112. $\Delta^{(T)}$ represent a threshold value; $\forall \Delta_\mu(f_0) < \Delta^{(T)}$ we will get from equation 112 a value lower than $\frac{1}{D}$ that correspond to $f_0 = 0$ because of the resolution of $f_0$ (fixed by the dataset size $D$). We can therefore compute $\mathbb{P}_\mathcal{W}(f_0 = 0)$ in two simple steps:

---

[18]The quantity $\hat{\sigma}_\infty^2$ depends on the details of the model; for more details see App. C.2.
[19]Note that the product of Gaussian distributions is itself proportional to a Gaussian distribution.

- Compute $\Delta^{(T)}$:
  first thing to do is to invert equation 112

$$f_0 = 1 - \left( \frac{1}{2} + \frac{1}{2} \operatorname{erf} \left( \frac{\Delta_\mu(f_0)}{2\sigma_\infty^2} \right) \right) \Rightarrow \Delta_\mu(f_0) = 2\sigma_\infty^2 \operatorname{erf}^{-1} (2f_0 - 1) . \tag{115}$$

- Integrate equation 114 over all $\Delta_\mu(f_0) < \Delta^{(T)}$, i.e.

$$\mathbb{P}_\mathcal{W} (f_0 = 0) = \int_{-\infty}^{\Delta^{(T)}} \mathcal{N} \left( \Delta_\mu(f_0); 0, 2\hat{\sigma}_\infty^2 \right) d\Delta_\mu(f_0) = \frac{1}{2} + \frac{1}{2} \operatorname{erf} \left( \frac{\Delta^{(T)}}{2\hat{\sigma}_\infty} \right) . \tag{116}$$

In general, if we want the probability mass between two values $f_0^{(min)}$ and $f_0^{(max)}$, we can compute it as

$$\int_{f_0^{(min)}}^{f_0^{(max)}} \mathbb{P}_\mathcal{W} (f_0 = x) \, dx = \int_{\Delta_\mu\left(f_0^{(min)}\right)}^{\Delta_\mu\left(f_0^{(max)}\right)} \mathcal{N} \left( \Delta_\mu(f_0); 0, 2\hat{\sigma}_\infty^2 \right) d\Delta_\mu(f_0) =$$
$$\frac{1}{2} \operatorname{erf} \left( \frac{\Delta_\mu \left( f_0^{(max)} \right)}{2\hat{\sigma}_\infty} \right) - \frac{1}{2} \operatorname{erf} \left( \frac{\Delta_\mu \left( f_0^{(min)} \right)}{2\hat{\sigma}_\infty} \right) . \tag{117}$$

## E   EFFECT OF MAX-POOLING

In this section we will discuss how the impact of Max-Pooling. Here we will show that increasing the value of $m$, the distribution $\mathbb{P}_\chi \left( \rho^{(1;m)} \mid \mathcal{W} \right)$ concentrates around an increasing value. For the sake of clarity, we will discuss here the asymptotic convergence, *i.e.* $m \to \infty$. Let us consider the set $\{g_i^{(1)}\}_{i=1}^m$ such that

$$\mathbb{P}_\chi \left( g_i^{(1)} = x \mid \mathcal{W} \right) = \frac{1}{2}\delta(x) + \Theta(x) \mathcal{N} \left( x; 0, \sigma_w^2 \right) , \tag{118}$$

for a fixed set $\mathcal{W}$. Let us now define the set of r.v.s $\{\omega_i\}_{i=1}^m$, calling $n = \log(m)$,

$$\omega_i \equiv \frac{g_i^{(1)}}{\sqrt{\log(m)}} \equiv \frac{g_i^{(1)}}{\sqrt{n}} . \tag{119}$$

Using the relation

$$\mathbb{P}_\chi (\omega_i = x \mid \mathcal{W}) \, d\omega = \mathbb{P}_\chi \left( g^{(1)} \mid \mathcal{W} \right) dg^{(1)} , \tag{120}$$

we can write

$$\mathbb{P}_\chi (\omega_i = x \mid \mathcal{W}) = \frac{1}{2}\delta(x) + \Theta(x) \mathcal{N} \left( x; 0, \frac{\sigma_w^2}{n} \right) . \tag{121}$$

We can now consider the number of elements in the set $\{\omega_i\}$ whose value fall in the interval $[\tilde{\omega}, (\tilde{\omega} + d\tilde{\omega})]$, $\tilde{\omega} > 0$; we call this number $\#(\tilde{\omega})$. In particular if we consider its expectation value, we'll have

$$\mathbb{E}\left(\#(\tilde{\omega})\right) = m\mathbb{E}\left(\mathbb{I}(\omega \in [\tilde{\omega}, (\tilde{\omega} + d\tilde{\omega})])\right) \sim m \sqrt{\frac{n}{2\pi\sigma_w^2}} e^{-\frac{\omega^2 n}{2\sigma_w^2}} d\tilde{\omega} = \sqrt{\frac{n}{2\pi\sigma_w^2}} e^{-n\left(\frac{\omega^2}{2\sigma_w^2} - 1\right)} d\tilde{\omega}. \tag{122}$$

Also, we see that, being the probability to get a value from $\mathbb{P}_\chi (\omega \mid \mathcal{W})$ to be between $\tilde{\omega}$ and $(\tilde{\omega} + d\tilde{\omega})$ small, this follows a Poisson law, so that the variance is equal to the mean. If we define

$$s(\omega) \equiv \frac{\omega^2}{2\sigma_w^2} - 1 , \tag{123}$$

we have that

- If $s(\omega) < 0$ the average number of draws with value $\omega$ is exponentially close to 0. From equation 123 we see that this happens (as we are interested in the positive interval) when $\omega > \sqrt{2\sigma_w^2}$. In this case, using the Markov inequality [20], we can conclude that, *w.h.p.* the number of draws (and not only its expectation value) is actually 0. In fact, from Markov inequality,

$$\mathbb{P}\left(\#(\tilde{\omega}) \geq 1\right) \leq \mathbb{E}\left(\#(\tilde{\omega})\right). \tag{124}$$

Therefore, *w.h.p.* as $n \to \infty$, all draws have values lower than $\sqrt{2\sigma_w^2}$.

- If $s(\omega) > 0$, the expectation value of the number of draws is different from 0. We can use again the Markov inequality to show that, also in this case, the actual number of draws concentrate around the non null expectation value, as $n \to \infty$. In fact

$$\mathbb{P}\left(\left|\frac{\#(\tilde{\omega})}{\mathbb{E}\left(\#(\tilde{\omega})\right)} - 1\right| \geq k\right) = \mathbb{P}\left(\left(\frac{\#(\tilde{\omega})}{\mathbb{E}\left(\#(\tilde{\omega})\right)} - 1\right)^2 \geq k^2\right) \leq \frac{\mathbb{E}\left(\left(\#(\tilde{\omega}) - \mathbb{E}\left(\#(\tilde{\omega})\right)\right)^2\right)}{k^2 \left(\mathbb{E}\left(\#(\tilde{\omega})\right)\right)^2}$$

$$\leq \frac{\text{Var}\left(\#(\tilde{\omega})\right)}{k^2 \left(\mathbb{E}\left(\#(\tilde{\omega})\right)\right)^2} \simeq \frac{\cancel{\mathbb{E}\left(\#(\tilde{\omega})\right)}}{k^2 \left(\mathbb{E}\left(\#(\tilde{\omega})\right)\right)^{\cancel{2}}} \propto \frac{e^{-ns(\tilde{\omega})}}{k^2},$$

$$\tag{125}$$

where we have used the fact that, being small the probability of $\omega_i$ taken randomly from $\mathbb{P}_\chi\left(\omega \mid \mathcal{W}\right)$ to be between $\tilde{\omega}$ and $\tilde{\omega} + d\tilde{\omega}$, $\#(\tilde{\omega})$ follows a Poisson law; therefore the variance is equal to the mean. As $n$ grows, the probability of deviation goes exponentially to zero when $s(\tilde{\omega}) > 0$, so that *w.h.p.*, $\frac{\#(\tilde{\omega})}{\mathbb{E}(\#(\tilde{\omega}))}$ is arbitrary close to 1.

We shown, therefore that, when $n$ grows the maximum among the set $\{g_i^{(1)}\}_{i=1}^m$ converge to the deterministic value $\sqrt{2\sigma_w^2 \log(m)}$. In other words the maximum value asymptotically concentrates around an increasing value. More fine analysis are possible to characterize the extreme value statistics. For more information about the topic, see, for example, Schehr & Majumdar (2014) and references within it.

# F  DEEP ARCHITECTURES

In Sec. 4.1 we discussed how to derive $\mathbb{P}_\mathcal{W}\left(f_0\right)$ for a neural network with a single hidden layer. In the following section we will discuss the extention of the computation to a network with an arbitrary number of hidden layers.

## F.1  MULTI-LAYER PERCEPTRON

In this Section we will extend the result discussed in Sec. 4.1. Our analysis will clarify the decisive impact of the network deepness for the IGB phenomenon. There is in fact a foundamental difference between the two cases determining a possible exacerbation of the phenomenon, as we will discuss here.

The strategy that we will follow is similar to the one presented for the single hidden layer counterpart (see Sec. 4.1). In other word we will propagate across the network layers keeping track of the changes in the distribtions $\mathbb{P}_\chi\left(h_i^{(l+1)} \mid \mathcal{W}\right)$ and $\mathbb{P}_\mathcal{W}\left(\overline{h_i^{(l+1)}}\right)$. Using CLT considerably simplifies this propagation process because to keep track of changes in the distribution it is sufficient to see how certain quantities vary [21]. Compared to the analysis in Sec. 4.1 however, there is an important complicating element. While the elements in the set $\{h_i^{(1)}\}$ follow the same distribution (regardless of the choice of activation function), from the second layer onward the activation function can induce a break in symmetry among the layer nodes, in the sense that they will follow, fixed a given configuration for the network weights, different distributions. This symmetry breaking, as we shall see, can cause an accentuation of the IGB effect. To show this point we will consider in our analysis

---

[20]Markov inequality states that if $X$ is a non-negative random variable, then $\mathbb{P}\left(X \geq k\right) \leq \mathbb{E}\left(X\right)/k$

[21]a Gaussian distribution is completely determined from the first two cumulants

the ReLU activation function [22]. Starting from the same setting described in Sec. 4.1 we will derive a set of iterative equation to propagate across multiple layers.

- **layer-1**
  as showed in Sec. 4.1 and in App. C.1, in the limit $d \to \infty$,

$$
h_i^{(1)} \equiv \sum_j w_{ij}^{(0)} \xi_j \xRightarrow{\text{CLT}} \mathbb{P}_\chi \left( h_i^{(1)} \mid \mathcal{W} \right) \xrightarrow{d \to \infty} \mathcal{N}(0, \sigma_w^2)
$$

$$
\xrightarrow{\text{ReLU}} \mathbb{P}_\chi \left( g_i^{(1)} \mid \mathcal{W} \right) \xrightarrow{d \to \infty} \Theta(x) \mathcal{N} \left( x; 0, \sigma_w^2 \right) + \frac{1}{2} \delta(x) .
$$
(126)

All the nodes $\{h_i^{(1)}\}$ follow the same distribution and so do the nodes $\{g_i^{(1)}\}$. In particular, this means that

$$
\left\langle g_i^{(1)} \right\rangle = \left\langle g^{(1)} \right\rangle , \ \forall i .
$$
(127)

- **layer-2**
  We start again from a combination of r.v.s

$$
h_i^{(2)} \equiv \sum_j w_{ij}^{(1)} g_j^{(1)} .
$$
(128)

It is easy to proof that the generic r.v. $\left( w_{ij}^{(1)} g_j^{(1)} \right)$ involved in the sum satisfies the conditions of equation 26 and equation 27. So, in the limit $N_1 \to \infty$, again we have a convergence to a normal distribution. In particular

$$
\mathbb{P}_\chi \left( h_i^{(2)} \mid \mathcal{W} \right) \xrightarrow{N_1 \to \infty} \mathcal{N} \left( \left\langle g^{(1)} \right\rangle S_{\boldsymbol{w}_i}, \text{Var}_\chi \left( g^{(1)} \right) S_{\boldsymbol{w}_i^2} \right) =
$$
$$
\mathcal{N} \left( \left\langle g^{(1)} \right\rangle S_{\boldsymbol{w}_i}, \text{Var}_\chi \left( g^{(1)} \right) \overline{S_{\boldsymbol{w}_i^2}} \right) ,
$$
(129)

where

$$
\text{Var}_\chi \left( g^{(1)} \right) = \left\langle \left( g^{(1)} - \left\langle g^{(1)} \right\rangle \right)^2 \right\rangle .
$$
(130)

In the last step of equation 129, as done for $\mathbb{P}_\chi \left( O^{(c)} \mid \mathcal{W} \right)$ in Sec. 4.1, we used the concentration result derived in App. B.2.1. In particular that the distribution of the r.v. $S_{\boldsymbol{w}_i} \equiv \sum_{j=1}^{N_1} w_{ij}^{(1)}$ stays stable in the limit $N_1 \to \infty$ [23], while the distribution of $S_{\boldsymbol{w}_i^2} \equiv \sum_{j=1}^{N_1} \left( w_{ij}^{(1)} \right)^2$ asymptotically narrow around the mean value. Note that, as the distribution of $S_{\boldsymbol{w}_i}$ does not asymptotically concentrate, and since $\left\langle g^{(1)} \right\rangle \neq 0$, each node in the set $\{h_i^{(2)}\}$ will follows a different distribution. In particular we will have a set of normally distributed r.v.s, centred in different random points, $\{\left\langle g^{(1)} \right\rangle S_{\boldsymbol{w}_i}\}$. After passing through the ReLU activation function we will have

$$
\mathbb{P}_\chi \left( g_i^{(2)} \mid \mathcal{W} \right) \xrightarrow{N_1 \to \infty} \Theta(x) \mathcal{N} \left( \left\langle g^{(1)} \right\rangle S_{\boldsymbol{w}_i}, \text{Var}_\chi \left( g^{(1)} \right) \overline{S_{\boldsymbol{w}_i^2}} \right) +
$$
$$
+ \left( \frac{1}{2} + \frac{1}{2} \text{erf} \left( \frac{\left\langle g^{(1)} \right\rangle S_{\boldsymbol{w}_i}}{\sqrt{2 \text{Var}_\chi \left( g^{(1)} \right) \overline{S_{\boldsymbol{w}_i^2}}}} \right) \right) \delta(x) .
$$
(131)

- **layer-3**
  We can repeat the approach of the previous layer for the new set of variables

$$
h_i^{(3)} \equiv \sum_j w_{ij}^{(2)} g_j^{(2)} ,
$$
(132)

---

[22]we do not include the MaxPooling to underlining the amplification of the IGB; in the presence of MaxPooling, increasing the kernel size we observe deep IGB even with a single hidden layer architectures. Note that absence of MaxPooling is equivalent of a MaxPooling with minimum kernel size

[23]fluctuations and mean stay both $\mathcal{O}(1)$

getting

$$P\left(h_i^{(3)} \mid \mathcal{W}\right) \xrightarrow{N_2 \to \infty} \mathcal{N}\left(\sum_{j=1}^{N_2} w_{ij}^{(2)} \left\langle g_j^{(2)} \right\rangle, \sum_{j=1}^{N_2} \left(w_{ij}^{(2)}\right)^2 \text{Var}_\chi\left(g_j^{(2)}\right)\right). \quad (133)$$

Note that in this case we cannot take out from the sums $\left\langle g_j^{(2)} \right\rangle$ and $\text{Var}_\chi\left(g_j^{(2)}\right)$ since the nodes $\{g_j^{(2)}\}$ are not identically distributed. In App. B.2.3 we analyze the differences in $\mathbb{P}_\chi\left(h_i^{(3)} \mid \mathcal{W}\right)$ between the different nodes $\{h_i^{(3)}\}$. In particular, we show that

$$\mathbb{P}_\chi\left(h_i^{(3)} \mid \mathcal{W}\right) \xrightarrow{N_2 \to \infty} \mathcal{N}\left(\left\langle h_i^{(3)} \right\rangle, \text{Var}_\chi\left(h_i^{(3)}\right)\right), \quad (134)$$

$$\mathbb{P}_\mathcal{W}\left(\left\langle h_i^{(3)} \right\rangle\right) \xrightarrow{N_2 \to \infty} \mathcal{N}\left(0, \sigma_w^2 \overline{\left\langle g_j^{(2)} \right\rangle^2}\right), \quad (135)$$

$$\mathbb{P}_\mathcal{W}\left(\text{Var}_\chi\left(h_i^{(3)}\right)\right) \xrightarrow{N_2 \to \infty} \delta\left(\sigma_w^2 \overline{\text{Var}_\chi\left(g_j^{(2)}\right)}\right). \quad (136)$$

- **layer-l**
  The steps described for the propagation of the *layer-3* provide us an iteration scheme that we can follow for a generic layer $l \geq 3$. In fact, knowing the statistics of the previous layers, we can compute

$$\overline{\text{Var}_\chi\left(g_j^{(l-1)}\right)} = \int_\mathbb{R} \text{Var}_\chi\left(g_j^{(l-1)}\right)(\mu_j) \; \mathcal{N}\left(\mu_j; 0, \sigma_w^2 \overline{\left\langle g_j^{(l-2)} \right\rangle^2}\right) d\mu_j, \quad (137)$$

$$\overline{\left\langle g_j^{(l-1)} \right\rangle^2} = \int_\mathbb{R} \left\langle g_j^{(l-1)} \right\rangle^2 (\mu_j) \; \mathcal{N}\left(\mu_j; 0, \sigma_w^2 \overline{\left\langle g_j^{(l-2)} \right\rangle^2}\right) d\mu_j. \quad (138)$$

Again, in the above expressions we defined $\text{Var}_\chi\left(g_j^{(l-1)}\right)$ as

$$\text{Var}_\chi\left(g_j^{(l-1)}\right) = \left(\int_0^\infty x^2 \mathcal{N}\left(x; \mu_j, \sigma_h^2\right) dx\right) - \left(\int_0^\infty x \mathcal{N}\left(x; \mu_j, \sigma_h^2\right) dx\right)^2, \quad (139)$$

where, to make the notation lighter, we defined the r.v. $\mu_j \equiv \left\langle h_j^{(l-1)} \right\rangle = \sum_{j=1}^{N_{l-2}} w_{ij}^{(l-2)} \left\langle g_j^{(l-2)} \right\rangle, \sigma_h^2 = \sum_{j=1}^{N_{l-2}} \left(w_{ij}^{(l-2)}\right)^2 \sigma_{g_j^{(l-2)}}^2$. Note that now, the difference in the distributions of the nodes $\{g_j^{(l-2)}\}$ induce a difference in the values $\left\{\text{Var}_\chi\left(g_j^{(l-1)}\right)\right\}$. From equation 137 and equation 138 we can compute then

$$\mathbb{P}_\chi\left(h_i^{(l)} \mid \mathcal{W}\right) \xrightarrow{N_{l-1} \to \infty} \mathcal{N}\left(\left\langle h_i^{(l)} \right\rangle, \sigma_w^2 \overline{\text{Var}_\chi\left(g_j^{(l-1)}\right)}\right), \quad (140)$$

$$\mathbb{P}_\mathcal{W}\left(\left\langle h_i^{(l)} \right\rangle\right) \xrightarrow{N_{l-1} \to \infty} \mathcal{N}\left(0, \sigma_w^2 \overline{\left\langle g_j^{(l-1)} \right\rangle^2}\right). \quad (141)$$

Finally we can use equation 141 to iterate the set of equation 137 and equation 138 for the layer $l$, *i.e.*

$$\overline{\text{Var}_\chi\left(g_j^{(l)}\right)} = \int_\mathbb{R} \text{Var}_\chi\left(g_j^{(l)}\right)(\mu_j) \; \mathbb{P}_\mathcal{W}\left(\mu_{h_j^{(l)}} = \mu_j\right) d\mu_j, \quad (142)$$

$$\overline{\left\langle g_j^{(l)} \right\rangle^2} = \int_\mathbb{R} \left\langle g_j^{(l)} \right\rangle^2 (\mu_j) \; \mathbb{P}_\mathcal{W}\left(\mu_{h_j^{(l)}} = \mu_j\right) d\mu_j. \quad (143)$$

- **layer-(L+1)**
  Arriving at the output layer, following the same iterative scheme we will have

$$
\begin{cases}
\mathbb{P}_\chi\left(O^{(c)} \mid \mathcal{W}\right) \xrightarrow{N_L \to \infty} \mathcal{N}\left(\left\langle O^{(c)}\right\rangle, \sigma_w^2 \overline{\mathrm{Var}_\chi\left(g_j^{(L)}\right)}\right) \\
\mathbb{P}_\mathcal{W}\left(\left\langle O^{(c)}\right\rangle\right) \xrightarrow{N_L \to \infty} \mathcal{N}\left(0, \sigma_w^2 \overline{\left\langle g_j^{(L)}\right\rangle}^2\right)
\end{cases}
\tag{144}
$$

These two distributions are the only ingredient that we need to replicate the steps described at the end of App. D.1 to get $\mathbb{P}_\mathcal{W}\left(f_0\right)$.

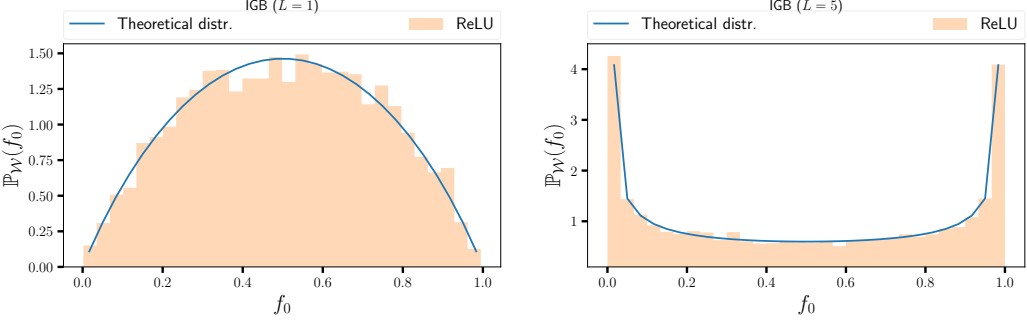

Figure 7: Effects induced by the depth of the network. The case of ReLU is emblematic because it clearly shows the differences in the two cases. For a single hidden layer network (left) we have a stable distribution (*i.e.* that does not narrow around $\mathbb{E}_\mathcal{W}\left(f_0\right) = 0.5$ for $D \to \infty$) but still peaked on $\mathbb{E}_\mathcal{W}\left(f_0\right) = 0.5$. Increasing the number of hidden layers the probability mass moves from the center to the extremes. For these simulations we used the `GB` dataset on model `MHLP` (see App. I.4 for more details).

## F.2  Convergence of $\mathbb{P}_\mathcal{W}\left(f_0\right)$ in deep neural networks

Here we prove the result discussed in Sec.4.2, *i.e.* that the distribution of $\mathbb{P}_\mathcal{W}\left(f_0\right)$ converge to a delta peaked distribution in a multi-layer perceptron with ReLU activation when the number of layers goes to infinity. More precisely

**Theorem F.1** $(\lim_{L \to \infty} \mathbb{P}_\mathcal{W}\left(f_0\right))$**.** *Let us consider a multi-layer perceptron with $L$ hidden layers and ReLU activation function. Let us assume that, in the $\lim_{N_l \to \infty} \forall l \in [0, \dots, L]$, w.h.p.:*

$$
Var_\chi\left(h^{(l)}\right) > 0, \ \forall l
\tag{145}
$$

*and in particular*

$$
\lim_{l \to \infty} Var_\chi\left(h^{(l)}\right) > 0,
\tag{146}
$$

*where $Var_\chi\left(h^{(l)}\right)$ indicate the variance of $\mathbb{P}_\chi\left(h^{(l)} \mid \mathcal{W}\right)$. Then*

$$
\lim_{L \to \infty} \frac{Var_\mathcal{W}\left(\left\langle O^{(c)}\right\rangle\right)}{Var_\chi\left(O^{(c)}\right)} = \infty
\tag{147}
$$

*where $Var_\mathcal{W}\left(\left\langle O^{(c)}\right\rangle\right)$ is the variance of $\mathbb{P}_\mathcal{W}\left(\left\langle O^{(c)}\right\rangle\right)$ and $Var_\chi\left(O^{(c)}\right)$ the variance of $\mathbb{P}_\chi\left(O^{(c)} \mid \mathcal{W}\right)$.*

To prove equation 147 we will show that

$$\text{Var}_{\mathcal{W}}\left(\left\langle h^{(l+1)}\right\rangle\right) > (K + \epsilon)\text{Var}_{\mathcal{W}}\left(\left\langle h^{(l)}\right\rangle\right), \tag{148}$$

$$\text{Var}_{\chi}\left(h^{(l+1)}\right) < (K - \epsilon)\text{Var}_{\chi}\left(h^{(l)}\right), \tag{149}$$

where $K$ is a constant. Once proved the above relations we can, indeed, write

$$\frac{\text{Var}_{\mathcal{W}}\left(\left\langle h^{(l+2)}\right\rangle\right)}{\text{Var}_{\chi}\left(h^{(l+2)}\right)} > \left(\frac{1}{K}\right)^l \frac{\text{Var}_{\mathcal{W}}\left(\left\langle h^{(2)}\right\rangle\right)}{\text{Var}_{\chi}\left(h^{(2)}\right)} (K + \epsilon)^l > l \underbrace{\left(\frac{1}{K} \frac{\text{Var}_{\mathcal{W}}\left(\left\langle h^{(2)}\right\rangle\right)}{\text{Var}_{\chi}\left(h^{(2)}\right)} \epsilon\right)}_{\equiv \frac{1}{C}}$$

$$\implies \frac{\text{Var}_{\chi}\left(h^{(l+2)}\right)}{\text{Var}_{\mathcal{W}}\left(\left\langle h^{(l+2)}\right\rangle\right)} < C\frac{1}{l}. \tag{150}$$

Therefore if we consider an infinite depth network, *i.e.* $L \to \infty$, we have:

$$\frac{\text{Var}_{\chi}\left(h^{(L)}\right)}{\text{Var}_{\mathcal{W}}\left(\left\langle h^{(L)}\right\rangle\right)} = \mathcal{O}\left(\frac{1}{L}\right) \implies \frac{\text{Var}_{\mathcal{W}}\left(\left\langle O^{(c)}\right\rangle\right)}{\text{Var}_{\chi}\left(O^{(c)}\right)} = \mathcal{O}\left(\frac{1}{L}\right) \implies \lim_{L\to\infty} \frac{\text{Var}_{\mathcal{W}}\left(\left\langle O^{(c)}\right\rangle\right)}{\text{Var}_{\chi}\left(O^{(c)}\right)} = 0 \tag{151}$$

*Proof.* Let us start from equation 148. From equation 140

$$\text{Var}_{\mathcal{W}}\left(\left\langle h^{(l+1)}\right\rangle\right) = \sigma_w^2 \overline{\left\langle g_j^{(l)}\right\rangle}^2. \tag{152}$$

$$\overline{\left\langle h_j^{(l)}\right\rangle}^2 = \int_{-\infty}^{\infty} \mathbb{P}_{\mathcal{W}}\left(\left\langle h^{(l)}\right\rangle = x\right) x^2 dx =$$
$$= \int_{-\infty}^{0} \mathbb{P}_{\mathcal{W}}\left(\left\langle h^{(l)}\right\rangle = x\right) x^2 dx + \int_{0}^{\infty} \mathbb{P}_{\mathcal{W}}\left(\left\langle h^{(l)}\right\rangle = x\right) x^2 dx. \tag{153}$$

We now use the symmetry of $\mathbb{P}_{\mathcal{W}}\left(\left\langle h^{(l)}\right\rangle\right)$ (which is a Gaussian centered in 0), *i.e.*

$$\mathbb{P}_{\mathcal{W}}\left(\left\langle h^{(l)}\right\rangle = x\right) = \mathbb{P}_{\mathcal{W}}\left(\left\langle h^{(l)}\right\rangle = -x\right), \tag{154}$$

to rewrite equation 153 as

$$\frac{1}{2}\overline{\left\langle h_j^{(l)}\right\rangle}^2 = \int_{0}^{\infty} \mathbb{P}_{\mathcal{W}}\left(\left\langle h^{(l)}\right\rangle = \mu\right) (\mu)^2 d\mu \equiv$$
$$\equiv \int_{0}^{\infty} \mathbb{P}_{\mathcal{W}}\left(\left\langle h^{(l)}\right\rangle = \mu\right) (\mu_+(\mu) + \mu_-(\mu))^2 d\mu, \tag{155}$$

where we defined

$$\mu_+(\mu) = \int_{0}^{\infty} \mathcal{N}\left(x; \mu, \text{Var}_{\chi}\left(h^{(l+1)}\right)\right) x dx > 0,$$
$$\mu_-(\mu) = \int_{-\infty}^{0} \mathcal{N}\left(x; \mu, \text{Var}_{\chi}\left(h^{(l+1)}\right)\right) x dx < 0. \tag{156}$$

We can thus rewrite

$$\frac{1}{2}\overline{\left\langle h_j^{(l)}\right\rangle}^2 = \int_{0}^{\infty} \mathbb{P}_{\mathcal{W}}\left(\left\langle h_j^{(l)}\right\rangle = \mu\right) \big(\mu_+(\mu)^2 + \mu_-(\mu)^2 + 2\underbrace{\mu_+\mu_-}_{\leq 0}\big) d\mu. \tag{157}$$

Note that $\mu_+\mu_- = 0$ implies either $\mu_+ = 0$ or $\mu_- = 0$ *i.e.* a distribution $\mathbb{P}_\chi\left(h^{(l)} \mid \mathcal{W}\right)$ with non-positive or non-negative support. Since $\mathbb{P}_\chi\left(h^{(l)} \mid \mathcal{W}\right)$ is a Gaussian distribution this may only happen in the limit of its variance going to 0 [24]. By hypothesis we excluded this possibility (equation 145 and equation 146); therefore $\mu_+\mu_- < 0$ and we can rewrite

$$\frac{1}{2}\overline{\left\langle h_j^{(l)}\right\rangle^2} < \underbrace{\int_0^\infty \mathbb{P}_\mathcal{W}\left(\left\langle h^{(l)}\right\rangle = \mu\right)\left(\mu_+(\mu)^2 + \mu_-(\mu)^2\right)d\mu}_{\equiv I} . \tag{158}$$

Now let us focus on the integral $I$ of equation 158. We note that

$$-\mu_-(\mu) = -\int_{-\infty}^0 \mathcal{N}\left(x; \mu, \text{Var}_\chi\left(h^{(l+1)}\right)\right)x\,dx = \tag{159}$$

$$= \int_0^\infty \mathcal{N}\left(y; -\mu, \text{Var}_\chi\left(h^{(l+1)}\right)\right)y\,dy = \mu_+(-\mu) \tag{160}$$

where in the second step we just changed the integration variable $x \to y \equiv -x$. We can therefore rewrite

$$I = \int_0^\infty \mathbb{P}_\mathcal{W}\left(\left\langle h^{(l)}\right\rangle = \mu\right)\left(\mu_+(\mu)^2 + \mu_+(-\mu)^2\right)d\mu = \int_{-\infty}^\infty \mathbb{P}_\mathcal{W}\left(\left\langle h^{(l)}\right\rangle = \mu\right)\mu_+(\mu)^2 d\mu$$
$$\overline{\left\langle g^{(l)}\right\rangle^2} \tag{161}$$

where for the second step we used again equation 154. To summarize we showed that

$$\overline{\left\langle g^{(l)}\right\rangle^2} > \left(\frac{1}{2}\right)\overline{\left\langle h^{(l)}\right\rangle^2} \tag{162}$$

Now let us pass to another quantity:

$$\overline{\left\langle \left(g^{(l)}\right)^2\right\rangle} \equiv \overline{\left\langle x^2\right\rangle_{g^{(l)}}} = \int_{-\infty}^\infty \mathbb{P}_\mathcal{W}\left(\left\langle h^{(l)}\right\rangle = \mu\right)\left\langle x^2\right\rangle_{g^{(l)}}(\mu)\,d\mu =$$
$$= \int_0^\infty \mathbb{P}_\mathcal{W}\left(\left\langle h^{(l)}\right\rangle = \mu\right)\left(\left\langle x^2\right\rangle_{g^{(l)}}(\mu) + \left\langle x^2\right\rangle_{g^{(l)}}(-\mu)\right)\,d\mu = \tag{163}$$
$$= \int_0^\infty \mathbb{P}_\mathcal{W}\left(\left\langle h^{(l)}\right\rangle = \mu\right)\left(\left\langle x^2\right\rangle_+(\mu) + \left\langle x^2\right\rangle_-(\mu)\right)\,d\mu,$$

where, analogously to equation 156, we defined

$$\left\langle x^2\right\rangle_+(\mu) = \int_0^\infty \mathcal{N}\left(x; \mu, \text{Var}_\chi\left(h^{(l+1)}\right)\right)x^2\,dx\,,$$
$$\left\langle x^2\right\rangle_-(\mu) = \int_{-\infty}^0 \mathcal{N}\left(x; \mu, \text{Var}_\chi\left(h^{(l+1)}\right)\right)x^2\,dx. \tag{164}$$

From the above definitions, we see that

$$\left\langle x^2\right\rangle_+(\mu) + \left\langle x^2\right\rangle_-(\mu) = \left\langle x^2\right\rangle_{h^{(l)}}(\mu); \tag{165}$$

therefore substituting into equation 163 we get

$$\overline{\left\langle \left(g^{(l)}\right)^2\right\rangle} = \int_0^\infty \mathbb{P}_\mathcal{W}\left(\left\langle h^{(l)}\right\rangle = \mu\right)\left\langle x^2\right\rangle_{h^{(l)}}(\mu)\,d\mu =$$
$$= \frac{1}{2}\int_{-\infty}^\infty \mathbb{P}_\mathcal{W}\left(\left\langle h^{(l)}\right\rangle = \mu\right)\left\langle x^2\right\rangle_{h^{(l)}}(\mu)\,d\mu = \frac{1}{2}\overline{\left\langle \left(h^{(l)}\right)^2\right\rangle}. \tag{166}$$

---

[24] *i.e.*, if the Gaussian shrink into a Dirac delta distribution

Finally, let us consider

$$\overline{\text{Var}_\chi\left(g^{(l)}\right)} \equiv \overline{\left\langle\left(g^{(l)}\right)^2\right\rangle} - \overline{\left\langle g^{(l)}\right\rangle^2} = \frac{1}{2}\overline{\left\langle\left(h^{(l)}\right)^2\right\rangle} - \overline{\left\langle g^{(l)}\right\rangle^2} <$$

$$< \frac{1}{2}\overline{\left\langle\left(h^{(l)}\right)^2\right\rangle} - \frac{1}{2}\overline{\left\langle h^{(l)}\right\rangle^2} = \frac{1}{2}\overline{\text{Var}_\chi\left(h^{(l)}\right)} \xrightarrow{N_{l-1}\to\infty} \frac{1}{2}\text{Var}_\chi\left(h^{(l)}\right). \tag{167}$$

where in the last step we used the concentration result discussed in App. B.2.2, *i.e.* that the distribution of $\text{Var}_\chi\left(h^{(l)}\right)$ asymptotically narrows around its mean value, becoming, *w.h.p.* independent of the realization $\mathcal{W}$.

Now we have all the ingredient we need; in fact, from equation 140 we know that

$$\text{Var}_\chi\left(h^{(l)}\right) = \sigma_w^2 \overline{\text{Var}_\chi\left(g_j^{(l-1)}\right)}. \tag{168}$$

Therefore, from equation 167, we can conclude

$$\text{Var}_\chi\left(h^{(l+1)}\right) < \left(\frac{\sigma_w^2}{2} - \epsilon_l'\right)\text{Var}_\chi\left(h^{(l)}\right), \tag{169}$$

with $\epsilon_l' > 0$. Similarly, combining equation 141 with equation 162 we get

$$\text{Var}_\mathcal{W}\left(\left\langle h^{(l+1)}\right\rangle\right) > \left(\frac{\sigma_w^2}{2} + \epsilon_l\right)\text{Var}_\mathcal{W}\left(\left\langle h^{(l)}\right\rangle\right), \tag{170}$$

with $\epsilon_l > 0$. Calling

$$\epsilon \equiv \inf_l \epsilon_l, \tag{171}$$

we can finally write

$$\frac{\text{Var}_\mathcal{W}\left(\left\langle h^{(l+2)}\right\rangle\right)}{\text{Var}_\chi\left(h^{(l+2)}\right)} > \left(\frac{2}{\sigma_w^2}\right)^l \frac{\text{Var}_\mathcal{W}\left(\left\langle h^{(2)}\right\rangle\right)}{\text{Var}_\chi\left(h^{(2)}\right)}\left(\frac{\sigma_w^2}{2} + \epsilon\right)^l > l\underbrace{\left(\frac{2}{\sigma_w^2}\frac{\text{Var}_\mathcal{W}\left(\left\langle h^{(2)}\right\rangle\right)}{\text{Var}_\chi\left(h^{(2)}\right)}\epsilon\right)}_{\equiv\frac{1}{C}} \tag{172}$$

$$\implies \frac{\text{Var}_\chi\left(h^{(l+2)}\right)}{\text{Var}_\mathcal{W}\left(\left\langle h^{(l+2)}\right\rangle\right)} < C\frac{1}{l}.$$

Therefore if we consider an infinite depth network, *i.e.* $L \to \infty$, we have:

$$\frac{\text{Var}_\chi\left(h^{(L)}\right)}{\text{Var}_\mathcal{W}\left(\left\langle h^{(L)}\right\rangle\right)} = \mathcal{O}\left(\frac{1}{L}\right) \implies \frac{\text{Var}_\mathcal{W}\left(\left\langle O^{(c)}\right\rangle\right)}{\text{Var}_\chi\left(O^{(c)}\right)} = \mathcal{O}\left(\frac{1}{L}\right) \implies \lim_{L\to\infty}\frac{\text{Var}_\mathcal{W}\left(\left\langle O^{(c)}\right\rangle\right)}{\text{Var}_\chi\left(O^{(c)}\right)} = 0. \tag{173}$$

$\square$

## G CONDITIONS FOR THE EMERGENCE/ABSENCE OF IGB

We discussed how the choice of the activation function can induce IGB. To make the discussion more concrete we picked some emblematic examples (*i.e.* the linear activation function and the ReLU) that we were able to work out analytically. Here we will discuss the problem from a more broad perspective showing that these examples belongs to different classes of activation functions. We can then conclude the emergence of IGB, for a generic activation function, depending on which of the two class it belongs.

The fundamental difference that distinguishes the two cases discussed is as follows: with the linear activation function the output nodes are asymptotically identically distributed; hence we have absence of IGB. With the ReLU in contrast we observe a break in the symmetry of the output nodes. These are still asymptotically Gaussian, but centered at different points.

The symmetry breaking is related to the fact that the ReLU in the first hidden layer maps the null-averaged inputs to a r.v., $g_i^{(1)}$, with a nonzero mean. We can therefore consider a generic activation function and distinguish two cases. In the following we will consider a generic activation function from each of the two classes and repeat the analysis to show the emergence/absence of IGB. We will use the same setting used in the previous analysis (*i.i.d.* Gaussian input and Kaiming initialization for the weights). As seen in App. C.1, irrespective to the activation function used we have

$$\mathbb{P}_\chi\left(h_i^{(1)} \mid \mathcal{W}\right) \xrightarrow{d\to\infty} \mathcal{N}\left(0, \sigma_w^2\right). \tag{174}$$

### G.1 NON-NULL MEAN ACTIVATION FUNCTION

Let us consider a generic activation function $g\left(\cdot\right)$, such that

$$\left\langle g_i^{(1)}\right\rangle \equiv \left\langle g\left(h_i^{(1)}\right)\right\rangle = \left\langle g^{(1)}\right\rangle \neq 0. \tag{175}$$

where the second step follow from the fact that $\{h_i^{(1)}\}$ are identically distributed. Starting from this hypothesis on the activation function (equation 175) we can follow exactly the same analysis presented in App.F.1. In particular:

$$\mathbb{P}_\chi\left(h_i^{(2)} \mid \mathcal{W}\right) \xrightarrow{N_1 \to \infty} \mathcal{N}\left(\left\langle g^{(1)}\right\rangle S_{\boldsymbol{w}_i}, \mathrm{Var}_\chi\left(g^{(1)}\right) S_{\boldsymbol{w}_i^2}\right) =$$
$$\mathcal{N}\left(\left\langle g^{(1)}\right\rangle S_{\boldsymbol{w}_i}, \mathrm{Var}_\chi\left(g^{(1)}\right) \overline{S_{\boldsymbol{w}_i^2}}\right) , \tag{176}$$

where, again, $\mathrm{Var}_\chi\left(g^{(1)}\right)$ follows the definition from equation 130. From equation 176 we can already see the nodes symmetry breaking; the centers of the distributions is a r.v. varying from node to node. Note that hor a single hidden layer architecture

$$\mathbb{P}_\chi\left(O^{(c)} \mid \mathcal{W}\right) = \mathbb{P}_\chi\left(h_c^{(2)} \mid \mathcal{W}\right). \tag{177}$$

For a generic deep architecture we can keep following the analysis of App.F.1, substituiting the right expression to $\mathbb{P}_\chi\left(g_i^{(2)} \mid \mathcal{W}\right)$ dependent to the specific used activation function. Proceeding in this way we will find, for each layer, a symmetry breaking in the nodes distribution, similarly to what we observed in the second layer (equation 176).

### G.2 NULL MEAN ACTIVATION FUNCTION

Let us now consider the opposite scenario, *i.e.* an activation function $g\left(\cdot\right)$, such that

$$\left\langle g_i^{(1)}\right\rangle \equiv \left\langle g\left(h_i^{(1)}\right)\right\rangle = \left\langle g^{(1)}\right\rangle = 0. \tag{178}$$

where, again, the second step follow from the fact that $\{h_i^{(1)}\}$ are identically distributed. In this case, instead of equation 176, we will have

$$\mathbb{P}_\chi\left(h_i^{(2)} \mid \mathcal{W}\right) \xrightarrow{N_1 \to \infty} \mathcal{N}\left(0, \mathrm{Var}_\chi\left(g^{(1)}\right) S_{\boldsymbol{w}_i^2}\right) = \mathcal{N}\left(0, \mathrm{Var}_\chi\left(g^{(1)}\right) \overline{S_{\boldsymbol{w}_i^2}}\right). \tag{179}$$

Therefore we don't have nodes symmetry breaking in this case. We can, therefore, pass to the next layer getting a similar result (asymptotically equally distributed nodes). We can finally iterate till reaching the output nodes, which will be equally distributed as well. If we consider a generic activation function may be not trivial to associate it in one of the two classes we just discussed. Below we provide an example of functions set whose elements all satisfies equation 178.

### G.2.1 ANTI-SYMMETRIC ACTIVATION FUNCTIONS

The analysis presented in Sec. 4.1 highlighted the differences induced by the choice of the activation function (see Fig. 5). In particular, we show how this choice can determine the emergence of IGB, absent with a (trivial) identity activation function. Here we will elaborate more on this showing the existence of a class of activation functions (anti-symmetric activation functions) that, similar to the identity[25], never show IGB. We can reformulate the definition of IGB (Def. 3.1) to the framework of the analysis presented.

---

[25]the identity is an element of this class

**Definition G.1** (IGB)**.** Let us assume that $\mathbb{P}_\chi \left( O^{(c)} \mid \mathcal{W} \right)$ asymptotically converge to a Gaussian distribution (*e.g.* see App. C.3) with a non-zero variance. Then, if

$$\mathbb{P}_\mathcal{W} \left( \left\langle O^{(c)} \right\rangle = x \right) = \delta \left( x - a \right), \ \ \forall c \tag{180}$$

for some $a \in \mathbb{R}$, we have absence of IGB.

If instead equation 180 does not hold, we have IGB, *i.e.* we will observe a disproportion between the values $\{f_i\}$, even in the limit $D \to \infty$ (so not due to finite size effects).

For concreteness we consider an example, to show the general idea. Let us consider the following activation function

$$g_i^{(l)} = g \left( h_i^{(l)} \right) = \tanh \left( h_i^{(l)} \right), \ \ l \in \{0, \dots, L\}, \ i \in \{0, \dots, N_l\}. \tag{181}$$

From equation 71 we know the asymptotic distribution of $\mathbb{P}_\chi \left( h_i^{(1)} \mid \mathcal{W} \right)$. Note that this result is independent of the choice of the activation function. Now we ask ourselves "How does the distribution change after passing through the activation function (equation 181)?" We can use the relation

$$\mathbb{P} \left( x \right) dx = \mathbb{P} \left( y(x) \right) dy, \tag{182}$$

to find

$$\mathbb{P}_\chi \left( g_i^{(1)} = y \mid \mathcal{W} \right) = \frac{e^{-\frac{(\text{atanh}(y))^2}{2\sigma_{h^{(l)}}^2}}}{\sqrt{2\pi\sigma_{h^{(l)}}^2}} \frac{1}{(1 - y^2)}. \tag{183}$$

We can immediately see from equation 183 that the distribution is symmetric, *i.e.* $\mathbb{P}_\chi \left( g_i^{(1)} = y \mid \mathcal{W} \right) = \mathbb{P}_\chi \left( g_i^{(1)} = -y \mid \mathcal{W} \right)$. This is a direct consequence of the anti-symmetricity of the activation function and from the symmetry of $\mathbb{P}_\chi \left( h_i^{(l)} \mid \mathcal{W} \right)$ [26]. This implies that

$$\left\langle g_i^{(1)} \right\rangle = 0 \Longrightarrow \left\langle h_j^{(2)} \right\rangle = 0, \quad \forall i, j \tag{184}$$

This represents a crucial difference with respect to the examples analyzed in the previous section because now all nodes $\{h_j^{(2)}\}$ are identically distributed, and, in particular,

$$\mathbb{P}_\mathcal{W} \left( \left\langle h^{(2)} \right\rangle = x \right) = \delta \left( x \right). \tag{185}$$

Propagating further across the remaining hidden layers, the situation doesn't change. Apart from the value of the variance, there is in fact no difference between $\mathbb{P}_\chi \left( h_i^{(2)} \mid \mathcal{W} \right)$ and $\mathbb{P}_\chi \left( h_i^{(1)} \mid \mathcal{W} \right)$, so we can reiterate the same reasoning, finding

$$\mathbb{P}_\mathcal{W} \left( \left\langle h^{(3)} \right\rangle = x \right) = \delta \left( x \right), \tag{186}$$

and so on $\forall l \in \{3, \dots, L + 1\}$. In particular, for $l = L + 1$ this implies that

$$\mathbb{P}_\mathcal{W} \left( \left\langle O^{(c)} \right\rangle = x \right) = \delta \left( x \right). \tag{187}$$

which means that, employing an anti-symmetric activation function (as $\tanh$), we don't observe IGB[27].

### G.3   ELIMINATE/TRIGGER IGB WITH A GENERIC ACTIVATION FUNCTION

---

[26]we would get therefore an analogue result considering a different anti-symmetric activation function

[27]note that we also are implicitly assuming that the initialization chosen keeps the variance of $\mathbb{P}_\chi \left( O^{(c)} \mid \mathcal{W} \right)$ asymptotically non-null

We just showed that the set of activation functions can be partitioned in two subsets; depending on which subset a given function belongs we will observe or not IGB. We emphasize however that our analysis allows us to characterize these subset by a simple attribute (*i.e.* equation 175 and equation 178). This means that, starting from a given activation function we can, in principle, modify it getting a new function, member of the opposite subset. In other words, by slightly modifying the definition of an activation function (or acting by some regularization) we can start by a network with IGB and make it disappear (or vice-versa). This idea is reminiscent of some procedures introduced in machine learning in similar contexts. For example in Ioffe & Szegedy (2015) they introduce the Batch-Normalization as a scaling procedure to mitigate the *Internal Covariate Shift*, *i.e.* the change in the distribution of network activations due to the change in network parameters during training. In our case we are not considering

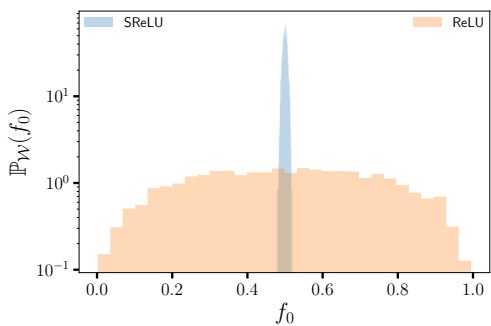

Figure 8: Comparison between SReLU and ReLU. A simple shift of the ReLU can make it satisfy equation 179 eliminating IGB. Here we can observe, from $\mathbb{P}_{\mathcal{W}}(f_0)$, the difference in the two regimes. For these simulations we used the dataset `GB` on the model `SHLP`. (see App. I.4 for more details).

these differences induced by the learning, but rather differences between nodes due to the initialization. On the other hand, for what just said, our analysis suggest that a similar scaling could be effective in eliminate IGB as well.

To make the discussion more concrete we provide now an example. We saw how the ReLU trigger IGB. Let us consider now a variant of ReLU defined as follow

$$g\left(h_i^{(1)}\right) = \begin{cases} h_i^{(1)} - \frac{\sigma_w}{\sqrt{2\pi}}, & \text{if } h_i^{(1)} > 0\,, \\ -\frac{\sigma_w}{\sqrt{2\pi}} & \text{if } h_i^{(1)} < 0\,, \end{cases} \tag{188}$$

where $\frac{\sigma_w}{\sqrt{2\pi}}$ is the ReLU mean value (we can easily compute it from equation 78 ). This shifted variant of ReLU, which we call SReLU , satisfy equation 178, and belongs therefore to the subset of functions with absence of IGB (see Fig. 8).

In this example we could easily compute the expectation value and directly subtract it to shift the activation function to a null mean value. For an arbitrary activation function we could however perform a shift based on the empirical average computed on the forwarding batch, in a similar spirit to the approach proposed in Ioffe & Szegedy (2015).

## H MULTI-CLASS EXTENSION

We want now to extend the analysis to the multi-class case, *i.e.* to problem with $N_C > 2$. Following the framework introduced we can easily extend the computation. We will again consider the distribution among the $N_C$ classes of the dataset elements after initialization. In particular we can define $N_C$ values associated to the fraction of dataset elements classified as belonging to each of the $N_C$ classes. The statistics of these variables will clearly change with the number of classes. To take track of this we slightly modify the notation, for the multi-class. We indicate explicitly the number of classes used to build the statistics, *i.e.* $\left\{f_i^{(M)}\right\}_{i=0}^{M-1}$, where $M$ is the number of classes considered.

We could also consider the distribution of the sorted frequencies; in other word in each experiment we order the classes according to the corresponding frequencies and not by their label. To distinguish this frequencies (and their statistics) from the set $\left\{f_i^{(M)}\right\}_{i=0}^{M-1}$, we call this new set $\left\{f_{\tilde{i}}^{(M)}\right\}_{\tilde{i}=0}^{M-1}$. In particular $f_{\tilde{0}}^{(M)}$ indicates the bigger frequency among the $M$, $f_{\tilde{1}}^{(M)}$ the second one and so on. Finally we add the information of the cardinality also on the output nodes set $\left\{O_M^{(c)}\right\}_{c=0}^{M-1}$. Analogously to

the set of fractions, we define the set of ranked output nodes $\left\{O_M^{(\tilde{c})}\right\}_{\tilde{c}=0}^{M-1}$ In the following, similarly to Sec. 4.1 we will derive the distribution of $f_0^{(M)}$.

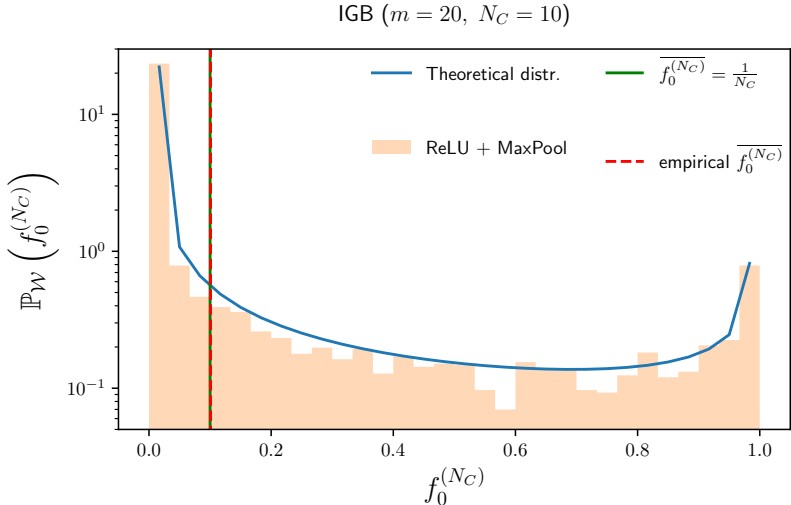

Figure 9: $\mathbb{P}_{\mathcal{W}}\left(f_0\right)$ in a multi-class problem with $N_C = 10$. The empirical mean value is reported; note that despite the difference in the distribution the mean value is the same that we would observe using a linear activation function $\overline{f_0^{(N_C)}} = \frac{1}{N_C}$. For these simulations we used the dataset GBon the model SHLPwith MaxPooling (m=20). (see App. I.4 for more details).

We will make in the derivation of $f_0^{(M)}$ the following approximation: *We will repeat the same approach of the binary setting comparing the generic class "0" with the class with bigger mean output value among the $N_C - 1$ remaining classes.*

The first ingredient that we need is therefore the statistics of $O_{N_C-1}^{(\tilde{0})}$. From the analysis of previous sections, we know that $\mathbb{P}_{\chi}\left(O_{N_C-1}^{(\tilde{0})} \mid \mathcal{W}\right)$ is asymptotically Gaussian. The mean value of this Gaussian variable will follow the distribution of the maximum among $N_C - 1$ drawn from the Gaussian distribution $\mathbb{P}_{\mathcal{W}}\left(\langle O^{(c)}\rangle\right)$ as, by definition,

$$\left\langle O_{N_C-1}^{(\tilde{0})}\right\rangle \equiv \max_{c \in \{1,\ldots,N_C\}} \left\langle O^{(c)}\right\rangle . \tag{189}$$

**Maximum over $N_C - 1$ Gaussian draws**   We will follow the same approach used in App. C.1 . We will again start from equation 82. In this case,

$$\mathbb{P}\left(X = x\right) \equiv \mathbb{P}_{\mathcal{W}}\left(\left\langle O^{(c)}\right\rangle = x\right) = \mathcal{N}\left(x; 0, \hat{\sigma}^2\right) , \tag{190}$$

where we used a generic $\hat{\sigma}^2$ to indicate the variance. We will perform the computation in this general settings; then to analyze the specific cases we can just substitute the right variance for the specific case (see App. C.2).

$$F_X(y) \equiv \int_{-\infty}^{y} \mathbb{P}\left(X = x\right) dx = \int_{-\infty}^{y} \mathcal{N}\left(x; 0, \hat{\sigma}^2\right) dx = \frac{1}{2}\left(1 + \operatorname{erf}\left(\frac{y}{\sqrt{2}\hat{\sigma}}\right)\right) . \tag{191}$$

Now, putting all pieces together in equation 82, we get our target distribution,

$$\mathbb{P}\left(Y = y\right) \equiv \mathbb{P}_{\mathcal{W}}\left(\left\langle O_{N_C-1}^{(\tilde{0})}\right\rangle = y\right) = (N_C - 1)\,\mathcal{N}\left(y; 0, \tilde{\sigma}^2\right)\left(\frac{1}{2}\left(1 + \operatorname{erf}\left(\frac{y}{\sqrt{2}\tilde{\sigma}}\right)\right)\right)^{N_C-2} . \tag{192}$$

Now we can proceed following again the steps presented at the end of Sec. 4.1. The only difference we will have is in equation 113; now we will have

$$\mathbb{P}_{\mathcal{W}}\left(f_0^{(N_C)}\right) = \int_{-\infty}^{\infty} \mathbb{P}_{\mathcal{W}}\left(\left\langle O_{N_C}^{(0)}\right\rangle = \tilde{x}\right) \mathbb{P}_{\mathcal{W}}\left(\left\langle O_{N_C-1}^{(\tilde{0})}\right\rangle = \tilde{x} - \hat{\Delta}\left(f_0^{(N_C)}\right)\right) d\tilde{x}, \quad (193)$$

where $\left\langle O_{N_C-1}^{(\tilde{0})}\right\rangle$ the biggest mean output among the $N_C - 1$. Note that $\hat{\Delta}\left(f_0^{(N_C)}\right)$ is a function of $f_0^{(N_C)}$. Substituting equation 190 and equation 192 we have:

$$\mathbb{P}_{\mathcal{W}}\left(f_0^{(N_C)}\right) = \int_{-\infty}^{\infty} \mathcal{N}\left(\tilde{x}; 0, \hat{\sigma}^2\right)(N_C - 1)\mathcal{N}\left(\tilde{x}; \hat{\Delta}, \hat{\sigma}^2\right)\left(\frac{1}{2}\left(1 + \mathrm{erf}\left(\frac{\tilde{x} - \hat{\Delta}}{\sqrt{2}\hat{\sigma}}\right)\right)\right)^{N_C-2} d\tilde{x}. \quad (194)$$

# I  EXPERIMENTS

The analysis presented is able to identify IGB and analytically describe the phenomena in a variety of settings (e.g., MLPs with different activation functions, arbitrary depth, and the presence or absence of pooling layers). However, IGB is significantly broader in scope, as it may be observed in a wide range of dataset and architectural combinations. Although our theoretical analysis does not go into these scenarios (a further extension of the analytical conclusions to more sophisticated designs will be the focus of future research), we provide some representative instances to demonstrate the breadth of scenarios where IGB is significant in real-world circumstances. The experiments presented, while not exhaustive, aim to provide insights and emphasize three fundamental points that will be developed in a separate publication:

- IGB is amplified in structured data, particularly in cases of strongly correlated data (App. I.1).
- IGB is a ubiquitous phenomenon, that happens with most combinations of datasets and architectures (App. I.1, I.2).
- The differences induced by IGB on the initial state of the network have an impact on its dynamics, rendering it qualitatively distinct, especially in the initial phase, compared to the case without IGB (App. I.3).

## I.1  EXPERIMENTS ON REAL DATA

In this section, we show some empirical evidence of IGB on combinations of models and data which are not covered by our theory, such as CNN architectures and the MNIST, CIFAR10 and CIFAR100 datasets.[28] In App.I.2, we will also include experiments on additional architectures (ResNet and Vision Transformers). The experiments we will discuss do not presume to show in a complete and exhaustive way the range of models and cases in which IGB can occur; this question is outside the scope of our work. Rather, they are some didactic examples intended to show how, beyond the assumptions used in our analysis to quantitatively treat the phenomenon, IGB also emerges in a broader context. We first present and discuss the experiments. For technical details for the reproducibility we refer the reader to App. I.4.

### I.1.1  A PROTOTYPE FOR HIGHLY CORRELATED DATA: MNIST

In the analysis we presented in the main part of the paper, we focused on a simple data structure, to underline effects induced by architecture design: a remarkable conclusion from this setting is the following:
*In a classification problem, an untrained network with biases set to zero and fed with i.i.d. data-points (i.e. different classes are identically distributed) often starts the training process with a strong bias toward one of the classes.*
For our theoretical analysis, we placed ourselves in a setting where the effect of IGB is minimal and unaffected by the data structure. This ensures that the sources of IGB, which our analysis

---

[28]MNIST is a particularly insightful dataset for IGB, because the correlations between pixels are high. This is opposite to what we cover with our theory, which is based on i.i.d. inputs.

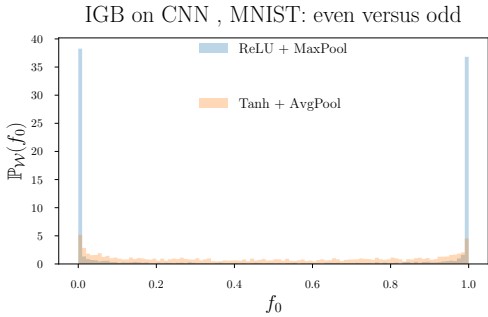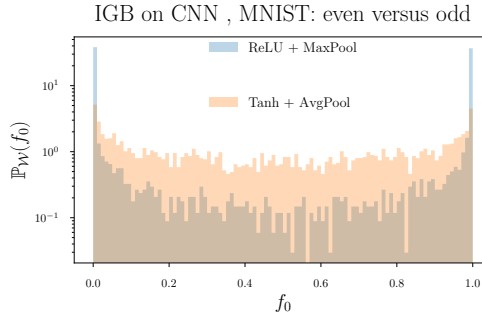

Figure 10: Comparison of $\mathbb{P}_{\chi}(f_0)$ between two untrained neural networks fed with MNIST dataset; the ten classes of the dataset were merged in two macro groups: even and odd. The two architectures employed in the comparison differ in the choice of activation function and pooling. For these simulations we used the `E&O` dataset on the model `CNN-B`. (see App. I.4 for more details). On the right, the same plot in a logarithmic scale to better visualize the differences in the low-density regions.

links to architectural design elements, do not stem from dataset characteristics. Below, we present experiments in which these hypotheses on data are violated, to demonstrate how the phenomenon manifests itself (more prominently) in real datasets, where:

- Data-points belonging to different classes will not be identically distributed.
- Components of a single data-point will not be independent; for example pixels of an image will clearly be correlated.
- Similarly, correlation between different data-points (belonging to the same class) is possible.

MNIST constitutes a good candidate to represent a scenario of strong correlations in the data. In fact, in this dataset the images are characterized by a small percentage of nonzero pixels; this leads to the formation of areas, within the same image, characterized by similar values and therefore strongly correlated. In addition, image areas are strongly correlated between different elements. For example, we are unlikely to observe the writing of a number on the edges of an image; therefore, different images will have similar values of pixels along the edges. When considering images of the same class, this correlation is even more pronounced. From the MNIST dataset, we defined two macro-classes, even and odd, by merging the 10 starting classes according to their parity. We then propagated the binary dataset, obtained in this way, through two untrained convolutional neural networks. The difference between the two networks lies in the choice of activation and pooling function employed. In particular the dataset is propagated through `CNN-B` (network details in App. I.4). The `CNN-B` model is a CNN where the last convolutional layer is directly connected to the outputs. We choose it this way, because, as earlier shown, IGB cannot arise without hidden layers. This indicates that the observation of IGB is caused by the convolutional layers or, to put it more precisely, it emerges from the propagation of structured data through the convolutional layers.

We show this in Fig. 10, which contains two important messages:

◇ Unlike the cases presented in our theoretical analysis, neither of the two choices has total absence of IGB. This is consistent with the intuition that correlations in the data cause or amplify IGB.

◇ While we observe presence of IGB in both cases, this is more pronounced for the architecture with ReLU and MaxPooling, consistent with what we saw in our study.

These observations show how IGB is a rather universal phenomenon, related to the combined effect of data structure and the design of the neural network.

### I.1.2 A FURTHER EXAMPLE: CIFAR10

We now show a multi-class example, on the same network as in App. I.1.1 (`CNN-B`) we propagated CIFAR10. The results are reported in Fig. 11. As with MNIST, the results are qualitatively consistent

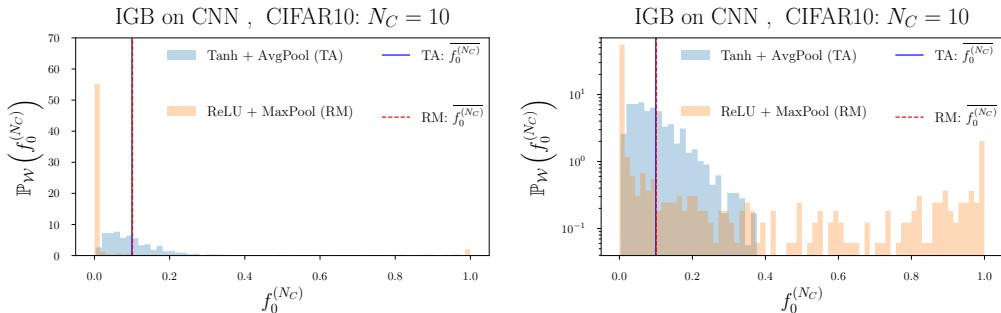

Figure 11: Comparison of $\mathbb{P}_\chi\left(f_c^{(N_C)}\right)$ between two untrained neural networks fed with CIFAR10 dataset (with 10 classes). The two architectures employed in the comparison differ in the choice of activation function and pooling. In the absence of IGB, the distributions would concentrate around the vertical line. For these simulations we used the dataset C10 on the model CNN-B described below (see App. I.4). On the right, the same plot in a logarithmic scale to better visualize the differences in the low-density regions.

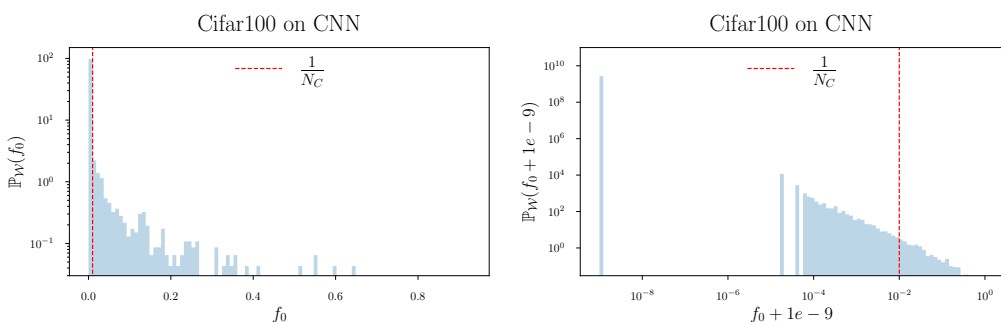

Figure 12: CNN-A (ReLU+MaxPool case) with C100 in this case ($N_C = 100$). The peak at $f_0 = \frac{1}{N_C}$, corresponding to the No IGB case (in the absence of IGB, the distribution should concentrate around this value), is reported as a reference. The plot on the right contains the same data as that on the left, but has a logarithmic $x$ axis. We add a small number ($10^9$) to the value of $f_0$, in order to show the peak at $f_0 = 0$.

with the conclusions of our theory: the network with ReLU and MaxPooling displays a stronger IGB than its counterpart with $\tanh$ and AvgPooling. Note, also, how in both cases, symmetry between classes is preserved at the ensemble level. In both cases, in fact, we get $\overline{f_c^{(N_C)}} = 1/N_C$. To show this we show in Fig. 11 two vertical lines at the mean values calculated on the empirical distributions (histograms), each surrounded by its own uncertainty, estimated through the standard error.

### I.1.3 HIGH CARDINALITY DATASET: CIFAR100

We also provide another example of a multi-class dataset. In this case, we choose Cifar-100 as a prototype of a high-cardinality dataset (with a high number of classes) to demonstrate the presence of IGB in this scenario as well. In the absence of IGB, the distribution should be peaked at $f_0 = 1/N_C = 1/100$. However, the histogram shows a peak at 0 (most of the time, the generic class does not receive any assigned elements). The gap between the peak at 0 and the rest of the distribution, particularly evident in the log-log scale plot, indicates a resolution limit due to the finite number of dataset elements ($D = 10^4$).

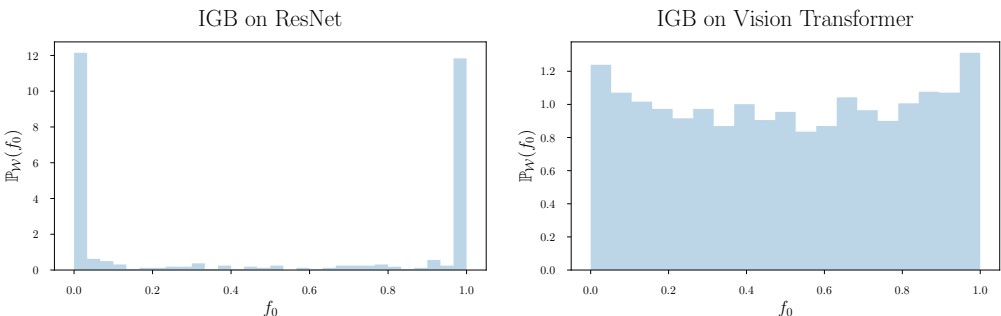

Figure 13: Plot of $\mathbb{P}_{\mathcal{W}}(f_0)$ for a ResNet34 (left), *i.e.* `ResNet`, and a Vision Transformer (right), *i.e.* `ViT` (see Sec. I.4), on two classes extracted from Cifar10 (same setting as Fig. 1 in the main paper). In this case, we also observe the wide profile that characterizes IGB.

## I.2 EXPERIMENTS ON OTHER ARCHITECTURES

The experiments presented in App. I.1 illustrate how IGB can also emerge in CNNs. Below, we provide some examples to give an idea of the breadth of settings and architectures related to the presence of this phenomenon. We specifically conduct simulations on a Residual Network and a Visual Transformer. Similarly to what was observed for CNNs, these architectures also exhibit the presence of IGB (Fig. 13).

## I.3 EFFECTS ON THE TRAINING DYNAMICS

As we will see, the presence or absence of IGB induces important differences in the subsequent training dynamics. The objective of this section is to present some key elements that distinguish between cases with and without IGB. Our intention is not to provide an exhaustive review, as this would demand an extensive discussion beyond the scope of this study. Nevertheless, we present several key experimental results that clearly demonstrate the practical significance of IGB, highlighting that this phenomenon is worth further investigation.
Considering that IGB is observed in a variety of settings, including advanced architectures capable of achieving state-of-the-art performance, it is reasonable to assume that IGB does not profoundly affect convergence at the end of the dynamics. However, the qualitative differences from the case without IGB and the resemblance to curves observed in the context of class imbalance suggest potential additional challenges associated with the presence of IGB. To effectively illustrate the differences in dynamics with and without IGB, we will compare the two cases in the regimes of small and large learning rates (slow/stable and unstable regimes). We will clarify the relevance of these two limits in the discussion.

**Small learning rate** Fig. 14 illustrates the comparison in the small learning rate regime (slow dynamics), which is of particular interest and widely studied (Sarao Mannelli et al., 2020; Tarmoun et al., 2021; Francazi et al., 2022). One noticeable difference is the behavior of individual class curves. In the absence of IGB, the curves averaged over both classes closely align with those of each individual class. In contrast, in the presence of IGB, there is a substantial difference between them; global measures, therefore, such as accuracy or loss do not serve as reliable indicators of individual class performance in this case. The differences in dynamics between the two cases are especially prominent in the initial training phase. Specifically, in presence of IGB, the per-class recall of one class remains low for an extended duration. This closely resembles situations observed in the presence of class imbalance, which has become a recent concern (see *e.g.*, Fig. 1 and Fig. 4 of Francazi et al. (2022)). To gain insight, it is worthwhile to consider the well-established impact of class imbalance bias on dynamics. For instance, Ye et al. (2021); Francazi et al. (2022) explore this from a theoretical perspective, discussing how the initial imbalance between classes leads to learning delays. It is well-known that the slowdown in dynamics poses a limitation, for example, in hyper-parameter tuning (Francazi et al., 2022), as the initial phase of the dynamics is uninformative.

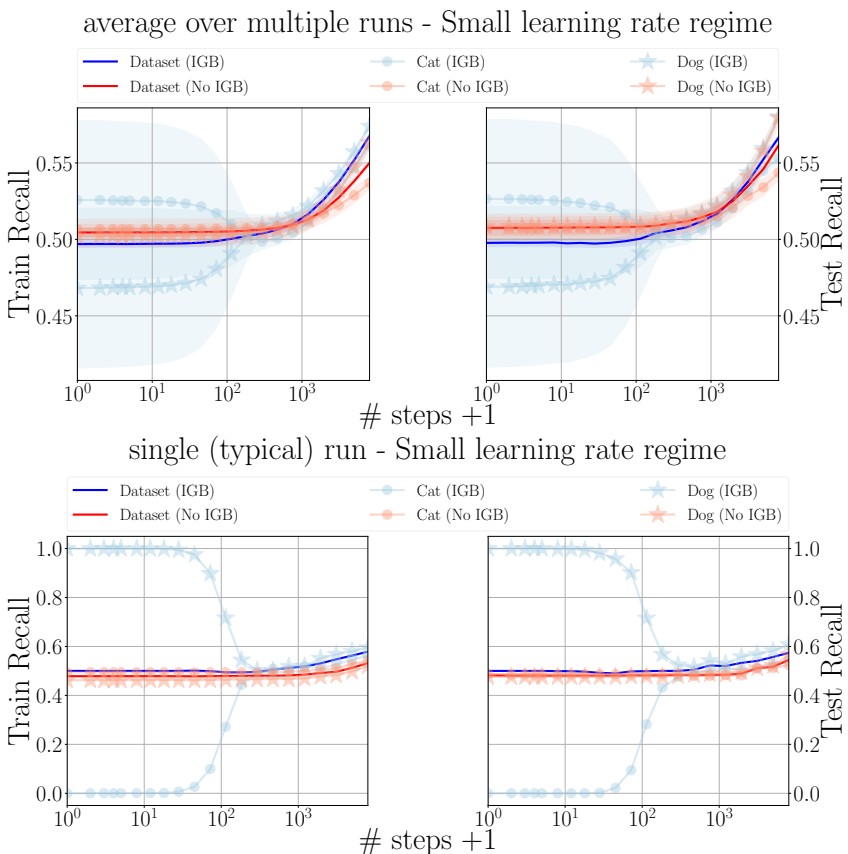

Figure 14: Comparison of the initial phase of the dynamics in the IGB and No IGB cases for `CNN-A` trained on two classes extracted from Cifar10 (same setting as Fig. 1 in the main paper) with a small learning rate (for both cases we used a constant learning rate: $\eta = 10^{-7}$). In the IGB case, by averaging over multiple realizations, it is not possible to observe the initial imbalance, as the symmetry between the two classes is preserved at the ensemble level (consistent with what is shown in the histogram in Fig.1 of the Main paper). Therefore, in addition to the average over multiple realizations (left), the curves corresponding to a typical single run are also presented.

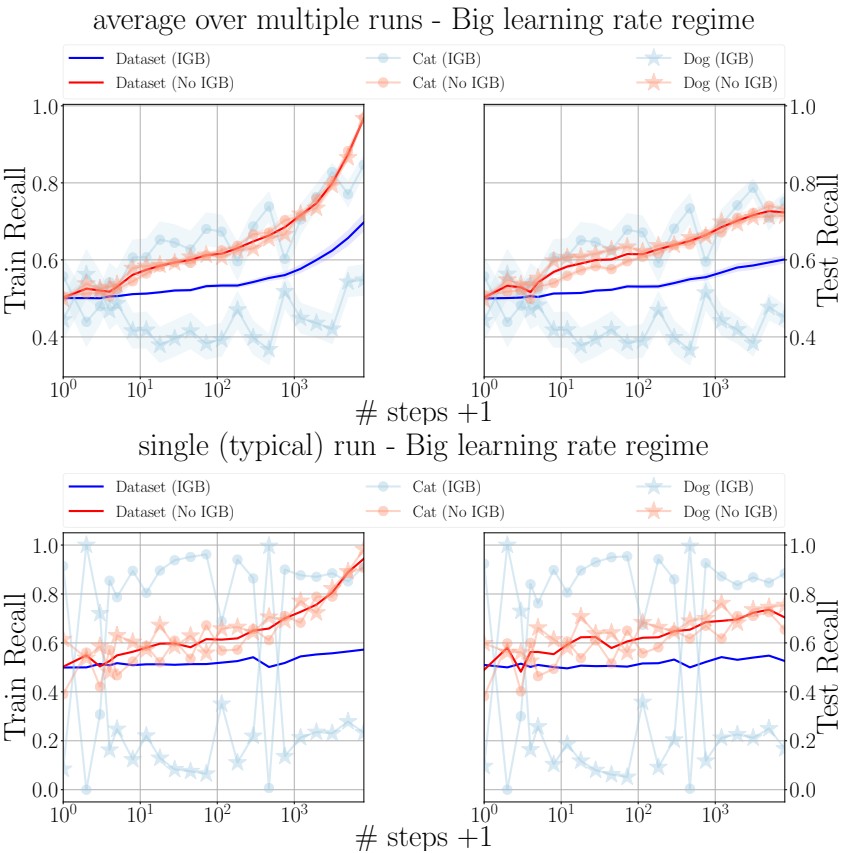

Figure 15: Comparison of the initial phase of the dynamics in the IGB and No IGB cases for `CNN-A` trained on two classes extracted from Cifar10 (same setting as Fig. 1 in the main paper) with a big learning rate (for both cases we used a constant learning rate: $\eta = 10^{-3}$).

**Large learning rate**  In the absence of class imbalance, the slowdown observed in the small learning rate regime is generally resolved by using higher learning rate values. In the presence of class imbalance, the difference in per-class gradients makes the dynamics more susceptible to instability, where the dominant gradient contribution shifts from one class to another at each step. Therefore, finding an intermediate learning rate value between these two regimes becomes non-trivial, complicating hyper-parameter tuning procedures. It is interesting to note that even in this scenario, curves computed on balanced datasets in the presence of IGB show similarities to the behavior observed in the presence of class imbalance (see Fig. 15).

Conversely, the fact that IGB creates an imbalance between classes in balanced datasets opens up the possibility of exploiting the phenomenon to counteract the effects of class imbalance for the dynamics' benefit by acting on the network's design rather than on the dynamics itself.

### I.4  REPRODUCIBILITY

Here we provide technical details about the experiments, to allow for reproducibility. The code used for the experiments presented in this work are available at https://anonymous.4open.science/r/Algorithms-82EC/README.md.

**Datasets**

- **Gaussian Blob** (GB): Consistent with the analysis presented, for most of the experiments shown we used a Gaussian blob as input. Specifically, all elements of the dataset are *i.i.d.* and each individual element consists of a random vector of $d = 3072$ *i.i.d.* normally distributed components, *i.e.*:

$$\xi_b^{(a)} \sim \mathcal{N}(0, 1)$$

  Note that the value of $d$ is chosen so that the random vectors we generate have the same dimension of CIFAR10.

- **CIFAR10** (C10): We use CIFAR10 (https://www.cs.toronto.edu/~kriz/cifar.html) (Krizhevsky et al., 2009) as an example of a real multi-class dataset. Before the start of the simulation we perform the standardization of the dataset: the pixel values are rescaled in the interval $[0, 1]$ and then shifted by the mean value and rescaled by the standard deviations (calculated on each channel).

- **CIFAR100** (C100): We use CIFAR100 (https://www.cs.toronto.edu/~kriz/cifar.html) (Krizhevsky et al., 2009) as an example of high cardinality dataset, *i.e.* a dataset with a big number of classes

- **MNIST** (E&O): We use MNIST (http://yann.lecun.com/exdb/mnist/) (Deng, 2012) to reproduce binary experiments on real data. The binary dataset is defined by merging the starting classes into two macro groups according to the parity of digits; thus, we will have the even number class $\{0, 2, 4, 6, 8\}$ and the odd number class $\{1, 3, 5, 7, 9\}$.

**Models**  We here provide details on the architectures we used for our experiments. Our description of the models does not include the loss functions because the loss function is irrelevant in untrained networks.

- **MLP**: Our analysis provides theoretical predictions for MLP. In order to support the results of the study, we considered two different MLP networks in the proposed experiments:
  - **MLP with a single hidden layer** (SHLP): The number of nodes nodes in the hidden layer varies between $N_1 = 100$ for networks without Pooling and $N_1 = 500$ for networks equipped with MaxPooling layer. Different activation functions have been coupled to the networks to show the differences; details regarding the choice of these elements are given case by case.
  - **MLP with multiple hidden layers** (MHLP): In this case we have $L$ hidden layers, each one composed by $N = 100$ nodes. As in the previous case, the activation function is an element we varied to set a comparison and underline the differences coming from this choice.

- **CNN**: To show how IGB manifests outside the setting employed in the quantitative treatment presented we propose, some experiments on convolutional neural networks (CNNs) as an alternative to the MLPs discussed above. In particular, we used:
  - `CNN-A`: This architecture was used for simulations related to the histogram in Fig. 1; two classes from CIFAR10 (three-channel images) were selected for these experiments. Starting from the input layer we have:
    * a first convolutional layer with: out channels=16 (Number of channels produced by the convolution), $m = 5$ ( Size of the convolving kernel), stride=1, padding=2. The output of the layer is then passed through an activation function and a pooling layer. The choice of these elements varies to compare two different scenarios (as explained in Fig. 1). In both cases, however, we set common parameters for the pooling layer, *i.e.* $m = 2$ (kernel size), stride=2.
    * Next comes a second convolutional layer with the same parameters as the previous one, except for the number of output chanels; in this case we have out channels=64. Again the convolutional layer is followed by an activation function and a pooling layer (same as the first layer). The parameters of the pooling layer in this case are *i.e.* $m = 4$ (kernel size), stride=4. The processed signal is then connected with a weights layer to the output layer.
  - `CNN-B`: We also consider a second CNN architecture deeper than `CNN-A`. Specifically starting from the input layer:
    * we start with a sequence of five convolutional layers (each followed by an activation function and pooling layer). Except for the number of output channels the rest of the parameters are fixed the same for each of these layers, in particular we have for the convolutional layer $= 5$, stride=1, padding=4. For the pooling layer, however, $m = 5$, stride=1. Finally, the number of output channels for the various layers is $[16, 32, 32, 64, 32]$.
    * This is followed by an additional convolutional layer defined by the following parameters: out channels=16, $m = 5$, stride=1, padding=2. This is accompanied by the activation function and a Pooling layer whose parameters are: $m = 2$, stride=2.
    * Finally, a last sequence of convolutional layers, activation function and pooling layer precedes the output layer. The only parameter that differs from the sequence that precedes it is the kernel size of the pooling layer; specifically in this case $m = 4$. The processed signal is then connected with a weights layer to the output layer.
  - **ResNet**: We propose ResNet34 (`ResNet`), introduced in He et al. (2016) as an example of an architecture equipped with skip connections.
  - **Vision Transformer**: We propose ViT (`ViT`), introduced in Dosovitskiy et al. (2020) as an example of an architecture equipped with skip connections.

Note that in both the CNN architectures we use, the final convolutional layer is directly connected to the output, without any additional fully-connected hidden layer (as the ones described by our theory). This indicates that the observed IGB is also also a feature of CNNs independently of the presence of fully-connected hidden layers at the end of the network (which as we proved would also cause IGB).

## J  LIMITATIONS AND ETHICS

**Limitations**   We can identify the following limitations to our work:

- Our work focuses on systems simple enough to clearly show the main aspects of the phenomenon and at the same time complex enough to investigate non-trivial effects induced, for example, by network depth. A comprehensive picture emerges that clarifies the effect of some particular elements of the architecture and their connection with IGB. On the other hand, the observation of IGB is not restricted to the subset of networks/datasets considered in the study. Although the treatment of more articulated setups is outside the scope of the study, whose main goal is to present the phenomenon (which to the best of our knowledge has never been reported in the literature) in a clear manner, the characterization of IGB for more realistic systems remains an interesting question.

- Our analysis suggests that some forms of regularization (*e.g.*, batch normalization) might be effective in mitigating IGB. To draw conclusions in this regard, however, an in-depth study comparing different kinds of normalization is needed: note how such regularizations are not included in our simulations.
  If architectural elements such as batch normalization help suppressing IGB, this would be a further step towards an aware choice of the architectures, allowing to choose whether to have initial conditions with IGB or not.

**Ethics**  By informing model selection, data preparation and initial conditions, our results can improve the training of machine learning models. Better-performing machine learning models allow to better address wide ranges of problems, but can also be adapted for potentially harmful applications (Hutson, 2021; Qadeer & Millar, 2021).

The CIFAR datasets, are subsets of the 80 million tiny images, which are formally withdrawn since it contains some derogatory terms as categories and offensive images (`http://groups.csail.mit.edu/vision/TinyImages/`). However, note that none of the experiments described in the paper was performed on the overall tiny images dataset: the said derogatory images are not present in CIFAR10 nor CIFAR100.

