# OpenReview forum: "Untrained Networks' Class Bias: A Theoretical Investigation"
_ICLR.cc/2024/Conference — ICLR 2024 Conference Withdrawn Submission_

### Official Review · Reviewer_8Cof · 2023-10-27

**Soundness:** 3 good
**Presentation:** 1 poor
**Contribution:** 2 fair
**Rating:** 3
**Confidence:** 4

**Summary:**

This paper studies the distribution of untrained neural network outputs in a binary classification task. The paper considers Gaussian random inputs and Gaussian random weights in the neural network. By defining a phenomenon named ”Initial Guessing Bias” (IGB), the paper shows that while linear activation introduces no IGB, adding ReLU and Maxpooling will result in IGB. Moreover, IGB is intensified by increasing the number of layers in the neural network.

**Strengths:**

The notion of IGB introduced in this paper is new. Based on this notion, the paper has several interesting results:

1. The paper shows that, while linear activation gives output distribution centered at one-half (balanced label output), using the ReLU activation actually shifts the distribution away from one-half, even when the width approaches infinity.

2. The paper also shows that the randomness in the weight initialization dominates the randomness in the dataset when the depth of the network grows to infinity (as in Eq. 15).

**Weaknesses:**

1. The writing of the paper needs to be greatly improved:

    a). Since this is a theoretical paper, it is astonishing to me that there are no theorems stated in the paper. All the theoretical results in the paper are discussed verbally instead of rigorously in formal theorems. It is not clear under what hypothesis these results hold, and whether some equations are formal results or intuitive deductions. There is no mathematical formulation of the neural network considered in the paper so I am not sure what is the exact setting of the paper’s result.

    b). Terms used in the paper need to be described precisely. For instance, in the second paragraph on Page 5, the paper writes $\mathbb{P}_{\mathcal{X}}(O^{(c)}|\mathcal{W})$ is asymptotically a Gaussian whose center $\langle O^{(c)}\rangle$...”. I believe that in this case, the paper means that $O^{(c)}$ is a random variable following a Gaussian distribution, not that the probability is a random variable following a Gaussian distribution.

    c). Another example is Eq. 9, where I am not sure what is $\mathcal{N}(y; \Delta_{\mu}(f_0), 2\sigma^2_{\infty})$, and I am not sure what the author means to integrate a distribution $\mathcal{N}(y; \Delta_{\mu}(f_0), 2\sigma^2_{\infty})$. Is the author somehow using $\mathcal{N}(y; \Delta_{\mu}(f_0), 2\sigma^2_{\infty})$ as the pdf function?

    d). Notations of the paper need to be introduced rigorously and mathematically. For instance, the only description I found about $f_c(\mathcal{W})$ is that it denotes ”the fraction of points classified as class $c$”. However, it is not clear whether this ”fraction” refers to the average over a finite number of data points or the expectation. The introduction of notations should be gathered somewhere instead of randomly popping up throughout the paper.

    e). In terms of formatting, the paper needs to add spacing between paragraphs for better readability.

2. It is not clear what is the significance of the topic (IGB) studied in this paper. The author tried to explain why it is important but provided no concrete examples and no previous works that directly consider the impact of IGB. How IGB is related to the following training process is also not described in the paper (other than an experiment in the appendix that I will raise concerns about below).

3. The assumption of applying max-pooling on random Gaussian input is weird since max-pooling is usually used to deal with images. In particular, if the input vector has no internal structure that allows dimensionality reduction tricks like max-pooling, it would not be so surprising that max-pooling will cause bad results such as IGB.

4. In Figure 11 the paper shows that CIFAR-10 has IGB on ResNet with ReLU and max-pooling. In Figure 15 the paper shows that with IGB the training process cannot achieve a recall higher than 0.7. Are these two results contradicting the well-known case that ResNet with max-pooling and ReLU works well (although not the best) on CIFAR-10? See e.g.: https://www.kaggle.com/code/kmldas/cifar10-resnet-90-accuracy-less-than-5-min

**Questions:**

Please see "Weaknesses" above.

---

> ### Author Response · Authors · 2023-11-22
>
> To avoid redundancy, we refer the reviewer to the general response, to which we'll now add further clarifications addressing more specific doubts.
> ## Reply to W.1
>
> -  b) Yes, the interpretation of the reviewer is correct. $\mathbb{P}\_{\chi}(O^{(c)} | \mathcal{W})$  is a Gaussian distribution, or equivalently, the random variable $O^{(c)}$, conditioned on the fixed realization of the weights $\mathcal{W}$, is Gaussian distributed. We appreciate the reviewer's comment and, in the new version, we will opt for this second formulation to make the point clearer, *i.e.*  $O^{(c)}_{| \mathcal{W}} \sim \dots$
> -  c) Again, the interpretation is correct: $\mathcal{N} (y \text{;} \Delta_{\mu} (f_0), 2 \sigma^2_{\infty})$ indicates a probability density function that we integrate. The used notation is extensively detailed in App. A. We will provide additional references that connect the various sections to the appendix throughout the text.
> - d) In the paragraph 'Permutation Symmetry Breaking: the foundation of IGB,' we introduce the quantity $f_c$ by defining it as 'the fraction of datapoints classified as class $c$.' This definition applies to both finite and infinite datasets. The fraction, in general, is computed over the datapoints that comprise the dataset (not an expectation over the input distribution). However, we are particularly interested in the limit of datapoints approaching infinity, in which the fraction indeed converges to the value averaged over the input distribution.\
> Considering the limit of infinite datapoints, we can then connect the quantity $f_c$ to the output distribution, derived in our analysis, through the law of large numbers (Eq.1). \
> The datasets used in practice, although large, are clearly of finite size; hence, we encounter finite-size effects that perturb the asymptotic prediction. Nevertheless, this does not pose an obstacle in terms of predictability. Our analysis is indeed capable of effectively describing these effects by accounting for the aforementioned corrections (as in Appendix C.1 (linear case)). Various plots in the paper display these corrections. For instance, the theoretical curve overlaid on the histogram of the linear case in Fig. 5 accounts for finite-size effects, which, if absent, would lead the distribution to converge to a spike at 0.5.\
> We kindly note that the notation is indeed grouped in App. A for ease of reference; however, we will follow the reviewer's advice and provide a more rigorous definition of the introduced quantities. If there are further unclear points in this regard, please, do not hesitate to highlight them. We would be more than happy to attempt to clarify.
>
>
> ## Reply to W.3
>
> We take advantage of this observation to clarify a fundamental point in our discussion that might have been misunderstood. It is not our objective to propose a narrative where IGB is unequivocally associated with a harmful effect. The article's objective is rather to highlight the presence of a non-trivial phenomenon and understand its nature through a rigorous analysis. We agree that the question of whether IGB is a desirable effect or not is inherently interesting; however, providing a complete answer to this question requires a dedicated, systematic study since the response is neither simple nor absolute (it might depend on the problem's nature whether the presence of IGB constitutes an advantage or an obstacle). From our standpoint, what our experiments aim to show is that certainly the presence or absence of IGB qualitatively modifies the dynamics (for example, by creating differences in the statistics of individual classes, which are not present in the absence of IGB). We focus on certain regimes that can naturally occur, for instance, during hyper-parameter tuning procedures or sometimes assumed as hypotheses (the assumption of a low learning rate is, for example, a fundamental assumption in many works). In such regimes, the difference between the two compared cases is evident.
>
> The analysis with MaxPooling helps to illustrate, with a clear example, how the presence of an architectural element of this kind, by amplifying the asymmetry between nodes, in turn amplifies the presence of IGB. This allows us to understand, within the framework of our theory, where a profile similar to that observed experimentally (for example, Fig. 1 of the paper) originates.
>
>
> ## Reply to W.4
>
> We would like to point out that the experiments in Fig. 15 do not refer to a ResNet but to a CNN. However, as mentioned in the response to the above point, the message these experiments aim to convey does not relate to a comparison of the performances achieved in the two cases. Instead, we aim at highlighting differences that make the two dynamics qualitatively distinct.
> In the new version, we will be clearer about this point to avoid confusion.

---

### Official Review · Reviewer_nZE9 · 2023-10-27

**Soundness:** 2 fair
**Presentation:** 3 good
**Contribution:** 2 fair
**Rating:** 5
**Confidence:** 3

**Summary:**

**Update after rebuttal:** I think the authors have made improvements to the paper that warrant an increase in my rating - I still do not think that the work is fully ready for ICLR, but I personally move away from a Reject to a Marginally Below. More detailed comments in my response to the authors' rebuttal.

The paper investigates the architectural factors which lead randomly initialized neural networks to have classification biases. A network is defined to have ‘initial guessing bias’ if the untrained network, across random initializations, favors one class over another in a binary classification task (with straightforward extension to the multi-class classification setting). The paper shows theoretically and empirically, how IGB arises in MLPs with common activation functions on synthetic data, and (in the appendix) on real-world benchmark datasets and architectures such as CNNs, ResNets, and VisionTransformers. The main takeaway is that the choice of activation function and use of MaxPooling cause IGB; deeper networks and use of real-world data amplify existing IGB of an architecture. Based on preliminary experiments in the appendix, the effects of IGB on training dynamics seem to be marginal and manifest mainly in the early training phase.

**Strengths:**

**Originality, Clarity:** To the best of my knowledge the phenomenon of IGB has not been investigated in detail before. The paper provides a definition of IGB and, somewhat counterintuitively, shows that commonly used neural network architectures and weight-initialization schemes lead to IGB. The (main) paper focuses on the theory behind IGB, and an empirical evaluation on synthetic data, which are both original. The paper presents the theoretical derivations in detail and with clarity, and the results look sound and correct to me. The appendix adds a lot of detail (to the point where it could perhaps use a bit of trimming).

**Pros:**
 * Formal Definition of IGB via the fraction of data that gets assigned to one class across random initializations (without training).
 * Theoretical analysis of the conditions for IGB to occur (mainly the choice of activation function and whether to use max-pooling or not).
 * Empirical verification of the theory on synthetic data and MLPs - the results match the theory very well.
 * (In the appendix) Analysis of other commonly used architectures on standard benchmark datasets.

**Quality and Significance:**
The significance of the results is currently unclear. While the paper leaves no doubt about IGB as a phenomenon, it does not sufficiently address the question of whether the phenomenon has any substantial effect in practice. The appendix shows some simple results, suggesting that training dynamics for networks with IGB differ very early in training and with low learning rates. But since virtually all commonly used architectures suffer from IGB, and there is no indication that training with IGB leads to lower final performance or significantly slower convergence, the impact of the results currently seems rather limited. I would rate the quality of the paper as good, except for the main motivation of the criterion of IGB: it is unclear why having no IGB should be a desirable criterion for neural networks. A derivation from first principles, and/or a theoretical justification in terms of how IGB affects training dynamics (particularly final accuracy and convergence speed) would significantly strengthen the paper. See more details under Weaknesses.

**Weaknesses:**

Disclaimer: I have reviewed a previous version of the manuscript for another conference. While I am happy to see that the paper has been improved, some of the main concerns in terms of significance and quality have not been addressed.

**Cons:**
* Significance: show that IGB actually has a significant effect on training dynamics beyond the very initial phase. The first sentence of the abstract (and other parts of the paper) build the expectation that IGB will have a significant effect on training dynamics; but no such results are shown in the main paper. The effects on training dynamics are not shown until page 46 in the appendix, and the results shown suggest a rather marginal effect on very early training dynamics, but nothing to worry about in terms of final classifier performance or overall convergence speed. Confirming these intuitions would be a result in itself, and would add to the paper. Some theoretical results on how IGB impacts training dynamics would potentially lead to a very strong publication. Without a thorough empirical investigation of training dynamics (which is promised to be upcoming in another publication) I am not fully convinced that ICLR is the right venue, but would suggest a more specialized venue or workshop.
* Quality: The paper assumes that absence of IGB is naturally a desirable property of good neural architectures. This claim (though somewhat intuitive) is never substantiated or derived from first principles. I believe that the claim can be motivated and formalized in general terms from first principles via a maximum entropy argument, which would increase the soundness of the paper. Formally the argument is to find an architecture that maximizes $H(Y|W,X)$ where $y$ are network outputs, $x$ are network inputs and $w$ are network weights. Maximum entropy $H(Y|W,X)$ is achieved when $p(y|w,x)=1/N_c$, meaning uniform over all $N_c$ classes for each weight configuration (drawn from the initialization distribution) and each input. As a corollary, if the condition just stated is satisfied in classification, $1/N_c$ of the inputs are assigned to class $C$ (in expectation) across all weight configurations, meaning that the architecture has no IGB following Definition 3.1 in the paper. Note, however, that the max. ent. criterion strictly requires uniform output probabilities for all inputs and all weights; which is stricter than requiring uniform assignment of datapoints to each class (at least theoretically, the network could be very “sure” that half of the datapoints belong to class 0 and the other half belongs to class 1, but with different sets of datapoints for each weight-initialization; the max. ent. criterion requires class probability $1/2$ for all datapoints). Many of the derivations in the paper would look quite similar if IGB were formulated in terms of requiring max. entropy (at least in the discrete case, and probably also the Gaussian case should go through; in the general continuous case differential entropy when analyzing deterministic reversible maps diverges). Additionally, since $H(Y|W,X) \leq H(Y|X)$; and we maximize $H(Y|W,X)$ we have $H(Y|W, X)=H(Y|X)$ and $I(Y;W|X)=H(Y|X)-H(Y|W,X) = 0$. Meaning the channel capacity between weights and outputs for the initialized networks is zero; which could be an alternative motivating first principle for the maximum entropy criterion: the weights carry no information of any order (even beyond class-assignment information) about the outputs. To me, the maximum entropy principle is more general and better motivated from first principles compared to the IGB criterion in the paper. I am very happy to hear arguments in favor of sticking to requiring uniform class assignments instead of uniform output probabilities.
* Impact and claims: (this is a bit of a repetition of my first weakness, the emphasis here is to place training dynamics results into the main paper) I want to encourage the authors to show the impact on training dynamics in the main paper and perhaps move some of the derivations to the appendix. I understand that the goal now is to write a theory paper, but I think that the ICLR audience would greatly appreciate an analysis of the impact on training dynamics in the main paper. I would even want to encourage the authors to analyze final classifier performance and training convergence speed - even if the presence or absence of IGB does not affect the latter two, that would be an interesting result in itself and I would be very curious to see it reported.

**Final Verdict:**
I think the phenomenon of IGB is interesting and the main idea in the paper is original and results are surprising. However, I think the current paper can be significantly improved. I am leaning towards suggesting another revision of the manuscript - unfortunately ICLR only allows a rating of 3 (reject) or 5 (marginally below). On a free scale I would give the paper a 4, but since it is not possible I have assigned a 3 for now. I am curious to hear the authors response and other reviews, and am happy to change my score accordingly (and I do believe there's a good chance that my issues can be mostly addressed in the rebuttal).

**Questions:**

No further questions. Suggestions stated in Weaknesses.

---

> ### Author Response · Authors · 2023-11-22
>
> To avoid redundancy, we refer the reviewer to the general response, to which we'll now add further clarifications addressing more specific doubts.
>
> ## Reply to the suggestion of the entropy formulation
>
> We thank the reviewer for the insightful and helpful comment. We would like to take this opportunity to clarify a fundamental point. Through our simulations, our aim was to illustrate that the presence of IGB leads to a qualitatively different dynamic. It's not our intention to assert whether IGB is beneficial or detrimental; the answer to this question is entirely non-trivial and potentially problem specific.
>
> We find the idea of formulating this in terms of entropy quite intriguing. An additional point not highlighted by the reviewer is that it would enable us to associate a functional with the distribution, allowing quantification of the level of IGB with a single value.

---

> ### Comment · Reviewer_nZE9 · 2023-11-23
> **Thanks for the responses and continued engagement**
>
> Thank you for the responses. I do believe that the work has substantially improved since the very first version that I saw. With a bit more work, I think the authors are on track for a potential high-impact publication. I personally think it is worth putting in that effort rather than rushing a publication of limited impact. The current version of the manuscript seems ready for an ICLR workshop or a more specialised conference, which might be a good way for the authors to create some attention and engagement. Ultimately, I would be very excited to read a strong version of this paper (including a nice set of empirical results). I will raise my score accordingly, but would still rate the paper as marginally below the threshold for ICLR (which, admittedly, is very high).
>
> Re: illustration of IGB vs. beneficial/detrimental. Thanks for clarifying - I had definitely read the paper as advocating for designing network architectures that are free from IGB (which does seem to make "intuitive" sense). Clarifying early and very explicitly in the paper that this is not the case would be very helpful. Apologies for the misintrepretation.
>
> Re: quantifying IGB with a single value. Yes, thanks for pointing this out. Another reason to see if the IGB criterion can be formulated more generally. (The entropic formulation might also translate fairly straightforwardly to regression tasks, which is not true for the classification criterion of course) Since the current IGB criterion requires collecting lots of class assignments across datapoints and weights, experimentally estimating the conditional entropy by collecting logit vectors in addition to class assignments should only lead to minimal extra effort.

---

### Official Review · Reviewer_GMZV · 2023-10-28

**Soundness:** 3 good
**Presentation:** 2 fair
**Contribution:** 1 poor
**Rating:** 3
**Confidence:** 4

**Summary:**

This paper studies the phenomenon of ‘Initial Guessing Bias’ (IGB), which means that the initial prediction of a NN is usually biased toward one single class rather than averaged across all classes, when the task is classification. The authors show that this is empirically true, and focuses on the theory behind it. In particular, they prove that for a 2-layer or a deep NN, the phenomenon provably happens even if the network is infinitely wide and the dataset is infinite. They argue that this phenomenon has some influence on the initial phase of training as well.

**Strengths:**

- They provide very detailed analysis of several settings, including 2-layer linear network, relu network, relu with max pooling, and deeper versions.
- The fact that max pooling exacerbates the IGB phenomenon is very interesting.

**Weaknesses:**

- My main complaint about this work is that the problem doesn’t seem interesting. I don’t see why the initial bias matters, and the authors also fail to provide convincing evidence that this bias leads to meaningful damage to the trained result. In some sense, I believe this bias can be quickly corrected during the course of training.
- There could be more empirical evidence for why this bias matters and how it evolves during training. Right now the authors mainly focus on the theory.
- There’s no outline of the proof, making it hard to tell if the underlying analysis is trivial or not. It would be good if the authors can provide some sketch on the most interesting parts of the analysis.

**Questions:**

- In figure 1, when there’s no IGB, shouldn’t it be the case that all f_0 is centered around 0.5? It’s unclear to me why there’s still a distribution.

---

> ### Author Response · Authors · 2023-11-22
>
> To avoid redundancy, we refer the reviewer to the general response, to which we'll now add further clarifications addressing more specific doubts.
>
> ## Reply to Q1
>
> Correctly noted, in the absence of IGB, we anticipate the distribution to tighten around a single value (in the case of a binary dataset like that in Fig.1, $f_0 = 0.5$). However, it's crucial to note that this convergence of the distribution to the true value formally occurs only in the limit of infinite datapoints (see also Definition 3.1). In our simulations on finite datasets, we never actually observe a spike at a single value but rather a distribution concentrated around it.
>
> Through our analysis, we go further to quantify these finite-size effects, resulting in theoretical curve profiles to compare with empirical histograms (for instance, the linear case in Fig.5).\
>  For more details regarding the analysis concerning corrections due to finite-size effects, refer to, for example, App.C.1, the paragraph 'Linear' (In Fig. 6, the effects of finite size are depicted as the number of data points increases, showing the curves derived in App.C1.).

---

### Official Review · Reviewer_uugM · 2023-10-28

**Soundness:** 2 fair
**Presentation:** 2 fair
**Contribution:** 3 good
**Rating:** 3
**Confidence:** 4

**Summary:**

This paper studied the initial state of neural network, which is an interesting question. They showed that the initial structure of neural network can condition the model to assign all predictions to the same class, which they called 'initial guessing bias (IGB)'. They also showed how architectural choices affect IGB.

However, I think there exist serious technique problems in this paper. I highly recommend the author to give more convincing proofs and more clear statements of their results.

**Strengths:**

The author proposed a very interesting phenomenon, IGB, in the initialization of neural network. Their results are insightful.

**Weaknesses:**

The authors' technical arguments and proofs are quite ambiguous, especially considering this is a theoretical paper.

1. On the use of limit symbols. For example, in equation 7&8, is $\mathcal{W}$ a deterministic, infinite array? If it is, under which conditions (of $\mathcal{W}$) equation 7&8 holds? Besides, I notice that in the appendix C, the authors say that the dimension of input variable $d$ must increase with $N_1$ in a larger order (equation 69&70) to obtain equation 7&8. I think it's necessary to write such an important asymptotic condition above the limit arrow in equation 7&8.
2. I think that the authors' arguments of the independence are very non-rigorous. In the Remark 1 in Appendix C, the authors try to argue $\{h_i^{(1)}\}_{i=1}^{N_1}$ are independent. This independence is indeed directly used later as the core of their analysis. However, their arguments about this independence are highly intuitive rather than a rigorous proof. How can "w.h.p." pairwise orthogonality (equation 64-68) implies asymptotic independence? Their arguments about the asymptotic behavior of $N_1$ and $d$ are also very causal. Does equation 69 insure that these random vectors are orthogonal? I think the authors should use some rigorous mathematical tools (such as random matrix theory) to give a convincing proof before using these arguments.
3. Besides, from their theoretical results I still don't understand where the phenomenon in Figure 1 arises. Even if the output is a somehow "wide" Gaussian, I don't know how it leads to Figure 1.
4. Indeed, I have conducted a very toy experiment on a one-dimensional, two-layer, fixed-width relu MLP, and obtain similar figures like Figure 1. The only difference is my MLP includes also bias as its parameter. Therefore, the IGB could not be an asymptotic phenomenon and I hope to get some reasonable analysis.

**Questions:**

Please see weaknesses.

---

> ### Author Response · Authors · 2023-11-22
> **Reply to Reviewer uugM (part 1)**
>
> To avoid redundancy, we refer the reviewer to the general response, to which we'll now add further clarifications addressing more specific doubts.
>
> ## Reply to W.1
>
> $\mathcal{W}$ indicates the entire set of weights. The weights are then organized into matrices that connect the various layers. In equations 7 and 8, we have expressed, for compactness reasons, the limit in terms of the divergence of the total number of parameters (cardinality of the set $\mathcal{W}$). As noted by the reviewer, and as we ourselves acknowledge in the text, this limit is ambiguous; we opted for this choice in order to maintain compactness and not burden the introduction of the phenomenon with technical details, which we have instead moved to more advanced sections of the document.
> Formally, for the single hidden layer network, the conditions we require (for the use of the Central Limit Theorem in our analysis) are $d \rightarrow \infty$ and$N_1 \rightarrow \infty$.
>
> ## Reply to W.2
>
> We appreciate the comments provided by the reviewer and acknowledge the need for a more rigorous treatment of the independence of the components in the given representation. In response to this feedback, we have rephrased the independence argumentation for clarity and rigor. Below, we present a concise summary of the key steps of the revised proof that will replace the arguments in Remark 1 in the updated manuscript.
> Consider the pre-activated nodes,
> $$h^{(1)}\_{i} = \\sum\_j w^{(0)}\_{ij} \xi_j \\, .$$
> Given the independence of $\boldsymbol{\xi}$ components, we can employ the Central Limit Theorem (CLT) to derive the distribution of $h^{(1)}\_{i}$.
>
> Since we also utilize the CLT for
> $$ O^{(c)} \left( \boldsymbol{\xi} ; \mathcal{W} \right) = \sum\_{m=1}^{N\_1} w^{(1)}\_{cm} g^{(1)}\_{m} \\, ,$$
> independence must also be established for the set $\\{ g^{(1)}\_{m} \\}$. As the activation function is an element-wise transformation of the set $\\{ h^{(1)}\_{m} \\}$, it suffices to ensure the independence of the latter. A well-known fact about jointly normally distributed random variables is that they are independent if and only if their covariance is zero.
> We then proceed to calculate the covariance $\textrm{Cov} (h^{(1)}\_i, h^{(1)}\_j)$ for $i \neq j$ . The covariance, in general, will be a function of the weights $w^{(1)}\_{i \cdot}$ and $w^{(1)}\_{j \cdot}$. However, we demonstrate that in the limit as  $d \rightarrow \infty$, $\textrm{Cov} (h^{(1)}\_i, h^{(1)}\_j)$  converges with high probability (*w.h.p.*) to 0.\
> Indeed
> $$
> \textrm{Cov} \left( \sum\_{m=1}^d w^{(1)}\_{i m}  \xi\_m, \sum\_{n=1}^d w^{(1)}\_{j n} \xi\_n \right)  \stackrel{\boldsymbol{a}}{=} \sum\_{m=1}^d \sum\_{n=1}^d w^{(1)}\_{i m} w^{(1)}\_{j n} \mathbb{E}(\xi\_m \xi\_n)\\
> \stackrel{\boldsymbol{b}}{=} \sum\_{m=1}^d w^{(1)}\_{i m} w^{(1)}\_{j m} \mathbb{E}(\xi\_m^2) \stackrel{\boldsymbol{c}}{=} \sum\_{m=1}^d w^{(1)}\_{i m} w^{(1)}\_{j m} .
> $$
> Here, step $\boldsymbol{a}$ follows from linearity and the fact that $\mathbb{E}(\xi_m) = 0 \\; \\;  \forall m$. Steps $\boldsymbol{b}$ e $\boldsymbol{c}$ directly result from substituting the covariance matrix of $\boldsymbol{\xi}$, given that $\boldsymbol{\xi} \sim \mathcal{N} (0, \mathbb{I})$.
> Regarding the final summation, each generic term comprises a product of *i.i.d.* variables distributed according to a Gaussian distribution with zero mean and variance $\mathcal{O} \left( \frac{1}{d} \right)$. Therefore, it follows from the CLT that, in the limit as $d \rightarrow \infty$, the summation converges, *w.h.p.* , to 0.

---

> > ### Author Response · Authors · 2023-11-22
> > **Reply to Reviewer uugM (part 2)**
> >
> > ## Reply to W.3
> >
> > The phenomenon of IGB is essentially linked to a breakdown in the symmetry of the distributions of nodes belonging to a given layer. By 'breakdown of symmetry,' here we mean that the nodes in the respective layer are not identically distributed. In particular, these distributions are Gaussian and centered at different points. Since each data point is assigned to the class associated with the largest output, the output node centered on the largest value will be assigned a greater number of data points.
> >
> > In the paragraph titled 'Permutation Symmetry Breaking: the foundation of IGB', we aim to explain the intuition behind the phenomenon. Specifically, in Figure 4, the aforementioned explanation is visually represented through the comparison of two extreme cases.
> >
> > ## Reply to W.4
> >
> > We thank the reviewer for the insightful and valuable comment. In our analysis, the step of sending the width of the layers (as well as the dimension of the inputs) to infinity is not essential for the emergence of the phenomenon but rather for its analytical description. In other words, while, for instance, the nontrivial profile of the distribution of $f_0$ also appears in finite networks, the convergence of such profiles to the theoretical distributions derived from our analysis is exact only within the aforementioned limits.
> >
> > Beyond theoretical rigor, our experiments also demonstrate a good match with the asymptotic theoretical curves for moderate values of the number of nodes per layer (below 100 units).

---

### Author Response · Authors · 2023-11-22
**General Reply**

We are sincerely grateful to the reviewers for the helpful feedback. We are pleased to note that all reviewers have shown interest in the phenomenon addressed in our study and its related analysis. From our understanding, the comments from the reviewers focus on the following two main aspects:

-  **Presentation style** : Some reviewers suggest a more concise presentation for the main paper, with formalized theorems in a compact form and outlines of proofs (with formal detailed proofs in the appendix). \
  While our study is theoretical in nature, given the novelty of the presented phenomenon, we aimed to give the main paper a more narrative style to make the content accessible to a wider audience and to highlight the intuition and practical implications linked to our analysis conclusions. For instance, this led us to center the narrative around $f_0$, which provides a clearer and more easily interpretable measure compared to output values. \
However, while maintaining clarity, we acknowledge the importance of maintaining rigor in presenting the results. Taking the reviewers' feedback into account, we will seek a balance between these two versions where the results are presented clearly and concisely in the main paper, accompanied by an equally concise, intuitive interpretation that helps connect the results to their implications.

-  **Relevance of the phenomenon** : Another aspect highlighted by multiple reviewers concerns the impact of IGB on dynamics. \
  It's essential to establish a premise. IGB is a new and non-trivial phenomenon challenging some 'natural' intuitions regarding untrained networks, making a theory that can fully model and explain the origin of the phenomenon inherently relevant. For instance, in the presence of IGB, some key measurements on a typical single initialized network differ substantially from estimations over the ensemble of initializations. It also demonstrates how class performances can exhibit strong heterogeneity at the beginning of training, an effect not attributable to the class characteristics themselves (observable even in identically distributed classes and unstructured datasets). If the dynamics proceed gradually (the small learning rate regime forms the basis of many studies), this gap between classes can persist for a considerable period. Hence, disregarding the presence of IGB may lead to incorrect assumptions due to the counter-intuitive effects associated with the phenomenon.\
We agree with the reviewers that demonstrating the impact of IGB on subsequent dynamics would significantly enhance the relevance of the study. As highlighted by reviewer nZE9, the outcome would be intriguing whether the imbalance persists or is quickly absorbed, as it would raise questions about the mechanism and conditions that enable a network's dynamics to absorb a strong predictive bias. Therefore, we are strongly committed to exploring this aspect, which is currently the subject of our ongoing study. While we plan to conduct an extensive theoretical and empirical study on the effect of IGB on dynamics as a standalone paper, we will further strengthen the empirical results currently reported in the appendix.

We acknowledge the validity of the reviewers' comments and thank them for their feedback. We believe that the received observations can further enhance the quality of the manuscript.
To avoid redundancy, we will limit our responses to addressing only specific questions raised by the reviewers regarding doubts or clarifications about passages in the text.